# NeuMoSync: End-to-End Neuromodulatory Control for Plasticity and Adaptability in Continual Learning

## Abstract

Continual learning (CL) requires models to learn tasks sequentially, yet deep neural networks often suffer from plasticity loss and poor knowledge transfer, which can impede their long-term adaptability. Drawing high-level inspiration from global neuromodulatory mechanisms in the brain, we introduce **Neu**ro**Mo**dulation and **Sync**hronization (NeuMoSync), a novel architecture that integrates dynamic, neuron-specific modulation into deep neural networks to enhance their adaptability and plasticity. NeuMoSync extends standard neural network architectures with learnable feature vectors per neuron that tracks network-wide historical context and incorporates a module operating at a higher level of abstraction. This module synthesizes neuron-specific signals, conditioned on both current inputs and the network's evolving state, to adaptively regulate activation dynamics and synaptic plasticity. Evaluated on diverse CL benchmarks, including memorization (Random Label CIFAR-10, Random Label MNIST), concept drift (Shuffle CIFAR-10), class-incremental (Class Split T-ImageNet, CIFAR-100) and domain-incremental (Permuted MNIST), NeuMoSync demonstrates strong performance in terms of retention of plasticity and achieves improvements in both forward and backward adaptation compared to existing methods. Ablation studies validate the necessity of each component, while the analysis of the learned modulatory signals reveals interpretable coordination patterns across tasks. Our work underscores the potential of integrating global coordination mechanisms into deep learning systems to advance robust, adaptive continual learning.

## 1 Introduction

Continual learning (CL) requires models to acquire knowledge sequentially (49; 35; 54), retaining past information while efficiently adapting to new tasks. A significant challenge for deep neural networks in this setting is the loss of plasticity observed over time, which severely hampers their ability to learn task sequences (16; 47; 48). In contrast, biological agents excel at both memory retention and fast adaptation by leveraging local plasticity (90; 23; 71) alongside global modulatory systems that dynamically coordinate neural circuit behavior (53). Mediated by neuromodulators, these global mechanisms tune synaptic efficacy, excitability, and learning rules across the network (6; 79; 19), allowing flexible adaptation to shifting environments without forgetting prior knowledge.

Although artificial neural networks (ANNs) were originally inspired by biological neurons (50; 63), they only loosely resemble biological learning systems. ANNs typically implement simplified local neuron-to-neuron interactions and omit the context-dependent global feedback loops seen in the brain (53; 19; 21). Neuroscience has shown that neuromodulation plays a crucial role in adjusting learning and behaviour in response to uncertainty, novelty, or reward prediction errors (32; 53). Moreover, subthreshold dynamics and homeostatic processes further contribute to maintaining a functional balance between plasticity and stability (79; 7; 23). Together, these mechanisms enable biological agents to continuously adapt to changing environments, a capability missing in deep neural networks. This gap prompts a fundamental question: *which elements of the brain's communication framework can we utilize to improve continual learning with deep neural networks?* One compelling hypothesis is that the brain employs complementary signaling pathways, extending beyond mere spike transmission, to orchestrate connectivity, plasticity, and function at the system level. Emulating these

global coordination processes may be crucial for endowing artificial networks with more resilient continual learning capabilities.

Motivated by the idea that global coordination signals can facilitate flexible adaptation, we introduce NeuMoSync, an architecture that incorporates dynamic, neuron-specific modulation into deep networks through a centralized controller, the NeuroSync module. The controller generates modulation coefficients for individual neurons based on both the current input and a summary of the network's internal state, allowing the network to reconfigure its effective computation across tasks and contexts. These coefficients dynamically regulate feature activity, synaptic strength, and reliance on past knowledge across the network while remaining fully trainable end-to-end via standard gradient descent. By incorporating these higher-order control dynamics, our approach aims to prevent loss of plasticity and enable faster adaptation to new tasks in continual learning benchmarks. Importantly, to clarify the motivation and scope of our contribution, we do not aim to exactly replicate the brain's neuronal structures; rather, we take inspiration from them at an abstract level. Also in Appendix B, we provide a comprehensive review of related work, including prior approaches that incorporate gain or bias modulation in other settings.

We rigorously evaluated NeuMoSync through comprehensive experiments across a wide range of continual learning benchmarks, including memorization, concept drift, class-incremental, and domain-incremental scenarios. We measure novel metrics designed to quantify fast adaptation, the ability of a model to quickly adjust to new tasks or rapidly relearn past ones. Our results demonstrate that NeuMoSync preserves plasticity, excels at both forward and backward adaptation, with particularly large improvements in memorization tasks over continual learning baselines. Ablation studies validate the necessity of each component, while analysis of the learned modulatory signals reveals emergent behaviors that parallel biological observations. Importantly, ablations reveal that parameter sharing as the critical inductive bias for the global neuromodulatory controller. Collectively, these findings highlight the effectiveness of global coordination mechanisms in enhancing both the adaptability and robustness of deep neural networks in continual learning settings.

## 2 METHOD

**Notation.** For a neural network with parameters $\theta$, the variable $i$ will be used to index over the $n$ neurons. For example, $\theta^i$ denotes the weights associated to neuron $i$.

The NeuMoSync architecture is comprised of three core components, each with a distinct role. First is the MainNetwork, which focuses on acquiring task-specific knowledge by adapting rapidly to the current data stream. Complementing this is the ConsolidatedNetwork, a slow-moving average of the MainNetwork that serves as a long-term memory, designed to preserve more generalized knowledge consolidated across tasks. Finally, the NeuroSync module learns how to dynamically combine the knowledge from these two networks and modulate the resulting fused network to achieve the final prediction. Figure 1 provides a high-level overview of the NeuMoSync architecture. In this illustration, the MainNetwork is implemented as a multilayer perceptron (MLP) for simplicity.

The NeuroSync module functions as a centralized controller that operates at a higher level of abstraction than the MainNetwork. It generates top-down feedback signals by processing the current input sample along with feature vectors associated with each neuron in the MainNetwork. These signals are output as a vector of $\alpha$-parameters, which are then used to construct an effective InferenceNetwork for making predictions. This construction is achieved by dynamically modulating and combining the contributions of the MainNetwork and ConsolidatedNetwork, thereby controlling three primary components of each neuron: its synaptic weights, activation function, and post-activation offset.

We implement the NeuroSync module using an architecture that enforces parameter sharing across all neurons, an inductive bias we demonstrate is critical for effective global modulation in Section 3.5. Accordingly, while our primary model uses an encoder-only transformer, we confirm that the underlying principle is key, as an alternative 1D CNN architecture also yields similar performance, as shown in Section 3.5.2. We favor the transformer in our main experiments, as its parameter count does not scale with the size of the MainNetwork, ensuring a more consistent overhead. Furthermore, to ensure NeuroSync can be extended to larger-scale architectures, we introduce a more scalable variant of our controller in Section 3.7 that employs a sparse, sampling-based approach. We now describe the information flow of NeuMoSync in greater detail.

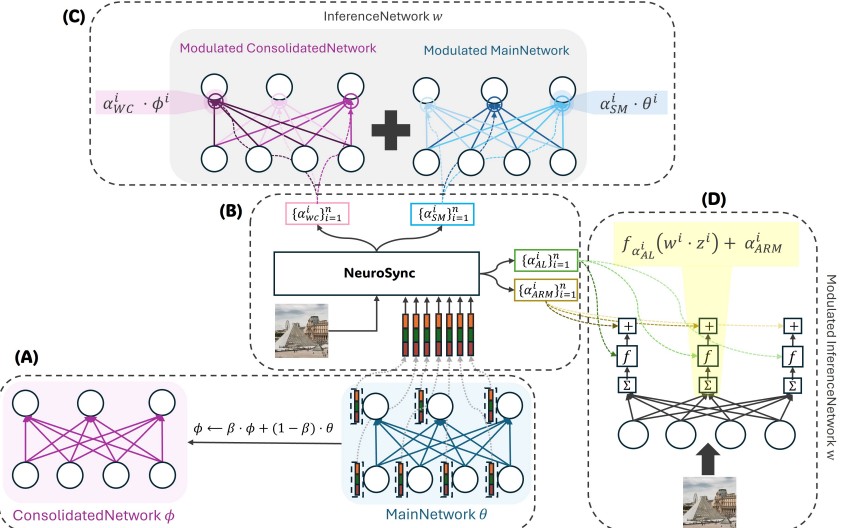

Figure 1: Overview of the architecture and information flow in NeuMoSync. (A) The MainNetwork, parameterized by $\theta$, is shown in the blue box with the blue connections. Each neuron in the MainNetwork is associated with a feature vector. Additionally, the ConsolidatedNetwork (purple box) preserves an exponential moving average of the MainNetwork, denoted as $\phi$. (B) The NeuroSync takes the current sample along with the feature vectors of all neurons as inputs. It produces four modulation coefficients per neuron: $\alpha_{SM}^i$, $\alpha_{WC}^i$, $\alpha_{AL}^i$, and $\alpha_{ARM}^i$ for each neuron $i$. (C) First view of the InferenceNetwork, which illustrates how the synaptic weights ($w$) of the Inference Network are formed. For each neuron $i$, the weights from the Main Network ($\theta^i$) and the Consolidated Network ($\phi^i$) are dynamically combined using the modulation coefficients $\alpha_{SM}^i$ and $\alpha_{WC}^i$. The resulting weighted sum, $w^i = \alpha_{SM}^i \theta^i + \alpha_{WC}^i \phi^i$, defines the effective weights used during inference. (D) Second view of the InferenceNetwork, shows how a neuron's activity is modulated. Given $z_i$ as the input to neuron $i$, the coefficient $\alpha_{AL}^i$ dynamically sets the negative slope of the PReLU activation function ($f$) for this neuron. After the activation is calculated on the weighted input, the coefficient $\alpha_{ARM}^i$ provides an additive offset to the final output.

**Inputs to NeuroSync.** The NeuroSync module receives two types of input: the current data sample and a set of feature vectors, one associated with each neuron in the MainNetwork. These feature vectors provide the module with a summary of each neuron's role and activity history. Specifically, as shown in Figure 1 (blue box of (A)), each vector is composed of three components: (1) a learnable feature vector which captures the neuron's intrinsic proprieties, (2) positional information (e.g., layer and neuron index), and (3) an exponential moving average (EMA) of its past activation statistics to capture its activity over time. The main goal of using these features is to ensure the NeuroSync module has sufficient context during continual learning to generate targeted modulation signals. Moreover, these components also bear a loose conceptual resemblance to biological phenomena, such as the diversity of intrinsic neuronal properties (14; 86) and the monitoring of activation history seen in homeostatic plasticity (64; 58; 79).

**Modulation Outputs and InferenceNetwork Construction.** The NeuroSync module's output is a set of four neuron-specific signals: $\{\alpha_{WC}\}_{i=1}^n, \{\alpha_{SM}\}_{i=1}^n, \{\alpha_{AL}\}_{i=1}^n, \{\alpha_{ARM}\}_{i=1}^n$, which we denote as the $\alpha$-parameters, that orchestrate the assembly of the final InferenceNetwork (Figure 1C). Rather than learning a fixed architecture, the InferenceNetwork is dynamically constructed for each input using these $\alpha$-parameters to modulate the contributions of the MainNetwork and ConsolidatedNetwork. This modulation targets three primary aspects of each neuron's computation: the synaptic weights, the activation function, and the post-activation offset.

**Synaptic Weight Modulation and Consolidation.** The first stage of modulation dynamically constructs the synaptic weights ($w^i$) for each neuron in the InferenceNetwork. This is achieved by computing a weighted combination of the parameters from the MainNetwork ($\theta^i$) and the ConsolidatedNetwork ($\phi^i$), governed by their respective modulation coefficients:

$$w^i = \alpha_{SM}^i \theta^i + \alpha_{WC}^i \phi^i \tag{1}$$

Here, $\alpha^i_{\mathrm{SM}}$ (Synaptic Modulation) and $\alpha^i_{\mathrm{WC}}$ (Weight Consolidation) dynamically gate the influence of the fast-adapting and slow-moving networks. The parameters $\phi$ of the ConsolidatedNetwork serve as a stable, long-term memory, and are updated via an exponential moving average (EMA) of the MainNetwork's weights:

$$\phi \leftarrow \beta\phi + (1 - \beta)\theta \tag{2}$$

The hyperparameter $\beta$ is a consolidation rate, set close to 1, that controls the stability of this long-term memory. The critical impact of this parameter's value on balancing plasticity and adaptability is explored in our sensitivity analysis in Appendix G.3. Combining the MainNetwork and ConsolidatedNetwork in this manner enables input-dependent amplification or attenuation of each networks' contribution within the InferenceNetwork. Consequently, when $\alpha_{\mathrm{WC}}$ is large, or when $\alpha_{\mathrm{SM}}$ has a large magnitude, the contribution of the corresponding network's neurons is amplified; when either coefficient is close to zero, the contribution of that network is effectively suppressed. This strategy of using a slow-moving average network to preserve consolidated knowledge aligns with prior continual learning approaches, which employ dual-memory mechanisms to mimic the interplay between fast and slow learning (2; 66), which is inspired by the fact that the human brain itself possesses two distinct memory modules: the fast-learning hippocampus and the slow-learning neocortex(38).

**Activation Function Modulation ($\alpha_{\mathrm{AL}}$).** The second form of modulation, $\alpha^i_{\mathrm{AL}}$ (Adaptive Linearity), dynamically parameterizes the neuron's activation function. Specifically, for each neuron, it sets the negative slope of a Parametric Rectified Linear Unit (PReLU) (29). The activation output $o^i$ for a given pre-activation value $x^i$ is therefore computed as:

$$o^i = f_{\alpha^i_{\mathrm{AL}}}(x^i) = \begin{cases} x^i, & x^i \geq 0, \\ \alpha^i_{\mathrm{AL}}\, x^i, & x^i < 0, \end{cases} \quad \text{with} \quad x^i = w^i \cdot z^i$$

This dynamic control of neuronal linearity is motivated by two key findings. First, adaptive slopes can improve gradient flow through the network, mitigating issues like shattered gradients (5). Second, and particularly relevant to our setting, adaptive activations have recently been shown to be highly effective for preserving plasticity in continual learning (59). For a comprehensive analysis of the effect of this modulation signal on training stability and gradient flow in our method, please refer to Appendix N.

**Additive Regulation Modulation ($\alpha_{\mathrm{ARM}}$).** Finally, the $\alpha^i_{\mathrm{ARM}}$ coefficient provides a direct additive offset to the neuron's activation ($o^i$), yielding the final output: $\mathtt{output}^i = o^i + \alpha^i_{\mathrm{ARM}}$. This form of additive modulation is consistent with established conditioning mechanisms in deep learning, such as the shifting operation used in (56) to modulate feature activity. Also, in a biological context, this bears a loose, high-level resemblance to global modulatory signals that influence neuronal excitability (55). We provide an empirical analysis of this effect in Appendix L.

**Training** Once the InferenceNetwork is constructed, it is used to process the current input and produce the predictions. Standard backpropagation is used to compute gradients and update both the MainNetwork and the NeuroSync module end-to-end. This allows all components, including modulation parameters, to be learned jointly without requiring meta-learning or task boundaries.

To empirically validate our design, we conduct comprehensive ablation studies that systematically investigate the necessity and influence of each $\alpha$-parameter and the global controller (Sections 3.5 and G). Beyond performance metrics, we also analyze the learned dynamics of the $\alpha$-parameters and analyze their emergent behaviors and patterns in Section 3.6. For a discussion on the biological inspiration behind these components and the distinctions between our model and biological systems, we refer the reader to Appendix I.5. Furthermore, the end-to-end training procedure for the model is detailed in Appendix J.

## 3 EXPERIMENTS

Our empirical evaluation is designed to rigorously assess two fundamental properties of continual learning systems. The first is plasticity preservation: the ability to maintain learning capacity over a long sequence of tasks. The second is fast adaptation, which we dissect into two key components: *forward adaptation*, the speed at which a model learns novel, previously unseen tasks, and *backward adaptation*, the efficiency of re-adapting to previously encountered tasks. We provide

further experiments, including an investigation into the consolidation of recurring knowledge, in Appendix F.1. All reported results are averaged over five independent seeds.

We adopt different network architectures for all the baselines and the MainNetwork depending on the difficulty and scale of the datasets: a two-layer MLP (100,100) for `MNIST`/`CIFAR10`, and a four-layer CNN (8,16,32,64) for `Tiny-ImageNet`/`CIFAR100`. Finally, for the `Shuffle Mini-ImageNet` benchmark used in Section 3.7, we use a ResNet-18 model (30). In all cases except ResNet-18, the controller is a single-layer, encoder-only Transformer with two heads and an embedding dimension chosen from $\{64, 128, 512\}$ so that it constitutes only 5–8% of total parameters. For the ResNet classifier, we instead employ the sparse controller from Section 3.7, which uses only 0.002% of the parameters.

To ensure a fair and robust comparison, we performed an extensive hyperparameter grid search for all baseline methods as well as our own. The optimal configuration for each method on each benchmark was determined by selecting for the highest average final accuracy across three independent random seeds. The complete search spaces and the selected hyperparameters are detailed in Appendix D.

### 3.1 SETTINGS AND EVALUATION METRICS

In this paper, we investigate a continual supervised learning scenario in which an agent sequentially learns $T$ distinct tasks, with $M$ training steps dedicated to each task. Below, we describe the evaluation metrics used to measure plasticity preservation and various facets of adaptation, while a more detailed explanation of metrics is provided in Appendix C.1.

**Plasticity Preservation (Setting** I**).**

To evaluate plasticity, we define the average online task accuracy as the average accuracy over a task as in prior work (37). For a task $k$, we have: $\text{Avg Online Task Accuracy}(k) = \frac{1}{M} \sum_{b=0}^{M} a_b$ where $a_b$ is the accuracy on the $b$-th minibatch of task $k$.

**Fast Adaptation (Setting** II**).** Beyond merely preserving plasticity, the ability to quickly adapt to tasks is also essential for a learner (27). To quantitatively assess the adaptation speed of a continual learner, we utilize the Learning Curve Area (LCA) (10) as a unified metric for both forward and backward adaptation. Specifically, we consider the speed of forward adaptation ($\text{LCA}_F$), how fast the model learns the next task after previous training, and backward adaptation ($\text{LCA}_B$), how fast the final model can readapt tasks previously trained on. These are defined as follows:

$$
\text{LCA}_F = \frac{1}{\gamma + 1} \sum_{b=0}^{\gamma} \left( \frac{1}{T-1} \sum_{k=1}^{T-1} A_{b,k}^{k-1} \right) \quad , \quad \text{LCA}_B = \frac{1}{\gamma + 1} \sum_{b=0}^{\gamma} \left( \frac{1}{T-1} \sum_{k=0}^{T-2} A_{b,k}^{T-1} \right).
$$

where $\gamma$ is the total number of training steps used to evaluate both forward and backward adaptation for each task. $A_{b,j}^i$ denotes the accuracy after training on $i$ tasks in sequence, and then being trained on task $j$ for $b$ batches[1]. In our experiments, we specifically choose $\gamma$ such that it is relatively smaller than $M$ to evaluate an agent adaptation in a low training budget regime.

To analyze adaptation speed in a finer-grained manner, we decompose it into two components: *knowledge transfer*, which quantifies the additional performance a model achieves from leveraging prior experience when trained on a new task for $b$ batches, and *intrinsic learning ability*, which reflects how quickly the same model learns each task from random initialization. We quantify knowledge transfer using two metrics: Backward Knowledge Transfer ($\text{BKT}_b$) and Forward Knowledge Transfer ($\text{FKT}_b$). Each measures the average improvement in performance, relative to training from scratch. Formally, for $T$ tasks and $b$ fine-tuning batches, we define:

$$
\text{FKT}_b = \frac{1}{T-1} \sum_{k=1}^{T-1} \left( A_{b,k}^{k-1} - A_{b,k}^0 \right), \quad \text{BKT}_b = \frac{1}{T-1} \sum_{k=0}^{T-2} \left( A_{b,k}^{T-1} - A_{b,k}^0 \right)
$$

where $A_{k,b}^0$ is the accuracy obtained by training from a random initialization for $b$ batches on task $k$. We also define $\text{LCA}_0$ as a metric for intrinsic learning speed. It is identical to $\text{LCA}_F$, except that the model is reinitialized with random weights before training on each new task. As discussed in

---

[1] In other words, after learning tasks 1 through $i$, we take a snapshot of the model, train that copy for $b$ additional batches on task $j$, and report its train accuracy on task $j$.

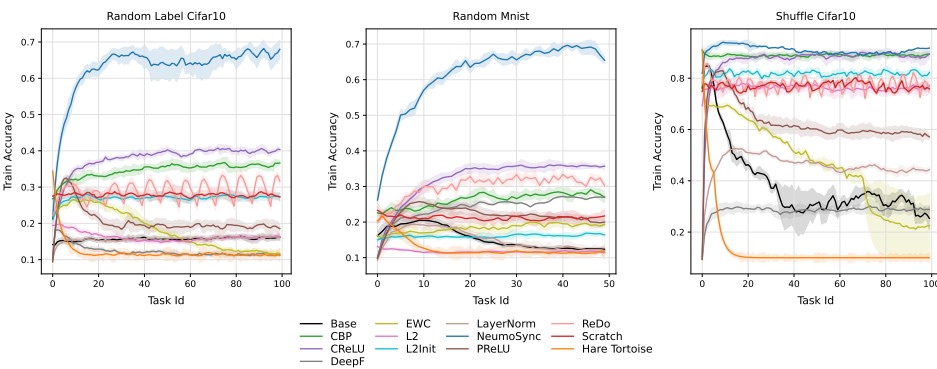

Figure 2: Learning curves for three plasticity-evaluation tasks and different baseline algorithms. NeuMoSync performs best on all three with a large gap on the more difficult random label tasks.

Appendix C.1, we establish a relationship among $\text{LCA}_0$, $\text{BKT}_b$, $\text{FKT}_b$, and the adaptation speeds $\text{LCA}_B$ and $\text{LCA}_F$, which supports the decomposition proposed here.

## 3.2 Baselines and Benchmarks

To compare our method, we consider a wide range of plasticity preservation methods, which can be categorized into three main approaches. For reset-based methods, we consider Continual Back-prop (CBP) (16), ReDo (72), and `Hare & Tortoise` (40). For regularization-based methods we consider L2, `L2Init` (37), and `EWC` (35) (for analyzing a learner that is specifically designed for preserving prior knowledge). Finally, for architecture-based methods, we evaluate Layer Normalization (`Layer Norm`) (47), Concatenated ReLU (CReLU) (1), per-neuron Parametric ReLU (`PReLU`) (59), and `DeepF` (42). Details about the baselines, the hyperparameters of each method, and architectures can be found in Appendix B.2, Appendix C, and Appendix D.1, respectively.

We compare our method with the baselines on three challenging plasticity benchmarks (59; 37; 16): `Random Label MNIST`, `Random Label CIFAR-10`, and `Shuffle CIFAR-10`. The first two require memorizing arbitrary label assignments that change in each task, while Shuffle CIFAR-10 tests adaptation to concept drift by shuffling class-label mappings between tasks without altering the images. We include three additional benchmarks to evaluate the adaptation ability of the top-performing methods identified in setting I: `Class Split T-ImageNet` (77), consisting of sequential binary classification tasks on novel classes; `Class Split CIFAR-100`, which introduces five new classes per task; and `Permuted MNIST`, where each task applies a unique pixel permutation. To increase the challenge, all benchmarks incorporate random data augmentations. Further details on each benchmark's configuration are provided in Appendix C.2. Finally, we evaluate the ResNet-18 on the `Shuffle Mini-ImageNet` benchmark, which is constructed by randomly permuting the class-label mapping of the Mini-ImageNet (83) dataset, as described in Section 3.7.

## 3.3 Setting I (Plasticity Preservation)

As shown in Figure 2, NeuMoSync demonstrates superior plasticity preservation, most notably on the memorization benchmarks. While several plasticity-focused baselines (e.g., `CBP`, `ReDo`) successfully prevent performance degradation, their learning capacity remains low and stagnant. In contrast, NeuMoSync achieves significantly higher absolute performance, suggesting it learns more effectively from each task. This result is particularly striking on tasks with random input-output associations, which are designed to prevent knowledge transfer. Performance gains indicate that NeuMoSync develops an emergent ability to "learning to memorize", that is, it acquires a transferable strategy to rapidly form new associations, a capability it learns without any explicit meta-learning objective.

## 3.4 Setting II (Dissecting the Dynamics of Fast Adaptation)

In this setting, we move beyond plasticity to dissect the dynamics of fast adaptation. We assess overall adaptation speed for novel tasks (forward adaptation, $\text{LCA}_F$) and previously seen tasks (backward adaptation, $\text{LCA}_B$). Crucially, we also disentangle this speed into two underlying components: the model's intrinsic learning speed ($\text{LCA}_0$), measured from a random initialization, and its capacity for knowledge transfer, measured via Forward and Backward Knowledge Transfer (FKT, BKT).

Table 1: Results on all continual learning tasks measuring transfer. $\gamma$ denotes the total number of update steps used for forward and backward adaptation. It is set to 20% of the training budget allocated to each task in the continual learning setting. The relationship stated in Equation C.1 between $LCA_0$, $LCA_F$, $LCA_B$, FKT, and BKT is empirically verifiable based on the results.

| Method | Backward Knowledge Transfer | | | Forward Knowledge Transfer | | | Learning Speed | | | Average |
|---|---|---|---|---|---|---|---|---|---|---|
| | $BKT_{\gamma/2}$ | $BKT_\gamma$ | $BKT_{mean}$ | $FKT_{\gamma/2}$ | $FKT_\gamma$ | $FKT_{mean}$ | $LCA_0$ | $LCA_F$ | $LCA_B$ | Final Accuracy |
| **Random Label CIFAR-10 (Memorization)** | | | | | | | | | | |
| CBP | 0.54 (0.19) | 2.60 (0.53) | 0.77 (0.41) | 2.20 (0.72) | 3.03 (1.24) | 1.60 (1.18) | **12.13 (1.39)** | 14.23 (0.59) | 12.30 (0.92) | 38.21 (0.24) |
| CReLU | 3.56 (0.53) | 5.87 (0.72) | 3.33 (1.01) | 2.80 (0.53) | 4.34 (0.70) | 2.13 (1.15) | 11.87 (1.04) | 14.42 (0.72) | 15.20 (1.09) | 35.03 (0.25) |
| ReDo | 0.70 (0.21) | 0.04 (0.01) | 0.43 (1.21) | 0.93 (0.05) | 0.76 (0.21) | 0.90 (0.20) | 11.87 (1.46) | 12.19 (0.84) | 11.33 (0.59) | 27.14 (0.87) |
| L2Init + EWC | 1.51 (0.05) | 2.65 (1.35) | 1.50 (1.10) | 0.98 (0.21) | 1.75 (0.36) | 0.17 (0.80) | 11.81 (1.47) | 12.58 (0.88) | 13.47 (1.05) | 25.14 (0.32) |
| NeuMoSync | **7.12 (1.33)** | **13.77 (1.12)** | **8.10 (1.36)** | **5.03 (1.08)** | **8.06 (1.20)** | **4.77 (0.55)** | 10.68 (0.73) | **16.06 (0.79)** | **18.90 (0.58)** | **64.74 (1.96)** |
| **Random Label MNIST (Memorization)** | | | | | | | | | | |
| CBP | 0.45 (0.15) | 1.02 (0.05) | 0.50 (0.42) | 1.33 (0.32) | 1.05 (0.18) | 0.70 (0.56) | 10.65 (0.71) | 11.68 (0.77) | 10.73 (1.44) | 36.41 (1.25) |
| CReLU | 2.10 (1.15) | 3.87 (1.11) | 1.80 (0.67) | 2.57 (1.23) | 3.50 (0.66) | 1.47 (0.28) | 10.74 (1.49) | 12.65 (1.14) | 12.53 (1.06) | 32.47 (0.81) |
| ReDo | 0.12 (0.03) | 0.03 (0.26) | 0.33 (1.16) | 0.72 (0.13) | 0.11 (0.05) | 0.16 (0.47) | **10.74 (0.77)** | 10.97 (0.71) | 10.10 (1.44) | 27.24 (0.74) |
| L2Init + EWC | 1.04 (0.44) | 1.04 (0.16) | 0.83 (0.98) | 0.09 (0.05) | 0.08 (0.13) | 0.34 (0.06) | 10.03 (0.76) | 10.55 (0.75) | 11.10 (1.06) | 17.49 (0.12) |
| NeuMoSync | **3.65 (0.76)** | **7.22 (1.08)** | **8.53 (1.40)** | **4.66 (0.90)** | **6.95 (0.72)** | **7.17 (1.50)** | 10.13 (1.01) | **18.81 (0.59)** | **19.87 (0.55)** | **58.67 (1.42)** |
| **Shuffle CIFAR-10 (Concept Drift)** | | | | | | | | | | |
| CBP | 0.21 (0.16) | 0.87 (0.19) | 0.36 (1.19) | 0.67 (0.26) | 0.13 (0.10) | 0.31 (0.05) | **37.17 (1.50)** | 37.72 (1.03) | 38.67 (1.47) | 90.14 (0.47) |
| CReLU | 14.56 (1.36) | **17.24 (0.51)** | 12.10 (1.22) | 6.87 (1.18) | 12.86 (1.04) | 8.22 (0.77) | 31.12 (1.14) | **39.93 (0.61)** | 43.29 (0.93) | 89.41 (0.34) |
| ReDo | 12.02 (0.95) | 13.55 (1.45) | 10.44 (1.38) | 2.34 (0.76) | 3.10 (1.20) | 1.44 (0.13) | 32.17 (1.41) | 33.88 (1.37) | 42.71 (0.80) | 77.36 (0.57) |
| L2Init + EWC | 5.10 (1.14) | 7.67 (1.11) | 4.53 (0.65) | 3.44 (1.26) | 4.50 (1.04) | 2.92 (1.28) | 32.05 (1.03) | 35.46 (0.50) | 36.75 (0.82) | 69.65 (0.24) |
| NeuMoSync | **14.90 (0.52)** | 16.45 (1.43) | **13.24 (1.38)** | 4.33 (1.33) | 12.37 (0.81) | 8.92 (0.56) | 32.02 (1.38) | 40.17 (1.45) | 45.90 (0.59) | **92.84 (0.36)** |
| **Permuted MNIST (Domain Incremental)** | | | | | | | | | | |
| CBP | 17.65 (0.99) | 14.88 (0.57) | 13.71 (1.26) | **14.77 (1.27)** | 18.32 (0.63) | 17.14 (0.98) | 20.50 (1.05) | 38.75 (0.77) | 35.29 (1.37) | 69.21 (0.27) |
| CReLU | 23.53 (0.92) | 20.98 (0.71) | 18.86 (1.04) | 4.93 (1.23) | 7.67 (0.70) | 5.57 (0.81) | | 39.38 (1.15) | 53.44 (0.94) | 69.38 (0.31) |
| ReDo | **25.87 (1.02)** | 26.33 (0.62) | 20.71 (0.72) | 13.87 (0.84) | 14.65 (1.09) | 12.43 (0.73) | 12.12 (0.72) | 25.50 (0.57) | 36.29 (1.13) | 35.71 (0.38) |
| L2Init + EWC | 24.74 (0.73) | 24.21 (1.41) | 19.86 (1.36) | 12.21 (0.57) | 13.33 (0.74) | 12.65 (1.17) | 12.12 (0.71) | 25.52 (0.63) | 35.57 (1.44) | 34.24 (0.42) |
| NeuMoSync | 22.76 (1.07) | **28.65 (0.97)** | **24.57 (1.28)** | 12.87 (1.31) | **19.22 (0.69)** | **17.71 (0.60)** | 30.02 (0.93) | **47.13 (0.92)** | **55.71 (0.97)** | **74.36 (1.36)** |
| **Class Split CIFAR-100 (Class Incremental)** | | | | | | | | | | |
| CBP | **12.04 (1.23)** | 10.65 (1.17) | **13.86 (1.48)** | **8.21 (0.60)** | 5.77 (0.90) | 7.52 (0.84) | 51.09 (1.36) | 59.08 (0.75) | **63.57 (0.69)** | 72.36 (0.67) |
| ReDo | 2.76 (1.50) | 2.87 (1.34) | 2.46 (1.47) | 0.31 (0.19) | 0.21 (0.27) | 0.11 (0.25) | **56.20 (0.99)** | 58.19 (0.71) | 56.06 (0.90) | 72.51 (0.75) |
| L2Init + EWC | 3.45 (0.56) | 3.01 (0.88) | 3.94 (1.49) | 2.00 (0.77) | 3.64 (1.28) | 2.10 (0.96) | 55.77 (0.92) | 57.52 (1.46) | 57.67 (1.50) | 68.24 (0.31) |
| NeuMoSync | 9.87 (1.06) | **19.05 (1.22)** | 11.95 (0.65) | 7.10 (0.80) | **17.44 (1.47)** | **8.19 (1.08)** | 51.69 (1.04) | **60.77 (1.25)** | 62.43 (0.56) | **78.68 (0.91)** |
| **Class Split T-ImageNet (Class Incremental)** | | | | | | | | | | |
| CBP | 4.67 (1.08) | 3.11 (1.34) | 5.27 (1.35) | 2.06 (0.66) | 1.01 (0.48) | 1.88 (0.58) | 70.30 (0.69) | **72.37 (1.10)** | 76.88 (1.18) | 83.21 (0.57) |
| CReLU | 1.06 (0.35) | 5.43 (0.18) | 3.65 (1.34) | **3.64 (0.75)** | **4.43 (1.09)** | 2.54 (1.12) | 69.07 (0.92) | 71.74 (1.08) | 72.76 (0.78) | 84.27 (1.02) |
| ReDo | 5.21 (1.43) | 4.43 (0.70) | 6.88 (1.22) | 2.76 (0.74) | 2.87 (0.90) | 2.19 (1.17) | 69.33 (0.80) | 70.52 (0.82) | 74.69 (1.25) | 83.26 (1.32) |
| L2Init + EWC | 1.04 (0.04) | 1.40 (0.23) | 3.54 (1.50) | 0.66 (0.14) | 0.06 (0.11) | 0.08 (0.32) | **72.48 (0.77)** | 71.56 (1.43) | 75.77 (1.38) | 82.76 (1.27) |
| NeuMoSync | **7.30 (1.38)** | **6.56 (0.87)** | **8.23 (0.66)** | 2.65 (1.33) | 3.57 (1.20) | **3.27 (1.11)** | 67.96 (1.49) | 70.44 (1.15) | **76.08 (0.51)** | **86.37 (1.06)** |

As shown in Table 1, NeuMoSync demonstrates superior overall adaptation, consistently achieving the highest $LCA_F$ and $LCA_B$ scores in most benchmarks under a restricted training budget (in each benchmark, we chose $\gamma$ so that it covers 20% of the overall training budget per task - $M$). This result prompts a critical question: does this advantage stem from an inherently faster learning algorithm, or from a more effective use of prior knowledge?

NeuMoSync does not exhibit the highest intrinsic learning speed; its $LCA_0$ is consistently outperformed by other baselines. Instead, its advantage is almost entirely attributable to superior knowledge transfer, as evidenced by its strong performance in both FKT and BKT. These findings indicate that NeuMoSync's rapid adaptation is not simply a function of a faster intrinsic learning algorithm, but rather an emergent property of its architecture. The model appears to acquire a transferable strategy for knowledge transfer by continually learning generalizable inductive biases, all without relying on explicit meta-learning procedures.

To further validate this conclusion, we conducted additional comparative analyses against meta-learning approaches that are meta-trained on task distributions prior to adaptation (Appendix F.2). Our method's superior adaptation performance, even compared to these explicitly meta-trained approaches, reinforces the effectiveness of our knowledge transfer mechanism. These findings underscore NeuMoSync's ability to acquire transferable inductive biases. Furthermore, while our primary focus was on backward adaptation ($LCA_B$) and measuring re-learning efficiency rather than zero-shot retention, we also address the complementary challenge of stability. While our method enhances adaptability and plasticity, it lacks explicit mechanisms for catastrophic forgetting, a key continual learning challenge. Still, it achieves highly competitive performance on forgetting benchmarks, often surpassing methods designed specifically for stability, when augmented with a standard replay buffer. Finally, we provide comprehensive evaluations of computational costs, run-time, and parameter counts across all methods in Appendixes D.6 and D.8.

### 3.5 Ablation Study

To understand the sources of NeuroSync's performance, in this section, we conduct a series of ablation studies designed to systematically dissect the architecture. Our primary investigation addresses two key questions: (1) How does the performance change when each mechanism is removed? and (2) What are the key inductive biases in the NeuroSync module?

#### 3.5.1 Impact of Modulatory Mechanisms

To assess the contribution of each component, we conduct two types of ablations (Figure 3). First, 'w/o' variants disable a single $\alpha$-parameter mechanism. Second, a 'w/o NeuroSync' variant removes the global controller entirely, forcing each neuron to learn its parameters to isolate the effect of global controller. The most significant performance degradation occurs by removing synaptic modulation ($\alpha_{\text{SM}}$), underscoring its essential role. Adaptive linearity via $\alpha_{\text{AL}}$ also proves vital for maintaining plasticity, a finding that aligns with prior work on the benefits of adaptive activation functions (59). Interestingly, while $\alpha_{\text{WC}}$ is primarily designed to facilitate knowledge transfer, its removal also leads to a notable loss of plasticity. The activation offset ($\alpha_{\text{ARM}}$) has a more subtle effect, reducing overall performance without being the primary driver of plasticity preservation.

Crucially, the sharp drop in performance of the 'w/o NeuroSync' variant highlights the necessity of the global modulation process. This result demonstrates that providing neurons with independent adaptive capabilities is insufficient; the coordinated, context-aware signals generated by the global controller are essential to the success of the architecture. For a more granular investigation, we provide additional ablation studies in the appendix. These include an analysis of each mechanism operating in complete isolation (Appendix G), a study designed to quantify the role of knowledge transfer through the NeuroSync module (Appendix G.2), and a detailed sensitivity analysis of the consolidation rate $\beta$ (Appendix G.3). Finally, Appendix M presents an ablation study examining how the granularity of plasticity modulation signals affects learning performance.

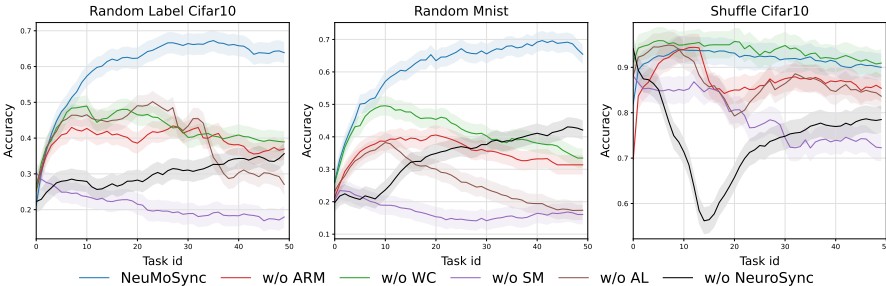

Figure 3: Learning curves for ablations of NeuMoSync. Each curve corresponds to removing one type of $\alpha$-parameters. The black curve corresponds to making all $\alpha$-parameters learnable (i.e., not input-dependent). All ablations but one show degraded performance compared to full NeuMoSync.

#### 3.5.2 Inductive Biases of Neurosync

Our previous analysis highlights the importance of the NeuroSync module's architecture, but a key question remains: which of its inductive biases is responsible for its success? We first establish that the performance gains are not merely due to the use of a powerful attention-based model. As shown in Table 2, a standard Vision Transformer (ViT)(17) baseline, when applied directly to the classification task, fails to preserve plasticity on the challenging `Random-label CIFAR10` benchmark. This result confirms that NeuMoSync's superior performance stems specifically from its function as a global modulator and its interaction with the MainNetwork and ConsolidatedNetwork, rather than the intrinsic power of its underlying components.

We hypothesize that the crucial inductive bias is *parameter sharing*, which equips the model with a single, universal function that can be applied to all neurons. To isolate the effect of parameter sharing from the attention mechanism, we conducted a controlled experiment. We replaced the transformer in the NeuroSync module with two alternatives: (1) a non-attentional, parameter-sharing architecture based on 1D convolutions, and (2) a standard MLP, representing a non-parameter-sharing model.

The results, presented in Table 2, provide support for our hypothesis. The 1D convolutional architecture achieves performance comparable to the transformer-based module, while the MLP-based variant performs significantly worse. This finding confirms that the parameter-sharing inductive bias, not the attention mechanism itself, is the key ingredient for an effective global controller in this context. While both parameter-sharing architectures are effective, we retain the Transformer in our primary model due to its practical advantage of having a parameter count that is independent of the number of neurons in the MainNetwork. Further architectural details and experimental specifics of this ablation study are provided in Appendix D.5.

Table 2: Ablation of the NeuroSync module's architecture to isolate the effect of parameter sharing. We compare a ViT baseline against NeuroSync variants using different controllers (MLP, 1D Conv, Transformer) on the Random-label CIFAR10 benchmark. We report the average performance over the **last** 10%, 25%, 50%, 75%, and 100% of tasks on Random-label CIFAR10.

| Method | 10% | 25% | 50% | 75% | 100% |
|---|---|---|---|---|---|
| ViT | 47.20(1.76) | 48.72(1.98) | 53.61(1.56) | 61.15(1.21) | 60.38(1.54) |
| NeuMoSync (MLP) | 48.47(1.15) | 49.17(1.23) | 49.31(1.54) | 50.26(1.26) | 47.51(1.32) |
| NeuMoSync (1D Conv) | **70.04(1.07)** | 68.11(1.12) | 67.11(1.11) | 65.24(1.32) | 63.34(1.82) |
| NeuMoSync | 69.31(1.10) | **68.32(1.04)** | **68.81(1.03)** | **67.33(1.46)** | **64.74(1.96)** |

### 3.6 EMERGENT BEHAVIOR

We also analyzed the internal dynamics of NeuMoSync's learned modulation signals. Our analysis, visualized in Figure 4, reveals three distinct emergent behaviors which we analyze in the following.

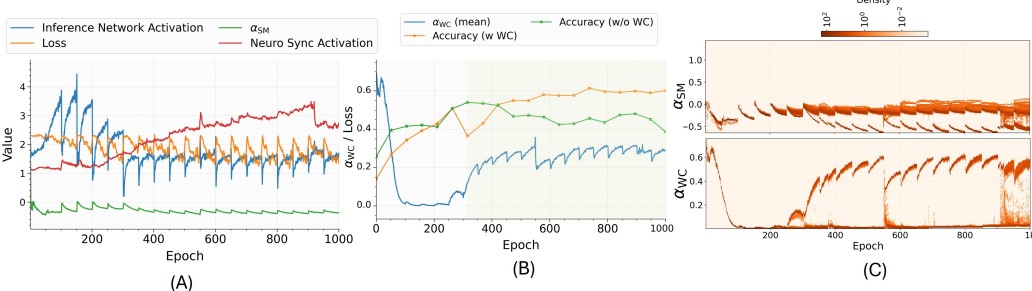

Figure 4: These plots show the analyses of the $\alpha$-parameters and the NeuMoSync module on the `Random Label CIFAR-10` tasks. (A) shows that the absolute value of $\alpha_{\text{SM}}$ decreases at task changes as losses spike up (B) shows that an increase in $\alpha_{\text{WC}}$ co-occurs with an increase in accuracy. (C) shows that two groups of neurons emerge in terms of magnitude of $\alpha_{\text{WC}}$ and $\alpha_{\text{SM}}$.

Figure 4(A) shows the model's immediate reaction to a task switch. At each task transition, the loss spikes, reflecting a prediction error due to the changed data distribution. The NeuroSync module responds instantly: the average synaptic modulation coefficient ($\alpha_{\text{SM}}$) sharply decreases in terms of magnitude, temporarily suppressing the MainNetwork's influence. Simultaneously, the internal activation[2] of the NeuroSync module (shown in red) itself increases, indicating that it is actively processing the novel context. Task switching is also evident in the InferenceNetwork 's activation (the average activation of all neurons in the InferenceNetwork in blue). At the onset of each new task, the activation drops before gradually increasing for the remainder of the task.

Beyond its immediate reactions, the analysis reveals an evolving long-term strategy (Figure 4B). For the initial phase of training, the model operates in a fully plastic mode, with the weight consolidation coefficient ($\alpha_{\text{WC}}$) held at zero. However, a strategic shift occurs later in training (around epoch 300, shaded), precisely when a slight dip in accuracy in Figure 4C suggests the onset of plasticity stress (a decrease in task accuracy). At this point, the NeuroSync module begins to gate in the influence of the stable, ConsolidatedNetwork by increasing $\alpha_{\text{WC}}$. This learned reliance on consolidated knowledge

---

[2]In this part, the NeuroSync module is implemented as a one-layer encoder-only Transformer. Here we use the average activations of the neurons in the last layer of the MLP in the Transformer, averaged over a batch of samples .

coincides with a recovery in performance, acting as an adaptive mechanism to counteract the decay of plasticity.

Finally, Figure 4(C) reveals that the modulation signals lead to functional specialization at the neuron level. The heatmaps show the distribution of $\alpha_{\mathrm{WC}}$ and $\alpha_{\mathrm{SM}}$ values across the entire neuron population over time. For both coefficients, a clear bimodal distribution emerges: a subset of neurons remain highly plastic (large absolute values of $\alpha_{\mathrm{SM}}$), while others have $\alpha_{\mathrm{SM}}$ values near zero, indicating reduced plasticity. At the same time, one population of neurons shows strong reliance on ConsolidatedNetwork, whereas others make minimal use of it, with $\alpha_{\mathrm{WC}}$ values close to zero. This emergent division of labor, where some neurons stabilize to preserve old knowledge while others remain flexible to acquire new information, provides a mechanism for plasticity and adaptability.

Another particularly interesting observation is that, for a subset of neurons, $\alpha_{\mathrm{SM}}$ and $\alpha_{\mathrm{WC}}$ take on opposite signs (e.g., $\alpha_{\mathrm{WC}} > 0$ and $\alpha_{\mathrm{SM}} < 0$). To carefully analyze the interplay between MainNetwork, ConsolidatedNetwork and the modulation parameters, and to explore a possible explanation for the underlying reason behind the superior performance on this task, we provide an in-depth analysis in Appendix P.

### 3.7 SCALING NeuMoSync

So far, we have implemented the NeuroSync module with either an encoder-only Transformer or a stack of 1D convolutions over all neurons. However, the Transformer's quadratic cost in neuron count and the CNN's parameter scaling with classifier size make both approaches impractical for larger models. To address this, we propose a more scalable encoder–decoder controller that samples a small fixed subset ($K \ll 1$) of neurons across all layers for each input, feeds their features into the same two-head Transformer encoder as before, and uses a lightweight cross-attention module to map encoder outputs to full neuron features for per-neuron modulation. Further details are in Section K.1.

We evaluate this architecture on the `Shuffle Mini-ImageNet` benchmark with $K = 0.1$, and employ a Resnet18 classifier, which is substantially larger than the models used in the previous sections. The results are shown in Figure 5. These results show that the modulation controller adds only a small overhead in terms of additional parameters and a modest runtime cost (as discussed in Appendix K.1), yet yields a substantial performance improvement over baselines such as `ViT`, whose computational requirements are considerably higher than ours.

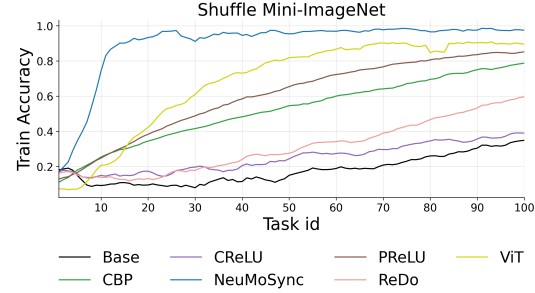

Figure 5: Training accuracy on `Shuffle Mini-ImageNet` for our method, NeuMoSync, and the baselines, using a Resnet18 classifier.

While none of the baselines show explicit plasticity loss, likely due to the classifier's high capacity, the learning curves reveal a clear gap between our global modulatory network and other plasticity-preserving methods.

## 4 DISCUSSION

In this work, we introduce NeuMoSync, a global coordinator architecture designed to maintain plasticity and enhance adaptability in continual learning. NeuMoSync integrates global modulation mechanisms that dynamically control plasticity and activity based on both inputs and the network state. Across a diverse set of benchmarks, our method shows superior performance in preserving plasticity and rapid adaptation. While NeuMoSync presents a promising step toward biologically inspired continual learning systems, scaling it to larger models may require approximations such as grouped neuron processing. Nonetheless, the core ideas of NeuMoSync open new avenues for global modulatory systems in the continual learning paradigm.

## 5 REPRODUCIBILITY STATEMENT

To ensure the reproducibility of our work, we have included our full source code as supplementary material. We commit to releasing the code publicly upon acceptance of the paper. All experimental

details are provided in the appendix: model architectures are described in Appendix D.1, and the complete set of hyperparameters for both main and ablation studies is listed in Appendix D. Furthermore, a detailed description of the benchmarks is available in Appendix C.3, and the precise mathematical definitions of our evaluation metrics are provided in Appendix C.1.

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

# Contents

## A  LARGE LANGUAGE MODEL USAGE

Throughout the preparation of this manuscript, we utilized public large language models as a general-purpose writing assistant. Their role was strictly limited to refining the language and grammar of the manuscript. The LLM was not used for core research ideation, experimental design, implementation, or the generation of novel scientific results.

## B  RELATED WORK

### B.1  KNOWLEDGE TRANSFER IN CONTINUAL LEARNING

Continual learning (CL), also known as lifelong learning, refers to scenarios where data arrives in a sequence and its distribution evolves over time, violating the standard i.i.d. assumption. In such settings, two primary challenges addressed in this paper are the loss of plasticity, where models gradually lose the ability to adapt to new information, and slow adaptation to changing data distributions

**Loss of Plasticity**   As training progresses, a model's ability to learn new tasks often declines. This phenomenon, known as loss of plasticity, is attributed to increasing dormant neurons (72), increased loss of curvature (48), and saturated neurons (47; 59). Methods to address this include resetting parts of the network to restore learning capacity (16; 72; 3; 21), modifying activation functions to maintain gradient flow (42; 1; 59), applying regularization terms that preserve plasticity (37). Some approaches use architectural adaptations like layer normalization (47) or dynamic expansion of the network (44). More recently, (70) proposed a new version of shrink and perturb algorithm in which the perturbation is informed by the direction of gradient for each weight. In this study, we provide an architectural solution to the problem of plasticity loss while keeping the overhead in terms of number of learnable parameters negligible (as described in Appendix D.7).

**Slow Adaptation**   Adaptation speed, often associated with forward transfer, refers to a model's ability to leverage prior knowledge to quickly adapt to new tasks (27). This is closely related to generalization, which involves transferring knowledge to unseen data (85). Strong generalization enhances forward transfer and enables efficient learning in scenarios with limited time or data. While a few works have explored fast adaptation in online continual learning settings relying on pretraining (28) or foundation models (88), most approaches in this area incorporate some form of meta-learning, resulting in hybrid frameworks such as continual meta-learning or meta-continual learning (12; 33; 84; 9). For example, (9) propose using pretrained meta-parameters within a continual setting and updating them in response to out-of-distribution tasks. A comprehensive review of these approaches can be found in (73).

Unlike these methods, which rely on meta-learning either during training (e.g., meta-pretraining) or through specialized learning rules (e.g., meta-gradients or their approximations), our method aims to learn useful inductive biases, specifically, for controlling plasticity and activity, without such overhead.

### B.2  NEURO-INSPIRED AI

The earliest models of artificial neural networks were inspired by the functionality of individual neurons (50; 63). Another major biological inspiration, the organization of neurons in the visual cortex, led to the development of convolutional neural networks (CNNs) (39). In recent years, a wide range of neuro-inspired methods have been proposed, which can be broadly categorized into architectural designs (61; 15; 80; 13), learning rules (43; 45), and loss functions (35). For comprehensive reviews on neuro-inspired AI, see (60; 67). More recently, (44) propose a solution based on neuroscience observations to preserve plasticity which include dormant neuron pruning, weight consolidation, and neural expansion, and tested the method on continual RL tasks.

A number of methods have tried to simulate neuromodulatory effects within their approach (13; 82; 51; 52; 45; 78). Specifically, (13) introduces a new term into the governing ODEs of recurrent neural networks (RNNs), which is modeled with a separate network, to simulate neuromodulatory influences by applying synaptic scaling in a low-rank RNN. Additionally, (82) demonstrates that

controlling neuron activation functions with parameters inferred from a trajectory-aware RNN improves performance in meta-reinforcement learning settings. In (51), networks are trained using a Hebbian learning rule, with the plasticity parameters themselves learned via gradient descent, whereas (52) employs evolutionary algorithms for learning these parameters.

While prior methods primarily rely on dynamical systems induced by recurrent networks or enforce specific learning rules like Hebbian plasticity, our approach differs in that all parameters are trained end-to-end with standard gradient descent, making it broadly applicable across different neural network architectures. To the best of our knowledge, our approach is the first to simultaneously modulate both the functionality and plasticity of each neuron based on the input and the internal state of the network.

Another related line of work draws inspiration from the Complementary Learning Systems (CLS) theory, a neuroscience framework that explains how biological systems, particularly the brain, balance rapid learning and long-term stability (75). Based on this theory, (40) propose an algorithm that maintains a slowly changing version of the network and periodically reinitializes a fast learner with the weights of the slow learner, thereby preserving plasticity. Similarly, (66; 2) leverage a multi-level memory system to prevent forgetting of recurring knowledge across tasks.

While our approach also maintains a slow-changing version of the MainNetwork, it serves a different purpose: rather than implementing a multi-level memory system akin to the division of roles between the neocortex and hippocampus, our method uses the slow learner to simulate the neuromodulatory process responsible for forming consolidated synaptic weights.

### B.3 HYPERNETWORKS

Hypernetwork methods refer to a class of neural network architectures in which one network (the hypernetwork) generates the weights or parameters of another network (the target network). This approach introduces an additional level of abstraction, enabling greater flexibility, adaptability, and efficiency in learning (26). For example, (13) propose using a recurrent neural network (RNN) as a hypernetwork to generate low-rank weights for a target RNN, while others condition the hypernetwork on noise (36) or task-specific inputs (76). A comprehensive review of hypernetwork methods can be found in (11). While these methods use a hypernetwork to generate the full parameters of a target network, NeuroSync in our method generates modulation signals rather than the parameters themselves. These signals dynamically control how the neurons in the MainNetwork activate or adapt, while keeping the MainNetwork architecture fixed across tasks.

As a related prior work, FiLM layers (56) modulate intermediate activations via feature-wise affine transformations, i.e. per-channel scaling and shifting of feature maps based on a conditioning input. In contrast, our method applies two distinct scaling signals, $\alpha_{\text{SM}}$, and $\alpha_{\text{WC}}$ directly to the neuron inputs / effective weights of two networks (MainNetwork and ConsolidatedNetwork). Because this modulation happens before the nonlinearity and at the level of the effective weights, it can be viewed as defining an input-dependent "effective network" with potentially richer changes than per-feature affine re-weighting of activations at each layer.

## C EXPERIMENTAL SETTINGS

### C.1 METRICS

**Fast Adaptation (Setting** II). Beyond preserving plasticity, a critical measure of a continual learner's efficacy is its speed of adaptation (27). We assess this in two primary directions: *forward adaptation*, which quantifies how quickly a model learns novel, unseen tasks, and *backward adaptation*, which measures how efficiently the final model can re-adapt to previously encountered tasks.

To formalize these concepts, consider a sequence of $T$ tasks indexed $k \in \{0, 1, \ldots, T-1\}$. We define $A_{b,k}^i$ as the training accuracy on task $k$ after the model has been trained on it for exactly $b$ optimization steps, given that the model was previously trained sequentially on tasks $0$ through $i-1$. The special case $A_{b,k}^0$ therefore represents the accuracy on task $k$ when trained from a random initialization.

To quantify these adaptation speeds, we employ the Learning Curve Area (LCA) (10), a unified metric that integrates performance over a fixed adaptation budget of $\gamma$ steps. The forward adaptation speed, $\text{LCA}_F$, is the area under the average learning curve for novel tasks. The term inside the parentheses of its formula constitutes the average accuracy at a specific training step $b$ across all tasks $k$, each learned immediately after its predecessors. The outer summation then computes the area under this average curve:

$$\text{LCA}_F = \frac{1}{\gamma + 1} \sum_{b=0}^{\gamma} \left( \frac{1}{T-1} \sum_{k=1}^{T-1} A_{b,k}^{k-1} \right) \quad , \quad \text{LCA}_B = \frac{1}{\gamma + 1} \sum_{b=0}^{\gamma} \left( \frac{1}{T-1} \sum_{k=0}^{T-2} A_{b,k}^{T-1} \right).$$

where $\gamma$ denotes the total number of training steps used to probe both forward and backward adaptation for each task. Throughout the experiments, we set $\gamma$ to be substantially smaller than $M$ to assess adaptation under a low training-budget regime.

In the forward setting, the expression inside the parentheses reflects the model's mean accuracy while it learns each new task for $b$ batches in sequence. In the backward setting, it indicates the mean accuracy when, after finishing all tasks, the model revisits and relearns any earlier task for $b$ batches. We then calculate the LCA of these values across all batches up to $\gamma$ as an indicator of how quickly the model adapts.

While the LCA metrics provide a valuable measure of overall adaptation speed, they conflate two distinct phenomena: a model's intrinsic learning ability and the performance gains derived from knowledge transfer. A high LCA score could indicate either an inherently fast learner or a model that is good at leveraging past experience. To properly attribute performance gains and understand the true source of a model's adaptability, it is necessary to disentangle these factors.

We achieve this decomposition by first isolating and quantifying the knowledge transfer component. We define metrics that measure the direct performance benefit of prior experience by comparing it against a baseline: the same model trained from a random initialization. Formally, we define Forward Knowledge Transfer ($\text{FKT}_b$) and Backward Knowledge Transfer ($\text{BKT}_b$) at a specific training step $b$ as the average accuracy gain over this baseline:

$$\text{FKT}_b = \frac{1}{T-1} \sum_{k=1}^{T-1} \left( A_{b,k}^{k-1} - A_{b,k}^0 \right), \quad \text{BKT}_b = \frac{1}{T-1} \sum_{k=0}^{T-2} \left( A_{b,k}^{T-1} - A_{b,k}^0 \right)$$

Here, the crucial term $A_{b,k}^0$ represents the baseline accuracy on task $k$ after $b$ training steps from scratch, serving as a direct measure of the model's performance without any accumulated knowledge.

Complementing this, we define the model's intrinsic learning ability, $\text{LCA}_0$. This metric is designed to quantify the learning speed inherent to the model's architecture and optimizer, free from the influence of knowledge transfer. It is computed identically to $\text{LCA}_F$, with the modification that the model is reinitialized with random weights before training on each new task. Together, these metrics allow for a principled decomposition of adaptation speed into its constituent parts.

Building on the definitions, we establish the following relationships among $\text{LCA}_0$, $\text{BKT}_b$, $\text{FKT}_b$, and the fast adaptation metrics $\text{LCA}_B$ and $\text{LCA}_F$. These equations demonstrate that both forward and backward adaptation performance can be mathematically decomposed into two distinct components: the model's inherent learning ability, and the respective contribution from knowledge transfer, which can be evaluated separately.

$$\boxed{\text{LCA}_B = \text{LCA}_0 + \frac{1}{\gamma + 1} \sum_{b=0}^{\gamma} \text{BKT}_b, \quad \text{LCA}_F = \text{LCA}_0 + \frac{1}{\gamma + 1} \sum_{b=0}^{\gamma} \text{FKT}_b.}$$

These equations precisely partition adaptation speed into two distinct and interpretable components. The first term, $\text{LCA}_0$, represents the portion of performance attributable solely to the model's inherent learning architecture and optimization. The second term represents the **average knowledge transfer gain**. Since $\text{FKT}_b$ (or $\text{BKT}_b$) quantifies the performance benefit from prior knowledge after precisely $b$ training steps, the summation and normalization by $\frac{1}{\gamma+1}$ yields the total knowledge

transfer benefit, averaged across the entire adaptation budget $\gamma$. This framework provides a principled way to disentangle a model's raw learning capacity from its ability to effectively leverage past experience. We will prove the mentioned relationships below.

*Proof.* Starting with the definition of FKT:

$$\frac{1}{\gamma+1}\sum_{b=0}^{\gamma}\text{FKT}_b = \frac{1}{\gamma+1}\sum_{b=0}^{\gamma}\left(\frac{1}{T-1}\sum_{k=1}^{T-1}(A_{b,k}^{k-1}-A_{b,k}^0)\right) \tag{3}$$

$$= \frac{1}{\gamma+1}\sum_{b=0}^{\gamma}\left(\frac{1}{T-1}\sum_{k=1}^{T-1}A_{b,k}^{k-1} - \frac{1}{T-1}\sum_{k=1}^{T-1}A_{b,k}^0\right) \tag{4}$$

$$= \frac{1}{\gamma+1}\sum_{b=0}^{\gamma}\left(\frac{1}{T-1}\sum_{k=1}^{T-1}A_{b,k}^{k-1}\right) - \frac{1}{\gamma+1}\sum_{b=0}^{\gamma}\left(\frac{1}{T-1}\sum_{k=1}^{T-1}A_{b,k}^0\right) \tag{5}$$

$$= \text{LCA}_F - \text{LCA}_0 \tag{6}$$

which complete the proof.

Following the same algebraic steps used for the forward-transfer case,

$$\frac{1}{\gamma+1}\sum_{b=0}^{\gamma}\text{BKT}_b = \frac{1}{\gamma+1}\sum_{b=0}^{\gamma}\left(\frac{1}{T-1}\sum_{k=0}^{T-2}(A_{b,k}^{T-1}-A_{b,k}^0)\right) \tag{7}$$

$$= \frac{1}{\gamma+1}\sum_{b=0}^{\gamma}\left(\frac{1}{T-1}\sum_{k=0}^{T-2}A_{b,k}^{T-1}\right) - \frac{1}{\gamma+1}\sum_{b=0}^{\gamma}\left(\frac{1}{T-1}\sum_{k=0}^{T-2}A_{b,k}^0\right) \tag{8}$$

$$= \text{LCA}_B - \text{LCA}_0. \tag{9}$$

which complete the proof. $\square$

Note that the definition of $\text{LCA}_0$ (Learning from Scratch Adaptation speed) is slightly distinct between the two proofs. In our context, we define $\text{LCA}_0$ as the speed for learning from scratch, averaged over $T-1$ tasks. This metric is crucial for understanding the baseline learning efficiency without the influence of prior task knowledge.

### C.2 BASELINES

We give a brief description of all the baselines algorithms used.

**Scratch.** This is the network reinitialized before each task to be trained from scratch. This gives a basic baseline performance where no knowledge accumulation is possible.

**Base.** This is a vanilla network trained on all the tasks in a standard manner, with no special considerations taken. This represents the naive baseline of learning as if it was in default supervised learning setting, ignoring continual learning considerations.

**Parameterized ReLU (PReLU).** This baseline replaces the ReLU activation with PReLU (29), which has a learnable slope parameter per neuron. This has been shown to improve plasticity in certain settings (59).

**Concatenated ReLU (CReLU).** This baseline replaces the ReLU activation with CReLU (69), which produces two outputs in each neuron with opposite signs. This activation function has been found to help mitigate dead units and thereby improve plasticity (1).

**Deep Fourier Features (DeepF).** This baseline uses Fourier features as the activation function (42). This may introduce more linearity in the network, improving the gradient flow and plasticity.

**Continual Backprop (CBP).** (16) propose CBP as a method to reset the weights of neurons that are no longer useful. By doing so, it can help the network preserve plasticity throughout training.

**ReDo.** (72) propose ReDo as a mechanism to periodically reset the weights of dormant or dead ReLU units with the aim of refreshing plasticity.

**Hare & Tortoise** (40) proposes two networks. The first is updated via gradient descent, while the second is updated using an exponential moving average (EMA) at each step. Every $k$ steps, the weights of the first network are replaced with those of the second, and training continues.

**LayerNorm.** LayerNorm (4) is a common normalization layer in deep neural networks. It has been found to assist with plasticity-preservation in some settings (47).

**L2.** This baseline uses $\ell_2$ regularization on the weights, which may help weights stay in a region amenable to further learning.

**L2Init.** (37) refine $\ell_2$ regularization by pulling the parameters towards their initial values instead of towards zero, where further learning can be succesful.

**Elastic Weight Consolidation (EWC).** EWC (35) aims to mitigate catastrophic forgetting by adding regularization towards the final weights learned on previous tasks. By adding this soft constraint, the network may stay closer to previous good set of weights when learning on later tasks.

**L2Init + EWC.** Although (EWC) is designed to mitigate catastrophic forgetting, our analysis in Figure 2 reveals that it exhibits significant plasticity loss. To ensure our comparative evaluation of adaptation speed included a method primarily focused on preventing forgetting, we strategically augmented EWC with L2init, similar to (37). This addition aims to help preserve plasticity within the EWC framework, thereby offering a more robust baseline for assessing adaptation in continual learning scenarios.

**Model-agnostic Meta Learning (MAML).** (18) propose MAML as a meta-learning method to learn a good initialization point for fast adaptation. This method requires a meta-training phase preceding usual training, where the algorithm can sample tasks from the task distribution.

## C.3 BENCHMARKS

Our experimental framework includes six benchmarks that encompass various types of non-stationarity, including memorization, concept drift, class-incremental, and domain-incremental settings. Additionally, to increase the difficulty of all benchmarks, we apply random data augmentation to every sample. Specifically, each sample undergoes random cropping, flipping, and rotation, with the augmentations applied differently for each task.

- **Permuted MNIST.** In this setting, we sample 10,000 images from the MNIST dataset and apply a distinct, fixed random permutation to the pixel order for each task. This transformation disrupts the spatial structure of the original images, creating a different input distribution per task. While the digit labels remain the same, the altered input representations require the model to relearn the association between input and label for every task. This benchmark evaluates the model's ability to adapt to substantial changes in input structure while maintaining classification performance. The experiment is conducted over 50 tasks, with each task consisting of a single pass through the data (1 epoch) using a batch size of 64, further increasing the challenge by restricting the model's exposure to each permuted dataset.

- **Random Label MNIST.** In this benchmark, each task involves a random selection of 1,200 images from the MNIST dataset, with the original labels replaced by entirely random ones. The true digit identities are ignored, forcing the model to memorize arbitrary mappings between images and their assigned labels. As a result, the task lacks any inherent structure, emphasizing pure memorization rather than generalization. This setup is used to evaluate how learning unstructured, noisy targets influences the network's plasticity, particularly its capacity to adapt to new tasks after repeated exposure to such random associations. The experiment consists of 50 tasks, with each task trained for 200 epochs using a batch size of 16 to ensure sufficient memorization pressure.

- **Random Label CIFAR-10.** This benchmark follows the same approach as Random Label MNIST but uses 1,200 images randomly sampled from the CIFAR-10 dataset. Each image is assigned a randomly chosen label from the ten available classes, eliminating any semantic correspondence between inputs and targets. Due to the increased visual complexity and color variation in CIFAR-10 compared to MNIST, the task places greater demands on the network's memorization abilities. It requires the model to encode intricate visual patterns without relying on meaningful label structure. This setup enables us to explore how memorizing label noise impacts plasticity, particularly in settings with higher-dimensional inputs. The model is trained on 100 such tasks, each for 50 epochs using a batch size of 16, ensuring sufficient exposure to the randomized associations and allowing us to investigate their cumulative effect on the model's ability to adapt over time.

- **Shuffle CIFAR-10.** For this benchmark, each task consists of 5,000 CIFAR-10 images, where the label assignments are randomly permuted across the classes at the beginning of each task. Unlike the random-label setup, this permutation is consistent across samples in a task, meaning all images from the same original class are assigned the same new label. This generates a structured but shifted label mapping, introducing a mild form of distribution shift without altering the image content itself. The objective is to investigate how the network responds to systematic changes in label assignments and how such shifts influence its capacity to adapt.

  This scenario mimics an online learning environment, where the evolving class-to-label mapping requires continual parameter updates. We conduct this experiment over 100 tasks, training the model for 20 epochs per task using a batch size of 16. This setup allows us to observe how repeated exposure to modest, structured shifts affects the network's plasticity and ability to retain useful representations across tasks.

- **Class-Split CIFAR-100.** This benchmark is constructed from the CIFAR-100 dataset to simulate a practical continual learning scenario. In each task, the model encounters 5 previously unseen classes, with 500 samples per class, totaling 2,500 image–label pairs. Training is conducted over 20 epochs per task using a batch size of 16, continuing sequentially until all 100 classes have been introduced. This setup presents the model with a series of distributional shifts, requiring it to continually adapt to novel class distributions. As such, it offers a challenging setting for evaluating a model's ability to maintain plasticity and avoid degradation in performance as new classes are learned.

- **Class-Split T-ImageNet** In this benchmark, a new binary classification task is introduced at each step, utilizing two previously unseen Tiny ImageNet classes, with 600 images allocated for each class. This setup specifically assesses how well a continual learning agent can learn and generalize to novel categories over time, simulating a scenario where new classes are continually encountered (37). For this benchmark, the model is trained for 20 epochs at each step.

## D HYPERPARAMETER AND TRAINING SETUP

### D.1 ARCHITECTURE

Similar to previous work (37; 21), we adopt two types of network architectures in our experiments for all baselines: a multi-layer perceptron (MLP) and a convolutional neural network (CNN). We also use these architectures as the MainNetwork in our model. To be able to analyze the behavior of the network on the neuron level, we use the same MLP for the Permuted MNIST, Random Label MNIST, Shuffle CIFAR-10, and Random Label CIFAR-10 tasks, and reserve the CNN for the Class Split CIFAR-100 and Class Split T-ImageNet tasks, where the input complexity necessitates a deeper architecture. Additionally, in all experiments, except those in Section 3.5, we used an encoder-only single-layer Transformer with two attention heads as the NeuroSync module. The embedding size varies across tasks to ensure that the number of parameters in this module introduces minimal overhead, as discussed in Section D.7. When using a Transformer as the NeuroSync module, we patch and tokenize the image following the same principle as in (17).

**MLP.** We use a two-layer MLP with 100 hidden units in each layer, followed by ReLU activations. The output layer contains 10 units. This setup offers enough capacity to analyze plasticity loss and neuron dynamics across different types of non-stationarity, while remaining computationally efficient.

**CNN.** For the Class-Split CIFAR-100 and Class-Split T-ImageNet tasks, we use a four-layer convolutional neural network with channel sizes of 8, 16, 32, and 64, each followed by a max-pooling operation. The convolutional layers are followed by a fully connected layer that maps the flattened output to 100 units for CIFAR-100 and 200 units for T-ImageNet. This architecture is designed to handle the increased visual complexity of the datasets while enabling us to investigate the impact of plasticity loss in more realistic, large-scale continual learning scenarios.

### D.2 Shared Hyperparameters

The shared hyper parameters between all methods for each task can be found in Table 3. $\gamma$ denotes the number of training steps used to evaluate the behavior of algorithms in forward and backward adaptation under a low-training budget regime. It is important to note that $\gamma$ refers specifically to the initial training steps during which the model is evaluated on data from a new or previously encountered task. After these $\gamma$ steps, training continues until the task reaches the designated number of epochs. For experiments in Section 3.4, $\gamma$ is set to 20% of the total training steps allocated to each task.

Table 3: Shared hyperparameters among algorithms for each benchmark.

| Benchmark | Epochs | Batch Size | # Samples per Task | $\gamma$ |
|---|---|---|---|---|
| Permuted MNIST | 1 | 64 | 10,000 | 156 |
| Random-Label MNIST | 200 | 16 | 1,200 | 750 |
| Random-Label CIFAR-10 | 50 | 16 | 1,200 | 187 |
| Shuffle CIFAR-10 | 20 | 16 | 5,000 | 312 |
| Class Split CIFAR-100 | 20 | 16 | 2,500 | 156 |
| Class Split T-ImageNet | 20 | 16 | 1,200 | 750 |

### D.3 Method-specific Hyperparameters

Several of the baseline methods introduce additional hyperparameters, which we explain in Table 4.

Table 4: Additional hyperparameters introduced by each method.

| Method | Hyperparameters |
|---|---|
| L2, L2Init | Regularization weight $\lambda$ |
| EWC | Fisher-based regularization weight $\Gamma$ |
| CBP | Replacement rate $r$ |
| ReDo | Recycling period $s$, Recycling threshold $\rho$ |
| NeuMoSync | NeuroSync embedding size $e$ |

For our method (NeuMoSync), we fix the consolidation rate and the dimensionality of each neuron's learnable feature vector to 0.999 and 4, respectively, across all tasks. The optimizer type and learning rate are selected individually for each task. The embedding size of NeuMoSync is configured to ensure minimal overhead: the additional components introduced by NeuMoSync increase the total number of parameters by only 5% to 8% relative to the overall resulting network.

All hyperparameters listed in the following table were selected through grid search, using average final accuracy as the evaluation criterion. To ensure robust selection, each algorithm was trained with three different random seeds, and the mean of the evaluation metric (average final accuracy) was used to determine the best hyperparameter configuration for each method on each benchmark. The search space included: $\lambda \in \{1e{-}2, 1e{-}3, 1e{-}4, 1e{-}5\}$, $\Gamma \in \{10, 1, 1e{-}1, 1e{-}2\}$,

$r \in \{1e{-}1, 1e{-}2, 1e{-}3, 1e{-}4, 1e{-}5, 1e{-}6\}$, recycling frequency ($s$), which occurs every $\{1, 2, 5\}$ tasks, $\rho \in \{0.0, 0.1, 0.01\}$, optimizer $\in \{\texttt{adam}, \texttt{sgd}\}$, and lr $\in \{1e{-}2, 1e{-}3\}$.

For the Hare & Tortoise method, we initialized training with a warm-start phase using 20% of the dataset and the true labels (i.e., no label noise was introduced during this stage). Following the recommendation of the original article, we employed AdamW (46) as the optimizer for the warm-start and continual learning phases. The learning rate was tuned over the set {1e-2, 1e-3, 1e-4}, with 1e-3 consistently emerging as the optimal value across all tasks. The only hyperparameters that varied between benchmarks were (i) the resetting period $s$ (measured in training steps), selected from $\{3000, 4000, 6000\}$, and (ii) the momentum parameter of EMA, $\beta$, selected from $\{0.995, 0.999\}$. The chosen values for each benchmark are reported separately below.

Table 5: Optimal Hyperparameters on **Random Label MNIST**

| Method | Optimal Hyperparameters |
|---|---|
| Baseline | optimizer = sgd, lr = 1e−2 |
| Scratch | optimizer = sgd, lr = 1e−2 |
| PReLU | optimizer = sgd, lr = 1e−2 |
| DeepF | optimizer = sgd, lr = 1e−2 |
| CReLU | optimizer = adam, lr = 1e−3 |
| L2 | optimizer = sgd, lr = 1e−2, $\lambda = 1e-3$ |
| L2Init | optimizer = sgd, lr = 1e−2, $\lambda = 1e-3$ |
| ReDo | optimizer = adam, lr = 1e−3, $s = 30,000$, $\rho = 0.1$ |
| CBP | optimizer = adam, lr = 1e−3, $r = 1e-3$ |
| EWC | optimizer = sgd, lr = 1e−2, $\Gamma = 10$ |
| L2Init + EWC | optimizer = sgd, lr = 1e−2 $\Gamma = 10$, $\lambda = 1e-3$ |
| LayerNorm | optimizer = sgd, lr = 1e−2 |
| Hare & Tortoise | s = 4000, $\beta$=0.999 |
| NeuMoSync | optimizer = adam, lr = 1e−3, $e = 128$ |

Table 6: Optimal Hyperparameters on **Permuted MNIST**

| Method | Optimal Hyperparameters |
|---|---|
| Baseline | optimizer = adam, lr = 1e−3 |
| Scratch | optimizer = adam, lr = 1e−3 |
| PReLU | optimizer = sgd, lr = 1e−2 |
| DeepF | optimizer = sgd, lr = 1e−2 |
| CReLU | optimizer = adam, lr = 1e−3 |
| L2 | optimizer = sgd, lr = 1e−2, $\lambda = 1e-3$ |
| L2Init | optimizer = sgd, lr = 1e−2, $\lambda = 1e-3$ |
| ReDo | optimizer = sgd, lr = 1e−2, $s = 625$, $\rho = 0.0$ |
| CBP | optimizer = adam, lr = 1e−3, $r = 1e-3$ |
| EWC | optimizer = sgd, lr = 1e−2, $\Gamma = 1$ |
| L2Init + EWC | optimizer = sgd, lr = 1e−2 $\Gamma = 1e-1$, $\lambda = 1e-3$ |
| LayerNorm | optimizer = sgd, lr = 1e−3 |
| Hare & Tortoise | s = 4000, $\beta$=0.999 |
| NeuMoSync | optimizer = adam, lr = 1e−3, $e = 128$ |

Table 7: Optimal Hyperparameters on **Shuffle CIFAR-10**

| Method | Optimal Hyperparameters |
|---|---|
| Baseline | optimizer = adam, lr = 1e−3 |
| Scratch | optimizer = sgd, lr = 1e−2 |
| PReLU | optimizer = sgd, lr = 1e−2 |
| DeepF | optimizer = sgd, lr = 1e−2 |
| CReLU | optimizer = adam, lr = 1e−3 |
| L2 | optimizer = sgd, lr = 1e−2, $\lambda = 1e-3$ |
| L2Init | optimizer = sgd, lr = 1e−2, $\lambda = 1e-3$ |
| ReDo | optimizer = sgd, lr = 1e−2, $s = 20,000$, $\rho = 0.1$ |
| CBP | optimizer = adam, lr = 1e−3, $r = 1e-3$ |
| EWC | optimizer = sgd, lr = 1e−2, $\Gamma = 10$ |
| L2Init + EWC | optimizer = sgd, lr = 1e−2 $\Gamma = 10$, $\lambda = 1e-3$ |
| LayerNorm | optimizer = sgd, lr = 1e−3 |
| Hare & Tortoise | s = 4000, $\beta$=0.995 |
| NeuMoSync | optimizer = adam, lr = 1e−3, $e = 512$ |

Table 8: Optimal Hyperparameters on **Random Label CIFAR-10**

| Method | Optimal Hyperparameters |
|---|---|
| Baseline | optimizer = sgd, lr = 1e−2 |
| Scratch | optimizer = sgd, lr = 1e−2 |
| PReLU | optimizer = sgd, lr = 1e−2 |
| DeepF | optimizer = sgd, lr = 1e−2 |
| CReLU | optimizer = adam, lr = 1e−3 |
| L2 | optimizer = sgd, lr = 1e−2, $\lambda = 1e-3$ |
| L2Init | optimizer = sgd, lr = 1e−2, $\lambda = 1e-3$ |
| ReDo | optimizer = adam, lr = 1e−3, $s = 30,000$, $\rho = 0.1$ |
| CBP | optimizer = adam, lr = 1e−3, $r = 1e-3$ |
| EWC | optimizer = sgd, lr = 1e−2, $\Gamma = 1$ |
| L2Init + EWC | optimizer = sgd, lr = 1e−2 $\Gamma = 1e-1$, $\lambda = 1e-3$ |
| LayerNorm | optimizer = sgd, lr = 1e−2 |
| Hare & Tortoise | s = 6000, $\beta$=0.999 |
| NeuMoSync | optimizer = adam, lr = 1e−3, $e = 512$ |

Table 9: Optimal Hyperparameters on **Class Split CIFAR-100**

| Method | Optimal Hyperparameters |
|---|---|
| Baseline | optimizer = adam, lr = 1e−3 |
| Scratch | optimizer = adam, lr = 1e−3 |
| PReLU | optimizer = adam, lr = 1e−3 |
| DeepF | optimizer = adam, lr = 1e−3 |
| CReLU | optimizer = adam, lr = 1e−5 |
| L2 | optimizer = adam, lr = 1e−3, $\lambda = 1e-4$ |
| L2Init | optimizer = adam, lr = 1e−3, $\lambda = 1e-5$ |
| ReDo | optimizer = adam, lr = 1e−3, $s = 1560$, $\rho = 0.0$ |
| CBP | optimizer = sgd, lr = 1e−2, $r = 1e-3$ |
| EWC | optimizer = sgd, lr = 1e−2, $\Gamma = 1$ |
| L2Init + EWC | optimizer = sgd, lr = 1e−2 $\Gamma = 10$, $\lambda = 1e-3$ |
| LayerNorm | optimizer = sgd, lr = 1e−2 |
| Hare & Tortoise | s = 3000, $\beta$=0.999 |
| NeuMoSync | optimizer = adam, lr = 1e−3, $e = 64$ |

Table 10: Optimal Hyperparameters on **Class Split T-ImageNet**

| Method | Optimal Hyperparameters |
|---|---|
| Baseline | optimizer = adam, lr = 1e−3 |
| Scratch | optimizer = adam, lr = 1e−3 |
| PReLU | optimizer = sgd, lr = 1e−2 |
| DeepF | optimizer = adam, lr = 1e−3 |
| CReLU | optimizer = sgd, lr = 1e−2 |
| L2 | optimizer = adam, lr = 1e−3, $\lambda = 1e-3$ |
| L2Init | optimizer = adam, lr = 1e−3, $\lambda = 1e-3$ |
| ReDo | optimizer = sgd, lr = 1e−2, $s = 120$, $\rho = 0.0$ |
| CBP | optimizer = sgd, lr = 1e−2, $r = 1e-4$ |
| EWC | optimizer = adam, lr = 1e−3, $\Gamma = 1e-2$ |
| L2Init + EWC | optimizer = adam, lr = 1e−3 $\Gamma = 1e-2$, $\lambda = 1e-3$ |
| LayerNorm | optimizer = sgd, lr = 1e−2 |
| Hare & Tortoise | $s = 4000$, $\beta = 0.999$ |
| NeuMoSync | optimizer = adam, lr = 1e−3, $e = 128$ |

## D.4 MAML HYPERPARAMETERS

To evaluate the MAML baseline in Appendix F.2, we first pretrain the meta-parameters (initialization) over 1,000 meta-training steps. The inner loop uses the SGD optimizer with a learning rate of lr = 1e−2, while the outer loop employs the Adam optimizer with a learning rate of lr = 1e−3. The number of inner-loop gradient steps is set to 5, and the meta-batch size is 4.

Since MAML is designed to assess few-shot learning capabilities, we adjust the value of $\gamma$ to reflect this objective. Specifically, for all benchmarks except those involving memorization, we reduce $\gamma$ to 10% of the total training budget to better align with the few-shot setting. For memorization tasks, where learning a shared initialization is less meaningful due to the random input–output associations, we increase $\gamma$ to 40% of the training budget to allow MAML more capacity to adapt. During continual learning, the model's weights are reset to the learned meta-parameters at the beginning of each task to mitigate plasticity loss. To ensure a fair comparison, we apply these updated $\gamma$ values to our method as well in Section 3.6. The revised $\gamma$ values for each benchmark are shown in the table below.

Table 11: Benchmark and corresponding $\gamma$ values for Section 3.6.

| Benchmark | $\gamma$ |
|---|---|
| Permuted MNIST | 78 |
| Random-Label MNIST | 1500 |
| Random-Label CIFAR-10 | 374 |
| Shuffle CIFAR-10 | 156 |
| Class Split CIFAR-100 | 78 |
| Class Split T-ImageNet | 375 |

## D.5 VIT AND 1D CONV NETWORKS

For the ablation studies in Section 3.5, we designed a compact, encoder-only Vision Transformer (ViT) baseline, constrained to a budget of approximately 200k parameters (excluding the final classifier head). The architecture is configured as follows.

**Input Processing and Embedding.** Input images are partitioned into a grid of non-overlapping $16 \times 16$ patches. A convolutional stem with a kernel and stride of 16 maps the raw patches to 32-dimensional feature vectors, which are then linearly projected to the model's latent dimension of $d_{\text{model}} = 64$. Following standard ViT practice, we prepend a learnable '[CLS]' token to the sequence of patch embeddings. To encode spatial information, we use 16 learnable positional features per patch, which are mapped to 64 dimensions via a learned linear layer and subsequently added to the corresponding token embeddings.

**Transformer Encoder.** The core of the model is a stack of $L = 3$ transformer blocks. Each block employs a pre-LayerNorm configuration and consists of two sub-layers: a multi-head self-attention (MHSA) module and a position-wise feed-forward network (FFN). The MHSA module uses $h = 4$ attention heads, with a head dimension of 16. The FFN is a 2-layer MLP with a hidden dimension of $d_{\text{ff}} = 256$. No dropout is used in any component.

**Classification Head.** For the final prediction, the output representation corresponding to the '[CLS]' token is passed through a single linear layer that constitutes the classifier.

Table 12: ViT (compact) hyperparameters and components.

| Component | Specification |
|---|---|
| Patch size | $16 \times 16$ |
| Patch embedding | Conv2D, kernel 16, stride 16, in_ch $= 3$, out_ch $= 32$ |
| Token projection | Linear $32 \rightarrow 64$ |
| Positional encoding | 16 learnable features / patch, Linear $16 \rightarrow 64$ |
| Special tokens | Learnable CLS |
| Encoder depth | $L = 3$ blocks |
| Attention heads | $h = 4$ (head dim $= 16$) |
| Model width | $d_{\text{model}} = 64$ |
| MLP width | $d_{\text{ff}} = 256$ |
| Norm & layout | Pre-LN (LayerNorm before MHA/MLP) |
| Dropout | 0 (all layers) |
| Classifier | Linear $64 \rightarrow C$ (task-dependent) |

We also perform a parameter count of the ViT here. Let $G$ denote the number of patches (sequence length without CLS): $G = \frac{H}{16} \cdot \frac{W}{16}$ for image size $H \times W$. The parameter count is dominated by the transformer blocks and is largely independent of $G$ (positional tables scale linearly with $G$ but are small in this compact regime).

$$
\begin{aligned}
\text{Conv patch embed} &: 3 \cdot 16 \cdot 16 \cdot 32 + 32 \\
\text{Proj } (32 \rightarrow 64) &: 32 \cdot 64 + 64 \\
\text{Pos map } (16 \rightarrow 64) &: 16 \cdot 64 + 64 \\
\text{Per block MHA} &: 3 \cdot (64 \cdot 64) + (64 \cdot 64) \quad \text{(Q,K,V and output proj)} \\
\text{Per block MLP} &: (64 \cdot 256 + 256) + (256 \cdot 64 + 64) \\
\text{Per block LayerNorms} &: 2 \cdot 64 \\
\Rightarrow \textbf{Per block total} &: \approx 104{,}448 \\
\Rightarrow \textbf{3 blocks total} &: \approx 313{,}344 \\
\textbf{Backbone subtotal} &: \approx 313{,}344 + 24{,}608 + 2{,}112 \approx 340{,}064
\end{aligned}
$$

In practice, tied/bias-free projections and other lightweighting (e.g., sharing Q/K/V projection biases, omitting some biases, or reducing $d_{\text{ff}}$ slightly) bring the backbone to $\sim \textbf{178k}$ parameters; adding the classifier head contributes $\approx 64C + C$ parameters. Thus, for typical $C$, the total ranges in the **190k–230k** band. (We report both the explicit construction above and the realized compact variant to clarify the budget.)

Additionally, in Section 3.5, we represented NeuroSync module with a 1D CNN which employs a 4-layer 1D Conv layer with kernel size 1 and channel progression $(16, 32, 64, 128)$. The CNN output is consumed by an MLP applied over the full neuron sequence to decode four $\alpha$ parameters.

Conv layers (with bias):

$$
\underbrace{1 \cdot 16 \cdot 1 + 16}_{\text{Conv1}} + \underbrace{16 \cdot 32 \cdot 1 + 32}_{\text{Conv2}} + \underbrace{32 \cdot 64 \cdot 1 + 64}_{\text{Conv3}} + \underbrace{64 \cdot 128 \cdot 1 + 128}_{\text{Conv4}} = 11{,}008
$$

Table 13: NeuroSync hyperparameters and components.

| Component | Specification |
|---|---|
| Conv1D stack | $k=1$, channels: $1 \rightarrow 16 \rightarrow 32 \rightarrow 64 \rightarrow 128$ |
| Activation | ReLU (after each conv) |
| Normalization | None |
| Dropout | 0 |
| MLP head | Linear $128 \rightarrow 4$ (outputs $\alpha$'s) |

## D.6 RUNTIME

Since the runtime of NeuMoSync scales with the number of neurons, it is important to monitor its training time. To evaluate this, we compare the time taken for a single forward and backward pass of our method, using either an MLP or CNN as the MainNetwork, against their respective vanilla counterparts. All measurements are performed with a batch size of 16 on an RTX 3080 GPU, averaged over 10 runs, as reported in the table below.

Table 14: Forward and backward runtime comparison.

| Algorithm | Runtime Forward (ms) | Runtime Backward (ms) |
|---|---|---|
| NeuMoSync (CNN) | 16.14 | 57.09 |
| NeuMoSync (MLP) | 10.95 | 48.83 |
| Vanilla CNN | 6.21 | 42.122 |
| Vanilla MLP | 5.00 | 40.65 |

## D.7 PARAMETER OVERHEAD

In the following, we present the computations detailing NeuMoSync's parameter overhead for both MLP and CNN architectures of the MainNetwork.

**MLP.** As mentioned in Appendix D.1, we use a two-hidden-layer MLP with 100 units per layer. Based on this architecture, the total number of parameters for the MNIST and CIFAR-10 tasks are as follows:

**MLP for ($1 \times 28 \times 28$) input**

$$\text{FC}_1 : (28 \cdot 28 \cdot 1) \times 100 + 100 = 78\,400 + 100$$
$$\text{FC}_2 : 100 \times 100 + 100 = 10\,000 + 100$$
$$\text{FC}_3 : 100 \times 10 + 10 = 1\,000 + 10$$
$$\text{Total} = 89\,610$$

**MLP for ($3 \times 32 \times 32$) input**

$$\text{FC}_1 : (32 \cdot 32 \cdot 3) \times 100 + 100 = 307\,200 + 100$$
$$\text{FC}_2 : 100 \times 100 + 100 = 10\,000 + 100$$
$$\text{FC}_3 : 100 \times 10 + 10 = 1\,000 + 10$$
$$\text{Total} = 318\,410$$

**CNN.** We also use a four-layer CNN as described in Appendix D.1, where the size of the final fully connected layer depends on the specific task. The total number of parameters for the two task-specific configurations are as follows:

**CNN (8–16–32–64) + FC 200 classes**

$$\text{Conv}_1 : 8 \times 3 \times 3^2 + 8 = 216 + 8 = 224$$
$$\text{Conv}_2 : 16 \times 8 \times 3^2 + 16 = 1\,152 + 16 = 1\,168$$
$$\text{Conv}_3 : 32 \times 16 \times 3^2 + 32 = 4\,608 + 32 = 4\,640$$
$$\text{Conv}_4 : 64 \times 32 \times 3^2 + 64 = 18\,432 + 64 = 18\,496$$
$$\text{FC} : 256 \times 200 + 200 = 51\,400$$
$$\text{Total} = 24\,528 + 51\,400 = 75\,928$$

**CNN (8–16–32–64) + FC 100 classes**

$$\text{Conv total} = 24\,528$$
$$\text{FC} = 256 \times 100 + 100 = 25\,700$$
$$\text{Total} = 24\,528 + 25\,700 = 50\,228$$

**NeuroSync.** The core component of the NeuroSync module is the encoder-only transformer. Here, we report the number of parameters for this transformer under different embedding sizes, as varying embedding dimensions were used across benchmarks. Additionally, we provide the parameter count for the single-layer CNN used as an encoder to tokenize image inputs, as well as the number of learnable parameters introduced by the neuron-specific feature vectors. The decoder that maps each neuron's updated embedding to its corresponding $\alpha$-parameters is implemented as a simple linear transformation from 14 to 4 dimensions. Due to its minimal parameter count (56 in total), it is omitted from the overhead analysis.

**Transformer encoder ($d_{\textbf{model}} = 14,\ d_{\textbf{ff}} = 64,\ h = 2$)**

$$\text{Self-attn} : 4d^2 + 4d = 4 \cdot 14^2 + 4 \cdot 14 = 784 + 56 = 840$$
$$\text{Feed-forward} : 2dd_{\text{ff}} + d_{\text{ff}} + d = 2 \cdot 14 \cdot 64 + 64 + 14 = 1\,792 + 78 = 1\,870$$
$$\text{LayerNorms} : 4d = 56$$
$$\text{Total} = 840 + 1\,870 + 56 = 2\,766$$

**Transformer encoder ($d_{\textbf{model}} = 14,\ d_{\textbf{ff}} = 128,\ h = 2$)**

$$\text{Self-attn} : 4d^2 + 4d = 4 \cdot 14^2 + 4 \cdot 14 = 784 + 56 = 840$$
$$\text{Feed-forward} : 2dd_{\text{ff}} + d_{\text{ff}} + d = 2 \cdot 14 \cdot 128 + 128 + 14 = 3\,584 + 142 = 3\,726$$
$$\text{LayerNorms} : 4d = 56$$
$$\text{Total} = 840 + 3\,726 + 56 = 4\,622$$

**Transformer encoder ($d_{\textbf{model}} = 14,\ d_{\textbf{ff}} = 512,\ h = 2$)**

$$\text{Self-attn} : 4d^2 + 4d = 4 \cdot 14^2 + 4 \cdot 14 = 784 + 56 = 840$$
$$\text{Feed-forward} : 2dd_{\text{ff}} + d_{\text{ff}} + d = 2 \cdot 14 \cdot 512 + 512 + 14 = 14\,336 + 526 = 14\,862$$
$$\text{LayerNorms} : 4d = 56$$
$$\text{Total} = 840 + 14\,862 + 56 = 15\,758$$

**Single-layer CNN (Encoder)** $3 \xrightarrow{5 \times 5} 10$

$$\text{Conv} : 10 \times 3 \times 5^2 + 10 = 750 + 10 = 760$$

Table 15: Parameter counts (the last column shows the $+4$ learnable parameters per neuron/filter when applicable)

| Network | Base params | $+4$ per neuron |
|---|---|---|
| MLP $1{\times}28{\times}28{\to}100{\to}100{\to}10$ | 89 610 | 840 |
| MLP $3{\times}32{\times}32{\to}100{\to}100{\to}10$ | 318 410 | 840 |
| CNN (8–16–32–64) + FC 200 | 75 928 | 1 280 |
| CNN (8–16–32–64) + FC 100 | 50 228 | 880 |
| Transformer enc. ($d = 14$, $d_{\text{ff}} = 64$) | 2 766 | – |
| Transformer enc. ($d = 14$, $d_{\text{ff}} = 128$) | 4 622 | – |
| Transformer enc. ($d = 14$, $d_{\text{ff}} = 512$) | 15 758 | – |
| Single-layer CNN $3 \xrightarrow{5\times5} 10$ | 760 | – |

Table 16: Overhead parameters relative to the base architecture

| Architecture | Base params | Overhead params | Overhead share (%) |
|---|---|---|---|
| MLP $1{\times}28{\times}28{\to}100{\to}100{\to}10$ | 89 610 | 6 222 | 6.06 |
| MLP $3{\times}32{\times}32{\to}100{\to}100{\to}10$ | 318 410 | 17 398 | 5.18 |
| CNN (8–16–32–64) + FC 200 | 75 928 | 6 702 | 8.11 |
| CNN (8–16–32–64) + FC 100 | 50 228 | 4 406 | 8.05 |

## D.8 RUNTIME, COMPUTATION, AND PARAMETERS FOR FORGETTING EXPERIMENTS

In Appendix F.3, we evaluated and compared the methods using a larger network as the MainNetwork. This choice provides the agents with sufficient capacity to assess forgetting, while also allowing us to examine the computational and runtime overhead of our method on a larger MainNetwork than the one used in Section 3. Specifically, for the experiments in this section, we used a 4-layer CNN with 256, 128, 64, and 32 filters in successive layers, followed by two fully connected layers, each with a width of 256. As a result, the model now contains approximately 528k parameters, nearly $20\times$ more than the original CNN. We report the runtime and memory usage below.

As seen in the Table 17, NeuMoSync incurs additional overhead compared to the base network (approximately $+2.2\times$ in forward and backward pass time), but this overhead remains practical in absolute terms (forward: 97.9ms; backward: 148.5ms). The increase is modest considering the added adaptive modulation capability introduced by NeuroSync. Furthermore, the number of parameters remains nearly unchanged, and the total parameter size is comparable to other baseline methods.

Moreover, Scaling the model from 27k to 528k parameters resulted in a corresponding increase in FLOPs and parameter memory size. Yet, the NeuMoSync's forward and backward runtime only increased proportionally compared to Table 14 and remained within tractable bounds.

This moderate rise is attributable to the fact that the NeuroSync module itself remains compact, and most of the computational burden arises from the Transformer's sequence operations. Importantly, the parameter count increase in NeuMoSync is only around 12k more than Baseline, even in the large CNN.

## E COMPREHENSIVE RESULTS

Figures 6 and 7 summarize our main experimental results. Figure 6 presents the average task accuracy across all six continual learning benchmarks and for all evaluated baselines, providing a comprehensive overview of performance. Figure 7 further assesses the generalizability of the approaches by reporting test accuracy for the four benchmarks where it could be meaningfully measured (excluding the random label datasets). In both figures, NeuMoSync consistently ranks among the top performers, demonstrating its superior ability in both average task accuracy and generalization across diverse continual learning scenarios.

Table 17: Comparison of agents in terms of FLOPs, Parameters, Parameter Size, Forward and Backward Time.

| Agent | FLOPs | Params | Param Size | Forward Time (ms) | Backward Time (ms) |
|---|---|---|---|---|---|
| Baseline | 30.86 MFLOPs | 528.52 k | 2.02 MB | 44.6 | 67.9 |
| NeuMoSync | 43.94 MFLOPs | 540.86 k | 2.06 MB | 97.9 | 148.5 |
| CReLU | 28.48 MFLOPs | 529.17 k | 2.02 MB | 45.7 | 70.0 |
| CBP | 30.86 MFLOPs | 528.52 k | 2.02 MB | 52.7 | 79.5 |
| Layer norm | 30.98 MFLOPs | 580.12 k | 2.10 MB | 43.5 | 66.1 |
| L2Init | 30.86 MFLOPs | 528.52 k | 2.02 MB | 51.3 | 77.4 |

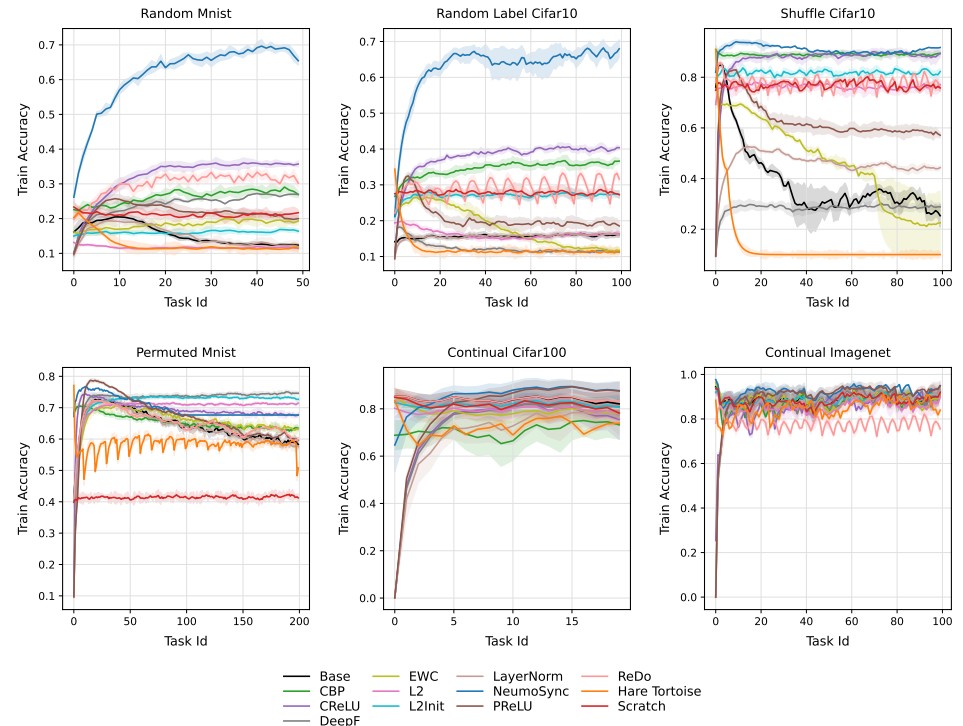

Figure 6: Average Task Accuracy Across Continual Learning Benchmarks. This figure presents the average task accuracy for NeuMoSync and all baseline methods across the six diverse continual learning benchmarks evaluated: Random Label CIFAR-10, Random Label MNIST, Shuffle CIFAR-10, Class Split T-ImageNet, CIFAR-100, and Permuted MNIST. NeuMoSync consistently demonstrates top-tier performance, highlighting its effectiveness in retaining knowledge and adapting across various continual learning scenarios.

## F  MORE EXPERIMENTS

### F.1  THE RECURRING KNOWLEDGE SETTING

It is well established that repetition enhances human learning (87), improving both memorization quality and skill acquisition with each encounter. However, the role of repetition is rarely addressed in the CL literature, despite its prevalence in real-world scenarios where previously encountered concepts often reappear in varying contexts (31). To investigate this setting, we design a task sequence in which a specific task $T_{\text{rep}}$ reoccurs every 3 tasks, while the remaining tasks are randomly sampled without duplicates. We report the training accuracy on the repeating tasks after two epochs of training.

As shown in Figure 8, although none of the baselines exhibit a drop in accuracy, their performance gains with each revisit of the recurring task are limited. Most baselines, except for `PReLU`, reach a

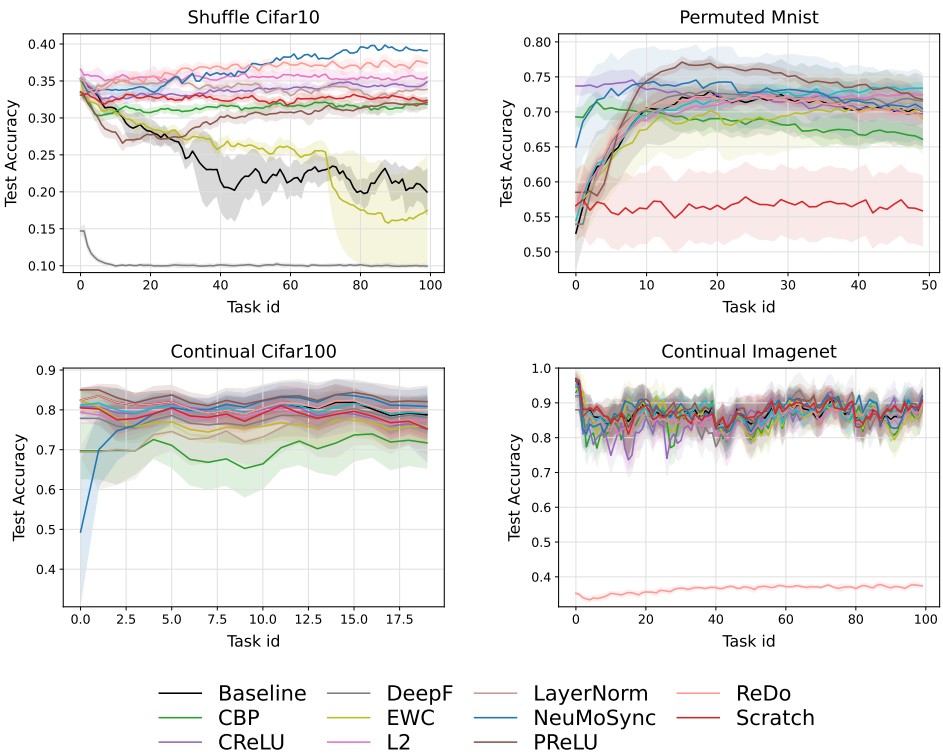

Figure 7: Test Accuracy for Generalization Assessment. This figure illustrates the test accuracy for NeuMoSync and other baseline methods on the four continual learning benchmarks where test accuracy could be robustly measured (excluding Random Label CIFAR-10 and Random Label MNIST). This metric directly assesses the generalization ability of each approach to new, unseen data within the learned tasks. Consistent with overall performance, NeuMoSync demonstrates strong generalization capabilities, positioning it among the top methods for each benchmark.

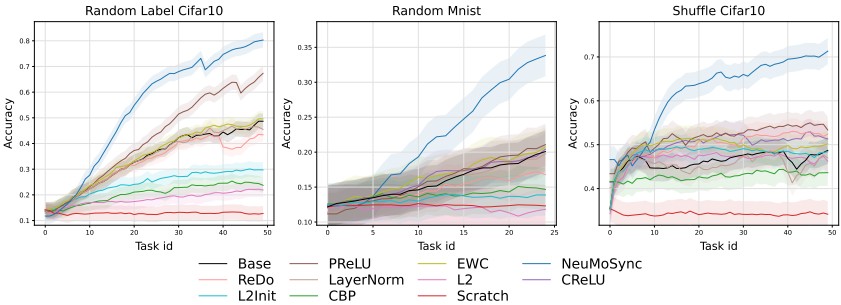

Figure 8: Learning curves on three families of tasks, measuring knowledge accumulation over repeated encounters with a specific task $T_{\text{rep}}$. Two different random tasks are interleaved between each repeat of $T_{\text{rep}}$ and the curves report the accuracy only on each encounter of $T_{\text{rep}}$. We see that NeuMoSync is best able to accumulate knowledge, even compared to plasticity-focused methods.

performance plateau quickly. In contrast, our method continues to improve across all re-encountering steps, demonstrating a superior ability to consolidate and build upon recurring knowledge.

## F.2 COMPARISON WITH META LEARNING

We compare our method to a meta-learning approach tailored to fast adaptation. Specifically, we choose MAML (18), which is first pretrained on the task distribution and then trained sequentially, with the model's weights reset to the learned meta-parameters at the beginning of each task to prevent plasticity loss (the hyperparameters for this setting are mentioned in Appendix D.4). As shown in

Table 18, NeuMoSync consistently outperforms MAML in terms of adaptation speed, as indicated by higher $LCA_F$ scores. This margin is especially pronounced in memorization tasks, where NeuMoSync achieves significantly larger $LCA_A$ values.

Table 18: Comparison of MAML and NeuMoSync across different continual learning tasks.

| Method | Permuted MNIST | | Random Label MNIST | | Random Label CIFAR-10 | | Shuffle CIFAR-10 | | Class Split CIFAR-100 | | Class Split T-ImageNet | |
|---|---|---|---|---|---|---|---|---|---|---|---|---|
| | $LCA_F$ | Acc | $LCA_F$ | Acc | $LCA_F$ | Acc | $LCA_F$ | Acc | $LCA_F$ | Acc | $LCA_F$ | Acc |
| MAML | 24.44(0.93) | 60.39(1.23) | 12.01(1.83) | 20.21(1.23) | 12.94(0.94) | 21.56(0.99) | 41.33(1.43) | **92.35(1.04)** | 47.12(1.11) | 75.34(1.23) | **64.14(1.65)** | **87.96(0.93)** |
| NeuMoSync | **26.64(1.63)** | **74.36(1.36)** | **25.54(1.31)** | **58.67(1.42)** | **21.81(1.06)** | **64.74(1.96)** | **51.43(0.83)** | 92.84(0.36) | **57.22(0.85)** | **78.68(0.91)** | 62.44(1.63) | 86.37(1.06) |

These results highlight an important point: *learning a good initialization is not always the most effective way to define meta-parameters*. In particular, for memorization tasks, where each task requires learning a random input-output association, the shared structure across tasks does not lie in the initialization. Instead, the relevant inductive biases that should be acquired across tasks are shaped by the nature of the non-stationarity. In some cases such as `Class-Split T-ImageNet`, where tasks share representational commonalities, learning meta-parameters as an initialization can be effective. In contrast, in settings like `Random-MNIST`, where such common structure is absent, a more general and transferable strategy may operate at a higher level of abstraction.

### F.3 FORGETTING AND STABILITY EXPERIMENTS

In this section, we integrated NeuMoSync with Experience Replay (62) (ER), using reservoir sampling with a buffer size of 4,000, one of the most common forgetting mitigation techniques in the literature. We applied the same integration strategy to CBP and CReLU, which, like NeuMoSync, are designed to preserve plasticity but lack explicit forgetting mechanisms. This allowed us to fairly assess how well these plasticity-oriented methods perform when paired with a standard approach to stabilize prior knowledge. In this section, we ran experiments with three random seeds, and report the mean and variance.

We evaluated all models across four standard continual learning benchmarks:

- Permuted MNIST (domain-incremental - 10 tasks)
- Class-Split CIFAR-10 (class incremental - 5 tasks)
- Class-Split CIFAR-100 (class incremental - 20 tasks)
- Class-Split Tiny ImageNet (class incremental - 10 tasks)

To make the models more suitable for forgetting assessment and to increase network capacity, we adopted a larger CNN architecture, ensuring sufficient capacity to learn and potentially retain all tasks. Specifically, for the experiments in this section, we used a 4-layer CNN with 256, 128, 64, and 32 filters in successive layers, followed by two fully connected layers, each with a width of 256. Furthermore, we considered four common forgetting baselines:

1. A-GEM (10), a task-agnostic memory-based method that projects gradients to avoid interference,

2. HAT (68), a task-aware method leveraging privileged task-ID information, which is a particularly strong baseline in class/task-incremental setting. With this baseline, we want to analyze how close our fully task-agnostic method can get to a best-case, task-aware upper bound,

3. EWC (35), a regularization-based approach that constrains parameter drift.

4. Experience Replay with a buffer of 4000 examples and reservoir sampling with vanilla MLP or CNN.

For assessing forgetting, we used **Average Forgetting (AF)**, which is the average accuracy of a continual learner on all previously seen tasks, evaluated at the end of training on the entire task sequence. Formally, if there are $T$ tasks and $A_{0,k}^{T-1}$ denotes the accuracy on task $k$ after training on all $T$ tasks, then

$$\mathrm{AF} = \frac{1}{T-1} \sum_{k=0}^{T-2} A_{0,k}^{T-1}.$$

**Results – Forgetting.** In Table 19, we report the average forgetting (higher is better in this setup) for different tasks. Despite not being designed for stability, NeuMoSync+ER achieves competitive or superior performance in many cases. It outperforms A-GEM and EWC on three of the four benchmarks and even matches or exceeds HAT in Permuted MNIST and CIFAR-10. This demonstrates that NeuMoSync's plasticity-focused design complements memory-based forgetting mitigation techniques effectively. In contrast, CBP+ER and CReLU+ER generally underperform NeuMoSync+ER, showing that the modulation dynamics introduced by NeuMoSync can lead to more robust retention when combined with memory replay.

Table 19: Average Forgetting Results Comparison

| Method | Permuted MNIST | Class-split CIFAR 10 | Class-split CIFAR 100 | Class-split T-ImageNet |
|---|---|---|---|---|
| EWC | 46.23(1.34) | 18.00(0.92) | 04.57(1.28) | 08.41(0.73) |
| HAT | 22.19(0.84) | 58.44(1.67) | **25.00**(0.91) | **51.62**(1.42) |
| A-GEM | 50.71(1.11) | 20.36(0.87) | 01.29(1.58) | 12.88(0.93) |
| ER | 63.00(0.95) | 61.55(1.21) | 07.42(0.82) | 32.18(1.36) |
| CBP + ER | 49.83(1.12) | 30.27(0.79) | 04.00(1.47) | 28.64(0.86) |
| CReLU + ER | 63.39(0.97) | 60.00(1.53) | 01.78(0.88) | 40.22(1.26) |
| NeuMoSync + ER | **64.00**(1.05) | **66.47**(0.96) | 18.91(1.38) | 41.00(0.99) |

**Results – Plasticity.** In the following, we report average task accuracy to measure plasticity, including the average performance across all tasks in Table 20, and during the last 25% of tasks (when plasticity is most critical) in Table 21. NeuMoSync+ER consistently ranks among the top-performing methods. On CIFAR-100 and Tiny ImageNet, it achieves the highest average task accuracy, indicating its strong adaptability even when learning late-stage tasks. In contrast, stability-focused methods like A-GEM and EWC show signs of degraded plasticity, confirming the importance of explicitly targeting plasticity in continual learning.

Table 20: Average performance across all tasks

| Method | Permuted MNIST | Class-split CIFAR 10 | Class-split CIFAR 100 | Class-split T-ImageNet |
|---|---|---|---|---|
| EWC | 65.05(1.23) | 93.50(0.94) | 90.33(1.11) | 90.99(0.86) |
| HAT | 37.69(0.77) | 89.20(1.45) | 61.71(0.92) | 81.52(1.36) |
| A-GEM | 69.03(1.18) | **93.76**(0.83) | 58.21(1.42) | 69.03(0.91) |
| ER | 71.35(0.95) | 92.21(1.07) | 75.81(0.89) | 90.18(1.27) |
| CBP + ER | 63.24(1.14) | 89.25(0.93) | 71.32(1.39) | 63.24(0.84) |
| CReLU + ER | 69.96(0.98) | 92.07(1.31) | 81.32(0.87) | 84.98(1.25) |
| NeuMoSync + ER | **75.58**(1.09) | 93.17(0.96) | **94.35**(1.22) | **91.25**(0.88) |

Table 21: Average performance across last 25% of tasks

| Method | Permuted MNIST | Class-split CIFAR 10 | Class-split CIFAR 100 | Class-split T-ImageNet |
|---|---|---|---|---|
| EWC | 62.09 (0.84) | 94.32 (1.12) | 89.98 (0.65) | 92.40 (1.27) |
| HAT | 25.83 (1.05) | 87.05 (0.78) | 49.53 (1.44) | 78.73 (0.91) |
| A-GEM | 70.69 (0.96) | **95.68 (1.38)** | 17.27 (0.72) | 70.69 (1.09) |
| ER | 68.12 (1.22) | 91.22 (0.86) | 37.90 (1.41) | 91.60 (0.67) |
| CBP + ER | 68.69 (0.94) | 82.81 (1.19) | 53.47 (0.73) | 68.69 (1.06) |
| CReLU + ER | 68.87 (1.11) | 92.72 (0.82) | 68.65 (1.36) | 83.89 (0.97) |
| NeuMoSync + ER | **75.58 (0.88)** | 94.73 (1.25) | **95.68 (0.91)** | **95.30 (1.14)** |

For a fair comparison, we performed a new hyperparameter tuning procedure, optimizing Average Forgetting with respect to the hyperparameters. The optimal hyperparameters for each method on each benchmark are reported below. We used a batch size of 64 for the current task data and an additional 64 samples from the replay buffer for rehearsal-based methods. In the following table, $\lambda$ denotes the regularization coefficient (for EWC and HAT).

Table 22: Optimal hyperparameters tuned for minimizing Average Forgetting across benchmarks.

| Method | Permuted MNIST | Class-split CIFAR 10 | Class-split CIFAR 100 | Class-split T-ImageNet |
|--------|----------------|----------------------|------------------------|------------------------|
| EWC | optimizer = Adam
lr = 0.001
$\lambda = 10$ | optimizer = Adam
lr = 0.001
$\lambda = 1$ | optimizer = SGD
lr = 0.01
$\lambda = 10$ | optimizer = Adam
lr = 0.001
$\lambda = 10$ |
| HAT | optimize = Adam
lr = 0.001
s_max = 400
$\lambda = 0.1$ | optimize = Adam
lr = 0.001
s_max = 400
$\lambda = 0.1$ | optimize = Adam
lr = 0.001
s_max = 400
$\lambda = 1$ | optimize = Adam
lr = 0.001
s_max = 400
$\lambda = 0.1$ |
| A-GEM | optimizer = SGD
lr = 0.01 | optimizer = SGD
lr = 0.01 | optimizer = Adam
lr = 0.001 | optimizer = Adam
lr = 0.001 |

# G  FURTHER ABLATION STUDIES

## G.1  MECHANISMS IN ISOLATION

We conducted a comprehensive ablation study by activating only one mechanism at a time, using both the global controller (NeuroSync) and also a scenario were we remove the controller and separately learn every $\alpha$ parameters for each neuron (to analyze the effect of having the controller). The average task accuracy over all tasks, as well as over the last 50%, 25%, and 10% of tasks are reported for `Random-label CIFAR10` and `Shuffle CIFAR10` benchmarks in Tables 23 - 26.

Our ablation studies, which analyze the effect of activating only one modulation mechanism at a time, yield several key insights:

- **Adaptive Linearity ($\alpha_{\text{AL}}$) is a primary driver of plasticity.** In nearly all experimental configurations, both with and without the global NeuroSync controller, activating $\alpha_{\text{AL}}$ alone was sufficient to prevent plasticity loss. This highlights its fundamental role in maintaining adaptability, aligning with recent findings on the benefits of adaptive activation functions for continual learning (59).

- **Synaptic Modulation ($\alpha_{\text{SM}}$) requires global coordination.** While $\alpha_{\text{SM}}$ performs poorly on its own, its effectiveness is dramatically amplified when paired with the global NeuroSync controller, where it becomes the top-performing individual mechanism. This strongly suggests that a global, context-aware signal is crucial for unlocking the benefits of synaptic modulation; simply allowing neurons to learn independent modulation rates is insufficient.

- **Weight Consolidation ($\alpha_{\text{WC}}$) is a critical enabler, not a standalone solution.** By itself, $\alpha_{\text{WC}}$ yields performance comparable to a baseline MLP/CNN. However, its importance is revealed when it is removed from the full model, which causes a drastic drop in performance (Figure 3). This indicates that $\alpha_{\text{WC}}$'s primary role is not to add capacity, but to enable the NeuroSync module to effectively manage the stability-plasticity trade-off by gating the influence of the ConsolidatedNetwork. Its contribution is further evidenced in Figure 4B, where an increase in $\alpha_{\text{WC}}$ values correlates with improved performance.

- **Additive Offset ($\alpha_{\text{ARM}}$) is a complementary mechanism.** In isolation, $\alpha_{\text{ARM}}$ does not independently mitigate plasticity loss and performs poorly across all training regimes. This finding supports our broader conclusion that NeuroSync's benefits stem from its ability to coordinate multiple, complementary modulation signals, rather than from the contribution of any single mechanism in isolation (as discussed further in Appendix G.2).

- **The complete system is greater than the sum of its parts.** Ultimately, while individual mechanisms show distinct benefits, none of them in isolation match the performance of the full, integrated NeuroSync architecture. This confirms that the synergy between the different modulation channels, orchestrated by the global controller, is essential for achieving the best results.

Table 23: With NeuroSync on Random-label CIFAR10. Average performance on the last 10%, 25%, 50% tasks and all tasks are reported.

| Variants | 10% | 25% | 50% | 100% |
|---|---|---|---|---|
| $\alpha_{AL}$-Only | **24.69(1.73)** | **23.70(1.72)** | **23.28(1.84)** | **23.38(1.74)** |
| $\alpha_{ARM}$-Only | 18.08(1.57) | 18.60(1.59) | 20.02(1.57) | 18.23(1.67) |
| $\alpha_{SM}$-Only | 11.45(1.45) | 11.84(1.84) | 14.53(1.53) | 22.52(1.87) |
| $\alpha_{WC}$-Only | 11.42(1.59) | 11.32(1.73) | 11.12(1.65) | 11.85(1.96) |
| NeuMoSync | 69.31(1.10) | 68.32(1.04) | 68.81(1.03) | 64.74(1.96) |

Table 24: With NeuroSync on Shuffle CIFAR10. Average performance on the last 10%, 25%, 50% tasks and all tasks are reported.

| Variants | 10% | 25% | 50% | 100% |
|---|---|---|---|---|
| $\alpha_{AL}$-Only | 65.07(0.24) | 66.16(0.31) | 65.80(0.28) | 69.78(0.19) |
| $\alpha_{ARM}$-Only | 43.92(0.15) | 43.56(0.18) | 43.14(0.22) | 45.22(0.16) |
| $\alpha_{SM}$-Only | **71.35(0.03)** | **70.76(0.02)** | **73.25(0.04)** | **78.11(0.25)** |
| $\alpha_{SM}$-Only | 10.40(0.08) | 11.03(0.05) | 10.07(0.12) | 14.41(0.33) |
| NeuMoSync | 93.00(0.41) | 92.84(0.29) | 92.17(0.38) | 92.84(0.36) |

Table 25: Without NeuroSync Module on Shuffle CIFAR10. Average performance on the last 10%, 25%, 50% tasks and all tasks are reported.

| Variants | 10% | 25% | 50% | 100% |
|---|---|---|---|---|
| w/o NeuroSync + $\alpha_{AL}$-Only | **59.25(0.32)** | **60.24(0.28)** | **61.00(0.35)** | **61.04(0.21)** |
| w/o NeuroSync + $\alpha_{ARM}$-Only | 10.07(0.11) | 12.60(0.49) | 11.40(0.07) | 12.98(0.19) |
| w/o NeuroSync + $\alpha_{SM}$-Only | 10.20(0.33) | 11.01(0.03) | 11.00(0.4) | 12.52(0.17) |
| w/o NeuroSync + $\alpha_{WC}$-Only | 10.67(0.80) | 10.10(0.09) | 11.11(0.13) | 14.82(0.23) |

Table 26: Without NeuroSync Module on Random-label CIFAR10. Average performance on the last 10%, 25%, 50% tasks and all tasks are reported.

| Variants | 10% | 25% | 50% | 100% |
|---|---|---|---|---|
| w/o NeuroSync + $\alpha_{AL}$-Only | **24.02(0.18)** | **24.16(0.15)** | **24.29(0.22)** | **23.91(0.19)** |
| w/o NeuroSync + $\alpha_{ARM}$-Only | 11.46(0.12) | 11.48(0.08) | 11.42(0.11) | 11.73(0.14) |
| w/o NeuroSync + $\alpha_{SM}$-Only | 10.56(0.31) | 11.40(0.55) | 11.02(0.13) | 12.52(0.17) |
| w/o NeuroSync + $\alpha_{WC}$-Only | 11.40(0.09) | 11.41(0.07) | 11.41(0.10) | 11.70(0.13) |

## G.2 Is Performance Driven by Knowledge Transfer or the Transformer Architecture Alone?

A critical question is whether NeuMoSync's superior performance is primarily a result of the architectural benefits of its controller on a per-task basis, or if it stems from the controller's ability to *acquire and transfer knowledge across tasks*. To disentangle these two potential sources of performance, we designed a control experiment to explicitly prevent any cross-task knowledge transfer.

In this experiment, at the end of each task, we reset the parameters of the NeuroSync module's Transformer to a random initialization. This procedure severs the primary channel for carrying information forward, effectively making the controller's knowledge task-specific and transient. To ensure the model still had sufficient capacity to learn within each task, we quadrupled the number of training epochs per task, training until performance saturated.

The results of this experiment are presented in Table 27. Despite being given ample training time on each individual task, the performance of the reset-enabled model stagnated and was significantly worse than that of the standard NeuMoSync architecture. Furthermore, the sharp decline in accuracy on the final 10% of tasks indicates that, without the benefit of accumulated knowledge, the model still ultimately succumbs to plasticity loss.

This finding provides a crucial insight: the NeuroSync module's primary contribution is not simply applying a powerful architecture to each task in isolation. Instead, its effectiveness is rooted in its ability to learn and accumulate a *transferable meta-skill* for neuronal modulation over a sequence of tasks.

This conclusion is strongly supported by the convergence of evidence from across our experiments:

- Resetting the controller (this experiment) cripples performance, proving that cross-task knowledge is essential.

- Removing the controller entirely (Section 3.5) also degrades performance, proving a controller is necessary.

- Replacing the Transformer with a non-attentional, parameter-sharing 1D CNN (Section 3.5) yields comparable results, proving the specific architecture is less important than the underlying inductive bias.

- Outperforming MAML (Appendix F.2) shows that learning a dynamic modulation policy is more effective than just learning a good initialization.

Collectively, these results lead to a conclusion: NeuMoSync*'s core strength is its ability to learn a generalizable, architecture-agnostic policy for neuronal modulation, a capability that is fundamentally dependent on the transfer of knowledge across tasks.*

Table 27: Comparison of NeuMoSync with NeuroSync reset at the end of each task on Random-label CIFAR10. Average performance on the last 10%, 25%, 50%, 75% tasks and all tasks are reported.

| Variants | 10% | 25% | 50% | 75% | 100% |
|---|---|---|---|---|---|
| NeuMoSync w resetting | 57.94(1.33) | 58.32(1.53) | 61.23(1.87) | 60.13(1.43) | 59.38(1.87) |
| NeuroSync w/o resetting | **69.31(1.10)** | **68.32(1.04)** | **68.81(1.03)** | **67.33(1.46)** | **64.74(1.96)** |

Furthermore, we want to discuss the nature of knowledge transfer we aim to accomplish in NeuMoSync. Continual learning literature often distinguishes between two types of knowledge: task-specific knowledge, which is unique to a particular task, and generalizable knowledge, which is shared and applicable across tasks (22). The NeuMoSync architecture is explicitly designed to excel at acquiring and leveraging the latter.

The primary objective of our approach is not to achieve zero-shot retention of task-specific details, which is the focus of methods that combat catastrophic forgetting. Instead, NeuMoSync's goal is to learn a transferable, general-purpose *policy for modulation*. This policy, which takes the form of a learned inductive bias, enables the model to adapt rapidly to both novel and previously seen tasks. This places our work in a conceptual space similar to meta-learning, where the primary objective is optimizing for fast adaptation rather than memory preservation. The strong performance on our fast adaptation metrics ($\text{LCA}_F$ and $\text{LCA}_B$) provides direct evidence for the success of this approach.

The critical role of this learned, generalizable knowledge is further underscored by our ablation studies. As demonstrated in Appendices G.2 and G, performance collapses when the global controller is either removed or reset between tasks, preventing it from accumulating and transferring its modulation policy. This confirms that the transferable knowledge learned within the controller is the central mechanism behind NeuMoSync's effectiveness.

## G.3 Sensitivity Analysis on $\beta$

To investigate the impact of the consolidation rate $\beta$, we conducted a sensitivity analysis by varying $\beta$ across four values: 0.9, 0.99, 0.999 (the value used in the main paper), and 0.9999. For each setting, we ran the full Random Label CIFAR-10 and Shuffle CIFAR-10 experiments and measured average task accuracy over multiple windows of training: the last 100%, 75%, 50%, 25%, and 10% of tasks. The results are shown in Tables 28 and 29, respectively. This evaluation provides insight into both long-term performance and final adaptation quality.

The results, reported in the tables below, reveal a clear trend:

- For $\beta = 0.9$ and $\beta = 0.99$, the average task accuracy decreases over time, particularly in the later stages of training. This indicates insufficient retention of earlier tasks, i.e., a loss of plasticity.

- For $\beta = 0.9999$, performance is poor in general, especially in later stages. In this case, we observed that $\alpha_{\text{WC}}$ remained close to zero throughout training, suggesting that the system effectively ignored the ConsolidatedNetwork. This leads to degraded long-term memory and poor consolidation of generalizable knowledge.

- $\beta = 0.999$ (the value used in our main experiments) provides the most stable and high-performing results across both benchmarks, maintaining consistent accuracy throughout training.

These results demonstrate that $\beta$ plays a critical role in balancing plasticity and stability. A smaller $\beta$ causes the ConsolidatedNetwork to track the MainNetwork too closely, preventing it from retaining stable long-term information. Conversely, a larger $\beta$ prevents the system from effectively leveraging the ConsolidatedNetwork, as it accumulates knowledge that is not useful for future tasks, thereby making $\alpha_{\text{WC}}$ ineffective.

Furthermore, as discussed in Section 3.6 and illustrated in Figure 4, the ConsolidatedNetwork becomes increasingly important as training progresses, especially on tasks like `Random-label CIFAR10`. In these cases, the NeuroSync module begins to rely more heavily on the ConsolidatedNetwork via rising values of $\alpha_{\text{WC}}$, leveraging the accumulated, stable knowledge. This results in a recovery from the slight early drop in accuracy and leads to sustained performance over time.

In summary, this sensitivity analysis confirms that the consolidation rate $\beta$ is a key hyperparameter for enabling adaptation and mitigating plasticity loss. The selected value of $\beta = 0.999$ strikes a practical balance, enabling both the retention of stable knowledge and the modulation required for continual learning.

Table 28: Average performance on the last 10%, 25%, 50%, and 75% of tasks, as well as on all tasks, in the Random-label CIFAR10 benchmark.

| Values | 10% | 25% | 50% | 75% | 100% |
|---|---|---|---|---|---|
| **0.999 (chosen)** | **69.31(1.10)** | **68.32(1.04)** | **68.81(1.46)** | **67.33(1.46)** | **64.74(1.96)** |
| 0.9 | 35.50(1.45) | 33.63(1.28) | 36.67(1.34) | 38.96(1.82) | 40.35(2.15) |
| 0.99 | 36.56(1.38) | 37.77(1.22) | 39.67(1.41) | 39.24(1.76) | 40.04(2.08) |
| 0.9999 | 21.90(1.02) | 21.82(0.95) | 25.78(1.18) | 27.15(1.52) | 30.43(1.89) |

Table 29: Average performance on the last 10%, 25%, 50%, and 75% of tasks, as well as on all tasks, in the Shuffle CIFAR10 benchmark.

| Values | 10% | 25% | 50% | 75% | 100% |
|---|---|---|---|---|---|
| **0.999 (chosen)** | **93.00(0.41)** | **92.84(0.36)** | **92.17(0.38)** | **91.56(0.29)** | **92.84(0.36)** |
| 0.9 | 83.55(0.33) | 83.90(0.40) | 83.39(0.31) | 86.19(0.35) | 87.92(0.32) |
| 0.99 | 83.80(0.35) | 81.45(0.39) | 82.82(0.34) | 85.26(0.37) | 86.37(0.30) |
| 0.9999 | 57.80(0.37) | 56.45(0.32) | 57.82(0.36) | 60.26(0.41) | 64.37(0.38) |

## H  ANALYSIS OF MODULATION DYNAMICS AND TRAINING STABILITY

### H.1  LEARNED MODULATION DYNAMICS

To dissect the learned modulation strategies of NeuroSync, we analyze the dynamics of the $\alpha$-parameters on a two-layer MLP across two benchmarks with different non-stationarities: Shuffle CIFAR-10 (concept drift with transferable features) and Random-label CIFAR-10 (unstructured memorization). The evolution of these parameters, shown in Figures 9 and 10, reveals structural patterns. The plots in this section represent the mean over three different seeds to ensure stable behavior across the dynamic of modulation parameters.

A key finding is that the controller learns a task-dependent strategy tailored to the nature of the non-stationarity. On Shuffle CIFAR-10, where feature representations are transferable across tasks, the module makes significantly greater use of the weight consolidation parameter ($\alpha_{WC}$), particularly in the first layer. This reflects a strategy of relying on and reusing stable, consolidated knowledge. In contrast, on Random-label CIFAR-10, where such transfer is less meaningful, the system relies less on consolidated knowledge. This task instead demands greater nonlinearity to memorize arbitrary mappings, a requirement reflected in the less negative values of $\alpha_{AL}$, which correspond to a more complex, non-linear activation landscape (59).

A consistent pattern of layer-wise functional specialization also emerges across both benchmarks. The first layer, responsible for lower-level features, consistently exhibits greater variability across all $\alpha$-parameters. Conversely, the second layer, which learns more abstract representations, is modulated to adapt more rapidly (higher $\alpha_{SM}$ values) and makes greater use of consolidated knowledge (higher $\alpha_{WC}$ values). This suggests a learned policy where higher-level, semantic features are quickly repurposed for new tasks, while lower-level feature extractors undergo more significant, but carefully controlled, modifications.

In summary, our analysis reveals a principle in the learned modulation strategy: the first layer exhibits high variance, reflecting task-specific adaptation, while the second layer converges to a more stable regime, focusing on the reuse of abstract knowledge. The consistently lower variance in the deeper layer suggests a direction for scaling NeuroSync. Employing per-layer (rather than per-neuron) modulation coefficients, particularly for deeper layers of a network, could offer a more parameter-efficient approach with minimal expected performance loss, enhancing the method's applicability to very large models. IFurthermore, the analysis of the $\alpha$-parameter dynamics shows that NeuroSync does not employ a single, fixed strategy, but rather learns an adaptive modulation that is sensitive to both the task structure and the hierarchical layer of the neurons it is controlling.

### H.2  TRAINING STABILITY

To assess the stability of our method, we performed grid search over several key hyperparameters, learning rate, optimizer choice, and the exponential moving average parameter $\beta$. Except for $\beta$ (which we discuss in detail in Section G.3), different configurations of the other hyperparameters resulted in stable training behavior across tasks, and we simply selected the configuration that achieved the highest accuracy.

Additionally, we did not perform extensive tuning of other architectural parameters such as the embedding dimension of the transformer or the number of learnable features per neuron (which we selected to ensure that the NeuroSync module constitutes only a small fraction of the total parameter count) . Despite this, the model achieved strong performance across benchmarks, suggesting a degree of robustness to these hyperparameters as well.

To further examine training stability and convergence behavior, we tracked several metrics for both our method and the continual learning baseline, separately for each layer of a simple two-layer MLP, including:

- The $\ell_2$ norm of weights,
- The effective rank of the MainNetwork 's weight matrices (S-rank) (37),
- Feature rank (47) of the MainNetwork,
- The fraction of dormant neurons in the MainNetwork according to (72).

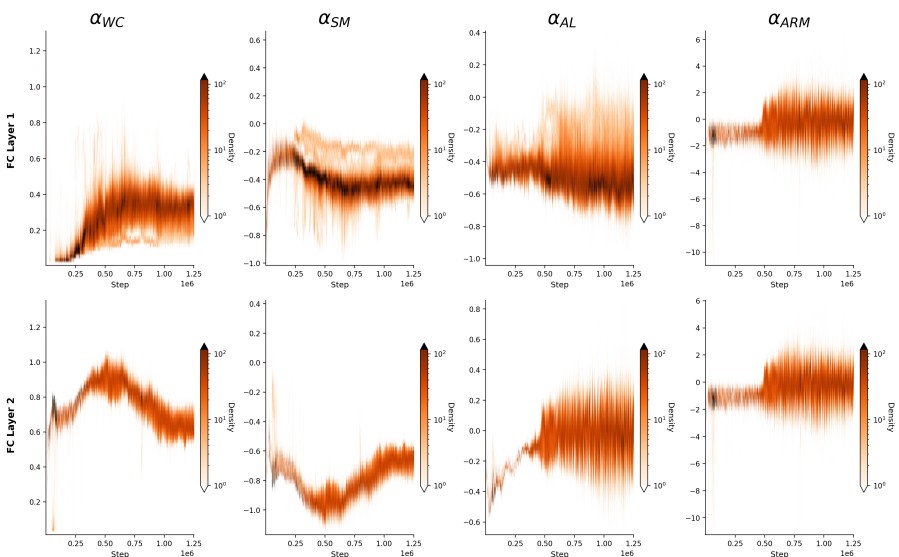

Figure 9: Heatmaps illustrating the evolution of $\alpha$ parameters in a continual learning setting on the `Shuffle CIFAR10` benchmark. The x-axis represents training steps, and the y-axis denotes the values of the $\alpha$ parameters.

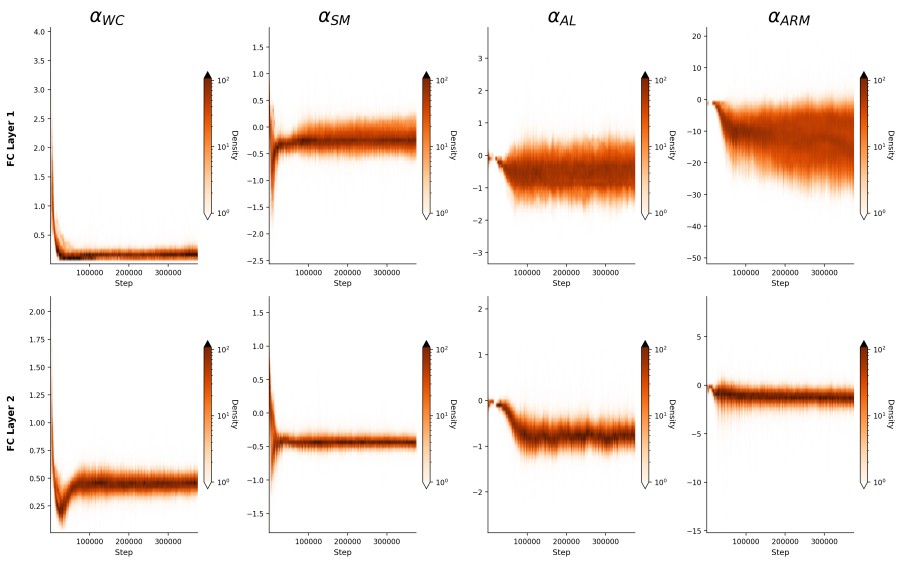

Figure 10: Heatmaps illustrating the evolution of $\alpha$ parameters in a continual learning setting on the `Random-label CIFAR10` benchmark. The x-axis represents training steps, and the y-axis denotes the values of the $\alpha$ parameters.

Our analysis, based on measurements from the Shuffle CIFAR-10 and Random-label MNIST benchmarks, reveals the following.

**Sustained Learning Activity.** As illustrated in Figure 11, the $\ell_2$ norm of the MainNetwork's weights exhibits continual growth under NeuMoSync's control. In contrast, the baseline's weight norms quickly saturate and plateau. This divergence indicates that the MainNetwork remains dynamic and continues to incorporate new information from incoming tasks, whereas the baseline network enters a state of effective learning stagnation.

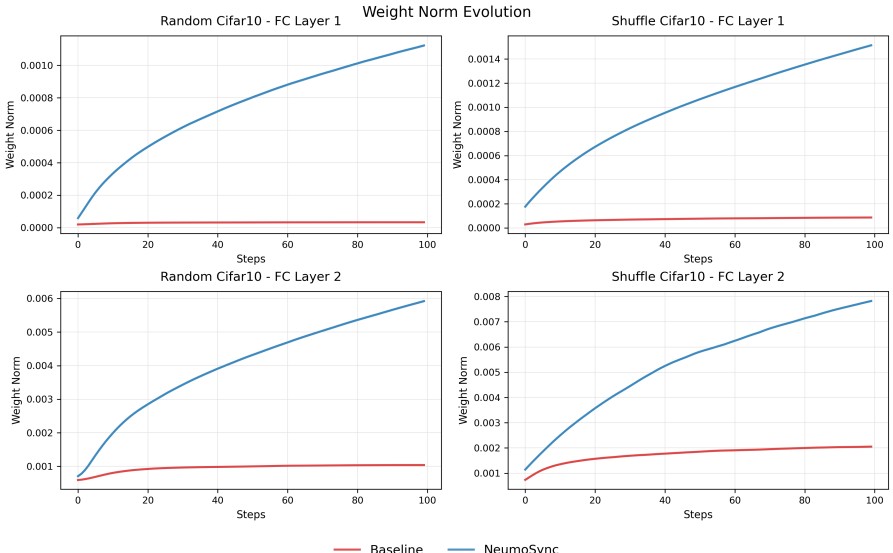

Figure 11: The $\ell_2$ norm of the weights across two layers and two benchmarks is shown. As training progresses, the weight norm of MainNetwork continues to grow, whereas the baseline remains on a plateau, suggesting a loss of adaptability.

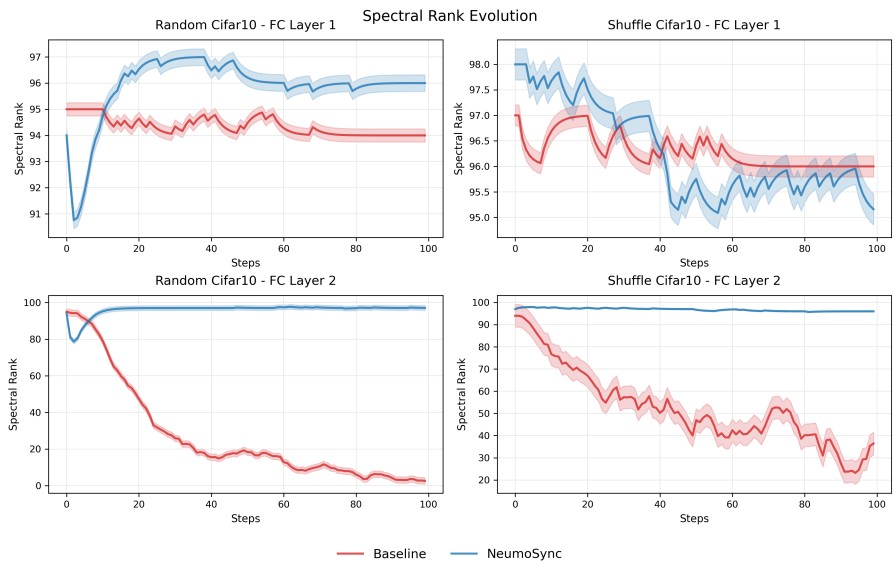

Figure 12: Spectral rank of weights across two fully connected layers and two benchmarks is shown. The figure demonstrates that NeuMoSync maintains a high and stable spectral rank, particularly in the second layer, indicating diverse and non-degenerate representations. In contrast, the baseline model shows a sharp decline in spectral rank, especially in the second layer, suggesting a progressive loss of representational capacity and adaptability during training.

**Preservation of Representational Richness.** The stability of NeuMoSync is further evidenced by its ability to maintain high-dimensional, non-degenerate representations. As shown in Figures 12 and 13, both the S-rank of the weights and the rank of the features remain consistently high throughout training. Conversely, the baseline suffers a progressive decline in both metrics, particularly in the second layer, a clear sign of representational collapse, where the network loses its capacity to learn diverse features.

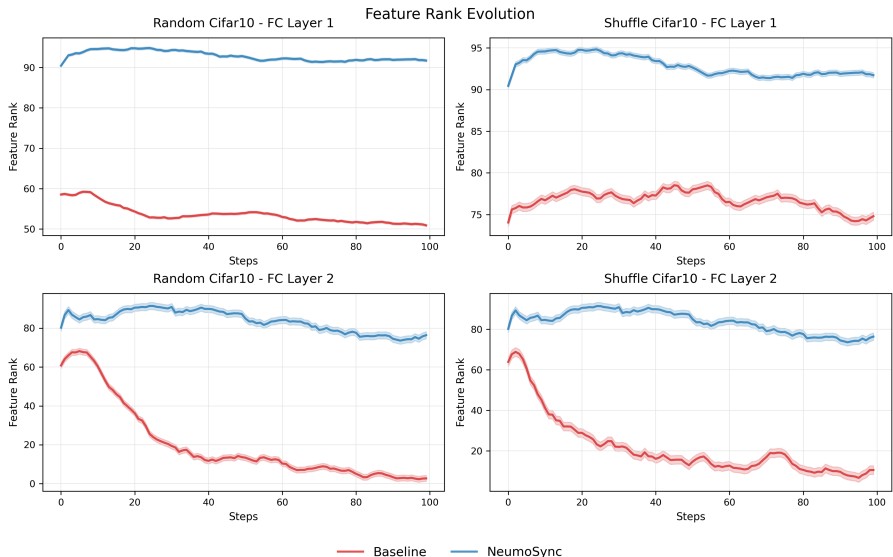

Figure 13: Feature rank across two fully connected layers and two benchmarks. NeuMoSync maintains a consistently high feature rank throughout training, reflecting its ability to represent rich and complex features across tasks. In contrast, the baseline model shows a steady decline in feature rank, most prominently in the second layer, indicating a progressive loss of representational capacity and diversity.

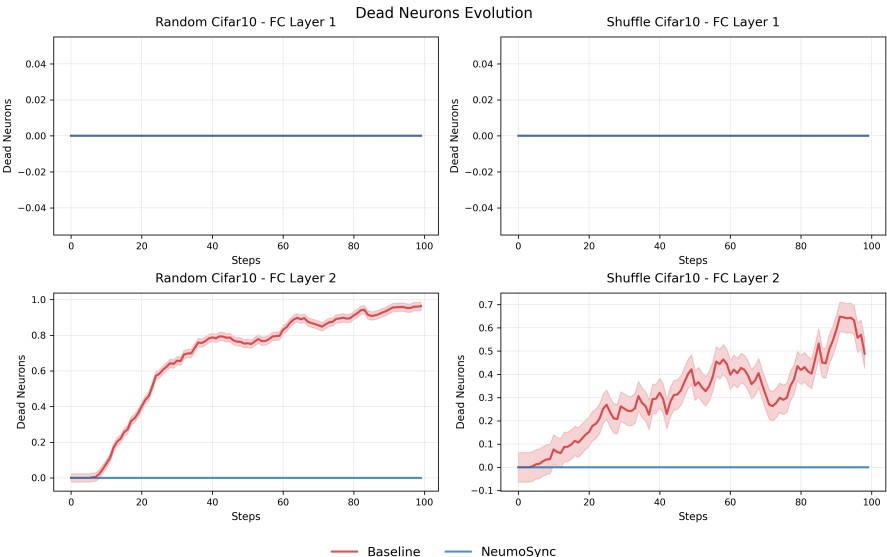

Figure 14: Dead neuron ratio across two fully connected layers and two benchmarks. NeuMoSync consistently maintains a near-zero ratio of dead neurons in both layers, preventing saturation and ensuring continued plasticity. In contrast, the baseline model exhibits a steadily increasing fraction of dead neurons, particularly in the second layer, ultimately leading to widespread neuron inactivity and loss of adaptability.

**Mitigation of Neuron Dormancy.** A primary contributor to plasticity loss is neuron saturation. Figure 14 demonstrates that NeuMoSync effectively mitigates this issue, maintaining a consistently low fraction of dormant neurons. The unmodulated baseline, however, exhibits a classic failure mode: a monotonic increase in dormant units across tasks, a clear manifestation of the "dormant neuron problem" (72), which is known to severely impede plasticity.

Together, these observations demonstrate that NeuMoSync maintains stable learning dynamics and avoids common pitfalls such as neuron saturation, degenerate features, or unstable gradient behavior, further confirming the robustness of our approach under varied training conditions.

# I  Modulation Mechanisms

In this section, we give additional details on the four sets of $\alpha$-parameters generated by the NeuroSync module. For each set of $\alpha$-parameters, we briefly highlight its resemblance to biological neural mechanisms from which our design is loosely inspired. These similarities are intended to be high-level and qualitative rather than precise mechanistic correspondences.

## I.1  Weight Consolidation ($\alpha_{\text{WC}}$)

In our framework, $\alpha_{\text{wc}}^i$ determines the extent to which the $i$-th neuron's consolidated input weights contribute to the InferenceNetwork. Formally, for neuron $i$, we modulate its corresponding consolidation weights through the following equation: $\phi_{\text{Modulated}}^i = \alpha_{\text{WC}}^i \cdot \phi^i$

During inference, this modulation acts through synaptic scaling: a higher value of $\alpha_{\text{WC}}^i$ increases reliance on previously acquired knowledge, promoting stability and enabling positive forward transfer. Surprisingly, our ablation study reveals that beyond facilitating positive forward transfer, this mechanism also plays a central role in preventing negative forward transfer and preserving plasticity (see Section 3.5).

The design of $\alpha_{\text{WC}}^i$ is inspired, at a high level of abstraction, by synaptic consolidation processes in the brain, particularly those observed in the hippocampus and neocortex (25). These biological mechanisms stabilize important synaptic connections over time, shielding them from disruption by future learning. As we will show in Section3.6, the InferenceNetwork progressively shifts its reliance toward the consolidated pathway as it accumulates more generalizable representations, enhancing robustness against plasticity loss.

## I.2  Synaptic Modulation ($\alpha_{\text{SM}}$)

In biological neural systems, synaptic modulation refers to the dynamic regulation of synaptic strength, often governed by neuromodulators such as dopamine (53). These neuromodulators influence how effectively a presynaptic neuron's activity impacts the postsynaptic neuron, thereby altering the neuron's excitability and plasticity (53). According to the Hebbian learning principle (commonly phrased as "cells that fire together wire together") such modulation plays a critical role in enhancing synaptic plasticity, enabling faster learning and adaptation to new stimuli (19). Prior works, inspired by these observations, has proposed new learning rules that computationally model such modulation as multiplicative gain changes (67).

Abstractly inspired by this observation, we introduce a synaptic modulation parameter, $\alpha_{\text{SM}}^i$, for each neuron $i$ in the MainNetwork, which scales the neuron's input synaptic efficacy in the InferenceNetwork. This coefficient enables input-dependent amplification or attenuation of a neuron's contribution within the InferenceNetwork. When combined with the weight consolidation mechanism $\alpha_{\text{WC}}^i$, which regulates reliance on long-term memory, the model achieves a flexible trade-off between acquiring new knowledge and exploiting previously learned knowledge. The resulting InferenceNetwork ' weights for neuron $i$ is computed as:

$$w^i = \alpha_{\text{SM}}^i \cdot \theta^i + \phi_{\text{Modulated}}^i \tag{10}$$

Where $\theta^i$ denotes the current input weight vector of neuron $i$ in the MainNetwork and $w^i$ is the input weights of neuron $i$ in the InferenceNetwork which is the effective weight used during inference, as it is shown in Figure 1(C). This formulation allows the model to dynamically adapt its functional connectivity in a manner loosely reminiscent of neuromodulatory control in the brain (19). An illustration of this part can be seen in Figure 1(C).

During implementation, rather than outputting $\alpha_{\text{SM}}^i$ directly, the controller outputs $\Delta \alpha_{\text{SM}}^i$, where

$$\alpha_{\text{SM}}^i = 1 + \Delta \alpha_{\text{SM}}^i, \tag{11}$$

since initially zero-centered $\alpha_{\mathrm{SM}}^i$ can zero out neuron activations and thus their gradients. In our implementation, we opt to output $\Delta\alpha_{\mathrm{SM}}^i$ rather than $\alpha_{\mathrm{SM}}^i$, but for simplicity, throughout our paper we explain NeuMoSync using $\alpha_{\mathrm{SM}}^i$, where we absorb the $+1$ offset as a constant value that does not affect the functionality of this modulation.

### I.3 ADAPTIVE LINEARITY ($\alpha_{\mathrm{AL}}$)

Recent studies have shown that using a parametrized ReLU (PReLU) (29) per neuron is equivalent to introducing an adaptive residual connection from the input to the output of the activation function, which helps mitigate plasticity loss in neural networks (59). Consequently, in our architecture, we adopt PReLU activations for neurons in the MainNetwork. In contrast to prior work (59), where PReLU parameters are learned for each neuron but fixed across inputs, our framework generates these parameters dynamically through the NeuroSync module. As a result, the activation functions are defined as:

$$o^i = f_{\alpha_{\mathrm{AL}}^i}(x^i) = \begin{cases} x^i, & x^i \geq 0, \\ \alpha_{\mathrm{AL}}^i\, x^i, & x^i < 0, \end{cases} \quad \text{with} \quad x^i = w^i \cdot z^i \tag{12}$$

Here, $f_{\alpha_{\mathrm{AL}}^i}(x^i)$ and $o^i$ denote the activation output of neuron $i$ parameterized by $\alpha_{\mathrm{AL}}^i$, where $x^i$ is the weighted sum of inputs to the neuron, $z^i$ is the output of the previous layer to neuron $i$, and $\alpha_{\mathrm{AL}}^i$ is the slope for the negative input region of PReLU generated by the NeuroSync module.

Interestingly, this dynamic modulation of the negative input region allows the model to adjust a neuron's response even in its inactive regime. If we interpret the negative response region of a PReLU neuron as loosely analogous, in a qualitative and high-level sense, to subthreshold activity in biological neurons, then this mechanism can be viewed as a simple abstraction of how such activity might be modulated. In this way, the parameter $\alpha_{\mathrm{AL}}^i$ governs this aspect of the neuron's behavior. Notably, substantial evidence from neuroscience shows that neuromodulatory signals can strongly influence subthreshold activity in the brain (32).

### I.4 ADDITIVE REGULATION MODULATION ($\alpha_{\mathrm{ARM}}$)

The final mechanism we incorporate is activation offset modulation via $\alpha_{\mathrm{ARM}}$. Specifically, we introduce a deterministic, neuron-specific parameter $\alpha_{\mathrm{ARM}}^i$ that additively modulates the output of neuron $i$. As a result, the final output of neuron $i$, denoted as $g^i(x)$, is defined as:

$$g^i(x) = o^i + \alpha_{\mathrm{ARM}}^i \tag{13}$$

where $o^i$ is the activation output of neuron $i$ as defined in Equation 12, and $\alpha_{\mathrm{ARM}}^i$ serves as an additive modulation term. An illustration of this modulation process at the activation level is shown in Figure 1(D).

This additive modulation can also be loosely related to ideas from neuroscience. In biological neurons, phasic depolarization[3] driven by presynaptic input can coexist with slower, tonic components of activity that reflect the broader state of the brain and are influenced by global neuromodulatory systems (55).In our model, the additive modulation term plays a loosely analogous role at an abstract, high level: it contributes activity that is not directly tied to the neuron's synaptic input and can be seen as a simple proxy for how global signals or network context might influence individual unit outputs. We stress that this connection is qualitative and illustrative rather than a detailed biological account.

### I.5 NEUROSCIENCE INTERPRETATION, RESEMBLANCES AND DIFFERENCES

While NeuMoSync is an engineering solution, its design is loosely inspired, at a very high level and without aiming to reproduce exact mechanisms, by principles of neuromodulation and memory consolidation in the brain. In what follows, we outline a few speculative analogies and clarify the scope of our neuro-inspired claims.

---

[3]Depolarization refers to the change in a neuron's membrane potential, making it more positive and likely to fire an action potential.

**Conceptual Parallels to Memory Consolidation.** The interaction between the MainNetwork and the ConsolidatedNetwork can be interpreted through two complementary neuroscience perspectives.

- **Consolidation at the Neuronal Level:** In the brain, important neural representations can become "consolidated," making them more stable and less plastic over time (74). Our architecture loosely mirrors this concept from a very high-level perspective: when the NeuroSync module assigns a high $\alpha_{\text{WC}}$ and a low $\alpha_{\text{SM}}$ to a neuron, it effectively "consolidates" it, causing the neuron in the InferenceNetwork to rely heavily on its stable trace in the ConsolidatedNetwork while reducing its plasticity to new updates.

- **Consolidation at the Pathway Level:** From a systems-level perspective, neural pathways strengthen with repeated use, forming reliable circuits for automatic processing, akin to System 1 thinking or the default mode network (20; 81). The ConsolidatedNetwork is speculatively analogous to these well-established pathways. As we demonstrate in Section 3.6, after the model has acquired sufficient generalizable knowledge, the NeuroSync module learns to increasingly rely on this stable network, effectively leveraging these "stronger pathways" to guide behavior.

**Scope and Limitations.** It is crucial to be precise about the nature of our claims. We do not claim that NeuMoSync is a neuroplausible model of the brain. Rather, it is a functionally inspired, high-level architecture that abstracts the some principle of global, state- and input-dependent mechanisms regulating local neuronal activity and plasticity (41). This places our work within a growing body of literature that draws on functional insights from neuroscience, such as the concepts of complementary learning systems and fast vs. slow learning.

It is worth mentioning that, although neuromodulatory systems are often described as global, their effects are not uniform at the level of individual neurons. Different neurons within a subpopulation can respond in distinct ways to the same neuromodulatory signal (57), due to factors such as receptor subtype (which can render the effect effectively excitatory or inhibitory), receptor density, and spatial location, which interacts with the diffusive properties of neuromodulators. These determinants are inherently local and neuron-dependent. Since our neuron model is intentionally kept simple, following an integrate-and-fire–like abstraction commonly used in deep learning, we do not attempt to model these biophysical details explicitly. Instead, we absorb this complexity into a global modulatory module that is allowed to assign fine-grained, neuron-specific coefficients, thereby maintaining a standard neuron model while still capturing heterogeneous modulatory effects at the population level.

We also acknowledge key differences between our model and its biological counterparts. For instance, as discussed before, biological neuromodulation often targets large populations of neurons in a uniform way, whereas our model employs fine-grained, per-neuron control. This and other limitations are discussed in our main Discussion (Section 4) and represent important avenues for future research.

## J  ALGORITHM

In this section, we provide a more formal and metathetical formulation of the algorithm and information flow in NeuMoSync. In the following, $f(\cdot)$ denotes a general function, with its parameters indicated by the subscript. For example, $f_\Omega(h)$ represents a function that processes input $h$ and is parameterized by $\Omega$. Additionally, the NeuroSync module is parameterized by $\Omega$, and the decoder responsible for producing the $\alpha$-parameters is denoted by $\Gamma$.

As it is shown in the following pseudocode, `Patch_and_Encode` routine first divides the input image into patches and encodes them using a single-layer CNN, resulting in a set of patch embeddings $\{\mathcal{I}_j\}_{j=1}^{K}$. These embeddings, along with the neuron features, are processed by the NeuroSync module to generate neuron-specific $\alpha$-parameters. Using these modulation signals, the InferenceNetwork is constructed on a per-neuron basis. This network is then responsible for performing the prediction task, while its neurons are further modulated through activation-level mechanisms. The resulting loss is used to update all learnable parameters in the system end-to-end via backpropagation. Finally, ConsolidatedNetwork will be updated through EMA update rule.

---

**Algorithm 1:** NeuMoSync Training Procedure

---

**Input:** MainNetwork parameters $\theta$, neuron features $\{h^i\}_{i=1}^N$;
ConsolidatedNetwork $\phi$, NeuroSync module $\Omega$;
consolidation rate $\beta$; training steps $T$, total neurons $N$; number of patches $K$;
training dataset $\mathcal{D} = \{(x_t, y_t)\}_{t=1}^T$

1   $\phi \leftarrow \theta$
2   **for** $t = 1$ **to** $T$ **do**
3     $\{\mathcal{I}_j\}_{j=1}^K \leftarrow \texttt{Patch\_and\_Encode}(x_t)$
4     $\{\alpha_{WC}^i\}_{i=1}^N, \{\alpha_{SM}^i\}_{i=1}^N, \{\alpha_{AL}^i\}_{i=1}^N, \{\alpha_{ARM}^i\}_{i=1}^N \leftarrow f_\Omega\big(\{h^i\}_{i=1}^N, \{\mathcal{I}_j\}_{j=1}^K\big)$
5     **for** $i = 1$ **to** $N$ **do**
6       $w^i \leftarrow \alpha_{SM}^i \cdot \theta^i + \alpha_{WC}^i \cdot \phi^i$ // Construct InferenceNetwork
7     $\hat{y}_t \leftarrow f_{\text{inf}}\big(x_t; \{w^i\}_{i=1}^N, \{\alpha_{AL}^i\}_{i=1}^N, \{\alpha_{ARM}^i\}_{i=1}^N\big)$ // InferenceNetwork prediction
8     $\mathcal{L} \leftarrow \text{CrossEntropy}(\hat{y}_t, y_t)$
9     Update $\theta$, $\Omega$ and neurons' learnable features using backpropagation on loss $\mathcal{L}$
10    Update neurons' activation statistics in $\{h^i\}_{i=1}^N$
11    $\phi \leftarrow \beta \cdot \phi + (1 - \beta) \cdot \theta$ // Update ConsolidatedNetwork

---

# K   SCALING NeuMoSync

## K.1   ENCODER-DECODER-BASED CONTROLLER ARCHITECTURE

Up to Section 2, we have used either an encoder-only Transformer or a stack of 1D convolutional layers, both of which operate over all neurons to produce the modulatory signals. The Transformer architecture incurs a quadratic computational cost in the number of neurons, while the 1D CNN variant still yields a parameter count that grows with the classifier size due to the final MLP-based fusion layer. In this section, we introduce a controller variant that scales more favorably with the number of neurons and is effectively agnostic to the size of the classifier in terms of parameter growth (Figure 17).

The controller follows an encoder–decoder design. Rather than feeding the full set of neuron feature vectors into the encoder, we uniformly sample a small subset of neurons for each input instance. Concretely, we select $K$ percentage of neurons across all layers of the classifier (with $K \ll 1$) and pass their feature vectors to the encoder. The encoder has the same configuration as in our earlier experiments: a single-layer self-attention Transformer with two attention heads that produces updated embeddings for these sampled neurons.

To obtain modulation parameters for all neurons, we then apply a lightweight decoder layer. We construct query vectors for all $N$ neurons from their learnable features and positional encodings, and use a cross-attention mechanism in which these $N$ queries attend to the $K \times N$ encoded neuron embeddings. The resulting $N$ output vectors are passed through a small two-layer MLP that produces the four modulation coefficients $\{\alpha_{\text{SM}}, \alpha_{\text{WC}}, \alpha_{\text{AL}}, \alpha_{\text{ARM}}\}$ for each neuron.

While this structure is reminiscent of latent-variable architectures (34), we do not introduce an explicit low-dimensional latent bottleneck. Instead, the encoder acts as a shallow self-attention layer that directly defines embeddings for the sampled neurons, enabling fully parallelizable processing. The rest of the controller is deliberately lightweight, consisting of a single cross-attention block and a compact MLP.

## K.2   EXTENDING TO RESNET

In this part, we use a ResNet-18 model as the MainNetwork and evaluate this architecture on the Shuffle Mini-ImageNet benchmark, which features higher-resolution inputs ($84 \times 84$), approximately 40,000 samples, and 100 classes. In our experiments, this controller accounts for only about 0.002% of the total learnable parameters of the full network. Due to the high capacity of the classifier network, we observed that with a sufficiently large training budget almost all methods converge to very high accuracy. To make the setting more challenging, we therefore reduce the

training budget to four epochs per task in order to evaluate performance under a fast-adaptation regime.

We perform an ablation study on the choice of $K$, the fraction of neurons selected for modulation, considering $K \in \{10\%, 5\%, 1\%, 0\%\}$ of the total neurons, shown in Figure 15. The results for different values of $K$ show that as the encoder receives a sparser subset of neurons, performance degrades; in particular, when $K = 0$ the model achieves the lowest training accuracy among other values. Nonetheless, even in this extreme case, the method does not exhibit catastrophic failure. It is worth noting that, even when $K = 0$, the controller still has access to neuron-specific learnable features through the cross-attention decoder, despite not receiving any neuron feature vectors in the encoder stage. The corresponding results are shown in Figure 5, reports the case $K = 0.1$, which corresponds to approximately $450$ sampled neurons for ResNet-18.

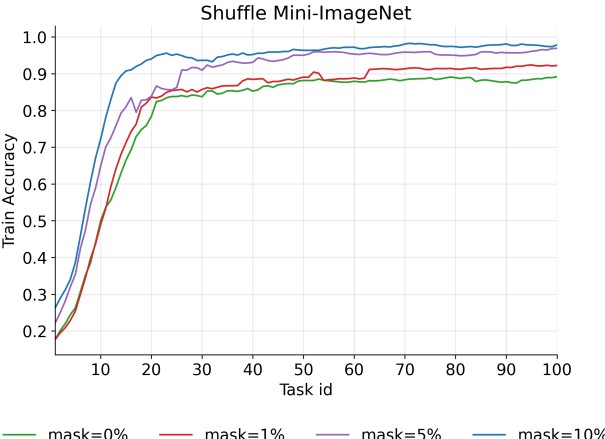

Figure 15: Comparison of training accuracy on Shuffle Mini-ImageNet under different sampeling ratios using an encoder–decoder controller.

We reused the same hyperparameters as in the `Shuffle CIFAR10` benchmark, due to the similar nature of the underlying non-stationarity. The only method-specific hyperparameters we tuned were the learning rate and optimizer, selected from lr $\in \{0.01, 0.001\}$ and optimizer $\in \{\text{adam}, \text{sgd}\}$, respectively, using a single seed and the average final-task accuracy as the selection criterion. The results of this hyperparameter tuning is reported in Table 30. In terms of parameter and runtime overhead, we evaluated the network size, inference runtime, and forward-pass computational cost for each method using a batch size of 16 reported in Table 31.

### K.3  EXTENDING TO TRANSFORMERS

To further validate the scalability and generality of the proposed global controller, we extend our framework to Vision Transformers (17). We investigate three distinct strategies for integrating NeuMoSync into the Transformer architecture. The first strategy restricts modulation to the neurons within the MLP blocks of each encoder layer, following the same protocol used for standard MLPs in Sections 2 and 3.7. The second strategy targets the multi-head self-attention mechanism. Here, we treat each attention head as a functional unit analogous to a single neuron. We apply $\alpha_{\text{SM}}$ and $\alpha_{\text{WC}}$ to dynamically regulate the reliance of each head on its current versus consolidated weights (specifically, the query, key, and value matrices). To support this, we construct a feature vector for each head consisting of a learnable embedding, positional information, and an EMA of the head's activity statistics. The third strategy combines both approaches, applying modulation simultaneously to both the MLP neurons and the attention heads.

For the controller, we utilize the sparse encoder-decoder architecture we described in Appendix K.1. The input to the controller consists of feature vectors from all attention heads alongside a random $10\%$ subsample of the MLP neurons. By doing so, this controller accounts for about $1\%$ of the total weights of the full network. We compare these strategies against a standard `Vanilla ViT` on the `Random-Label CIFAR10` benchmark, where, as previously reported in Appendix 3.5.2, a ViT alone suffers from loss of plasticity.

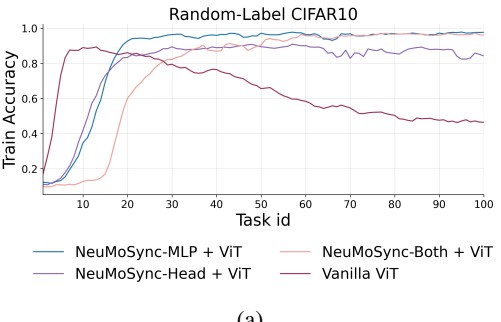 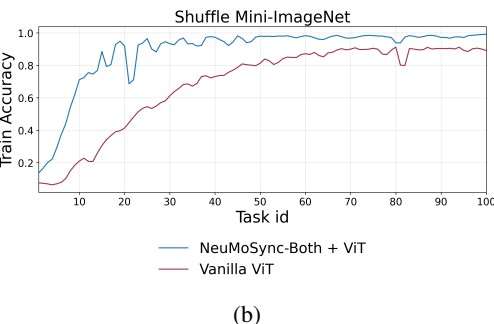

(a)                                            (b)

Figure 16: **(a)** Performance comparison on `Random-Label CIFAR10`. While the `Vanilla ViT` exhibits a sharp loss of plasticity, all NeuMoSync-enhanced variants preserve adaptability. The combined approach (`NeuMoSync-Both`), which modulates both attention heads and MLP blocks, yields the highest performance. **(b)** Evaluation on `Shuffle Mini-ImageNet`. The combined modulation strategy (`NeuMoSync-Both`) outperforms the vanilla baseline, demonstrating scalability to larger architectures and more complex tasks.

Table 30: Methods with corresponding optimizer and learning rate in `Shuffle Mini-ImageNet` benchmark for a ResNet-18 model.

| Method | Optimizer / Learning rate |
| --- | --- |
| Baseline | adam (0.001) |
| CBP | sgd (0.01) |
| CReLU | adam (0.001) |
| NeuMoSync | adam (0.001) |
| PReLU | adam (0.001) |
| ReDo | sgd (0.01) |
| ViT | adam (0.001) |

As shown in Figure 16(a), all NeuMoSync-based variants substantially mitigate the loss of plasticity observed in the baseline. The most effective configuration is the combined strategy (`NeuMoSync-Both + ViT`), which applies four modulation signals to the MLP neurons and two to each attention head; this variant achieves the fastest learning speed and the highest final-task accuracy. These experiments provide evidence that learned global modulatory signals can be effectively integrated into attention-based architectures.

Building on the success of the combined strategy on CIFAR-10, we applied the same configuration (modulating both MLP and attention heads) to a larger ViT backbone on the `Shuffle Mini-ImageNet` benchmark (the controller in this case will also have roughly $0.002\%$ of the total learnable parameters of the entire network). As shown in Figure 16(b), NeuMoSync leads to a clear improvement over the `Vanilla ViT`, achieving higher accuracy and doing so more quickly. These findings underscore the utility of global modulation: the same high-level mechanism scales effectively to different architectures with significantly larger parameter counts, incurring only modest overhead while yielding meaningful performance gains.

## L   A DISCUSSION OF THE RELATIONSHIP BETWEEN $\alpha_{\text{ARM}}$ AND THE TONIC BEHAVIOR OF NEURONS

To more closely examine the behavior of $\alpha_{\text{ARM}}$ and its effect on neuron activity, we consider two tasks: `Random-label CIFAR10` and `Shuffle CIFAR10`. The heatmaps of this modulator for each layer, along with the corresponding neuron activations in the InferenceNetwork, are shown in Figures 18 and 19.

A noticeable pattern, consistent across both benchmarks, is that $\alpha_{\text{ARM}}$ increases at the beginning of each task and then gradually returns to a more stable value as training progresses. This temporal

Table 31: Compute, parameter, memory, and runtime statistics for each method in `Shuffle Mini-ImageNet` benchmark.

| Method | FLOPs | # Params | Memory | Time (ms) |
|---|---|---|---|---|
| NeuMoSync | 2.18 GFLOPs | 11.27 M | 42.98 MB | 201.6 |
| Base | 2.14 GFLOPs | 11.23 M | 42.85 MB | 86.1 |
| PReLU | 2.14 GFLOPs | 22.47 M | 85.73 MB | 87.5 |
| CReLU | 1.86 GFLOPs | 17.73 M | 67.64 MB | 82.5 |
| CBP | 2.14 GFLOPs | 11.23 M | 42.85 MB | 91.5 |
| ReDo | 2.14 GFLOPs | 11.23 M | 42.85 MB | 89.0 |
| ViT | 12.78 GFLOPs | 12.91 M | 49.26 MB | 36.1 |

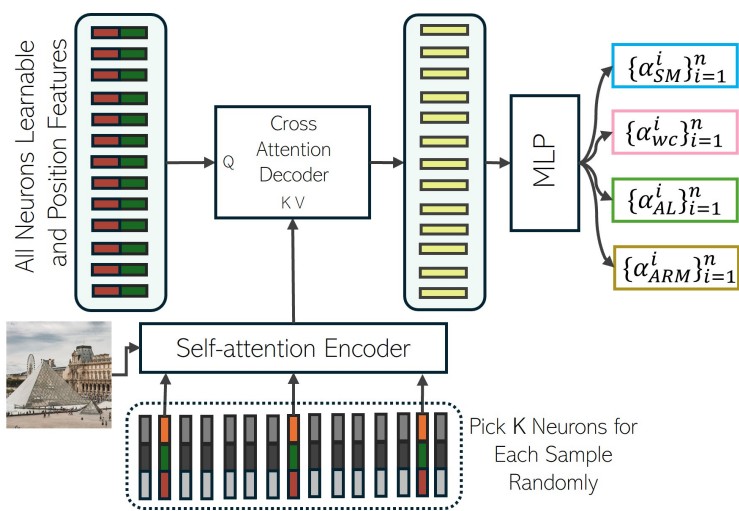

Figure 17: The sparse encoder–decoder architecture for the NeuroSync module used in Resnet-based classifier experiments in Section 3.7

profile bears a qualitative resemblance to the modulatory influence attributed to neuromodulators such as norepinephrine (NE) (65), where transient increases in NE have been reported in response to novel or unexpected stimuli, followed by a decline due to diffusion and regulatory mechanisms. In biological systems, these transient increases are thought to affect the firing threshold of neurons expressing the appropriate receptors (65; 8), thereby changing their tonic firing rate even in the absence of strong sensory input. Interestingly, in our method, we observe a speculative reminiscent pattern: as shown in the activation panels of Figures 18 and 19, neurons exhibit a temporary decrease in activity following the initial rise in $\alpha_{ARM}$. As training proceeds and the task becomes more familiar, $\alpha_{ARM}$ gradually decrease while the activity of FC1 and FC2 gradually increase toward a more stable regime across both benchmarks, presenting an interesting parallel to the novelty-dependent dynamics observed in some neuromodulatory systems.

To further contextualize this observation, it is useful to recall how neuromodulatory signals operate in biological circuits. In biological systems, phasic NE release is often associated with a change in tonic firing in many neurons (89). However, its effects are highly cell-type and receptor-specific and therefore do not uniformly elevate tonic activity (57). In our method, we observe a loosely similar behavior: the values of $\alpha_{ARM}$ vary across and within layers, yet they tend to induce a net positive offset in the state of many neurons.

It is also worth noting that this pattern contrasts with the behavior of $\alpha_{SM}$, which shows a reduction in magnitude at the onset of each task. This decrease, accompanied by a reduction in synaptic plasticity and an effective downscaling of the learning rate, might be in response to unexpected negative outcomes induced by distribution shifts (as reflected in drops in accuracy), as discussed in Section 3.6.

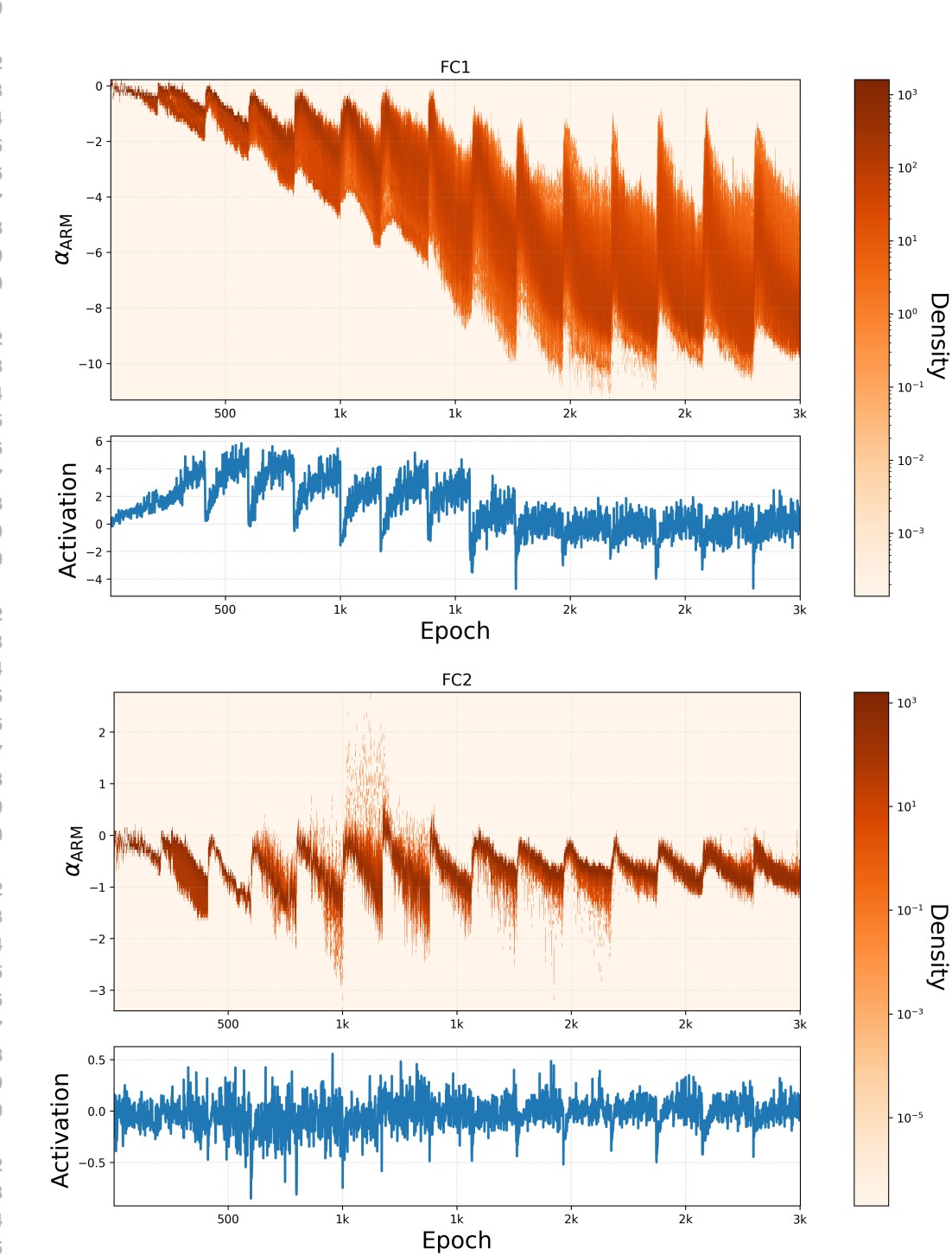

Figure 18: $\alpha_{\mathrm{ARM}}$ and activation of InferenceNetwork in both first and second layer of MLP in `Random-label CIFAR10` benchmark.

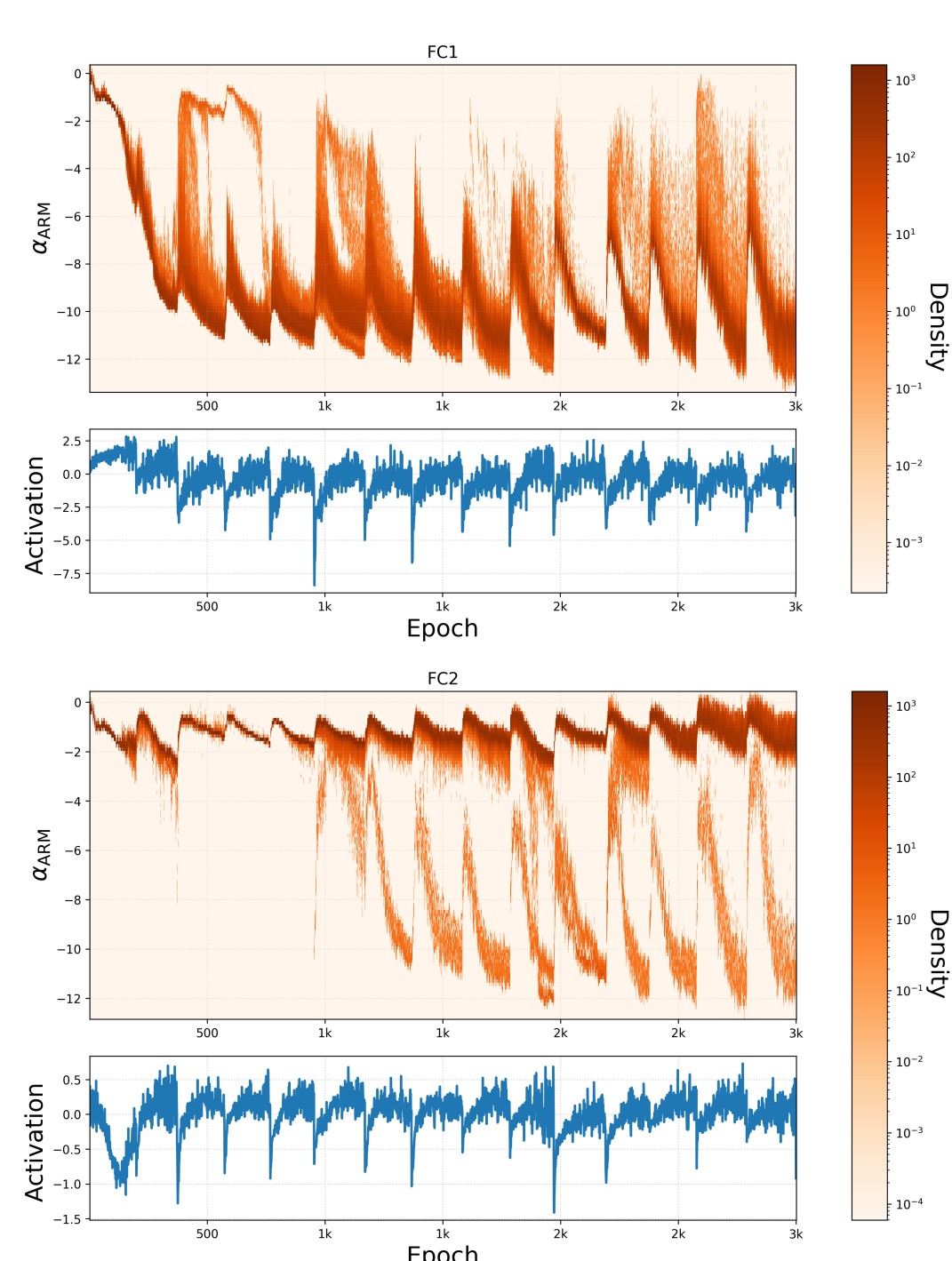

Figure 19: $\alpha_{\mathrm{ARM}}$ and activation of InferenceNetwork in both first and second layer of MLP in Shuffle CIFAR10 benchmark.

## M  COARSE-GRAINED PLASTICITY MODULATION

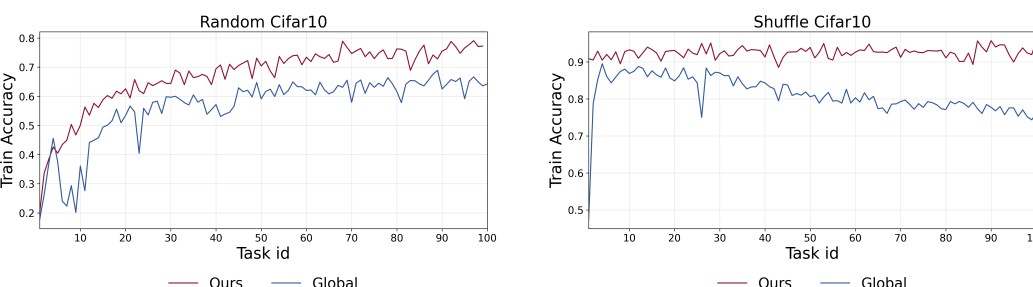

Figure 20: Effect of global (`Global`) versus per-neuron (`Ours`) synaptic modulation signals, $\alpha_{\text{SM}}$ and $\alpha_{\text{WC}}$.

To investigate the effect of using a shared plasticity-modulation signal instead of neuron-specific, fine-grained signals, we replaced the per-neuron plasticity parameters with a single, shared signal for all neurons in the network. Specifically, we applied global $\alpha_{\text{SM}}$ and $\alpha_{\text{WC}}$ parameters across the entire network, while preserving the two neuron-specific activity-modulation parameters ($\alpha_{\text{ARM}}$ and $\alpha_{\text{APM}}$) as in the original setup. This design isolates the impact of introducing global plasticity-modulation parameters. We evaluated this variant on two tasks, `Random-label CIFAR10` and `Shuffle CIFAR10`, with results shown in Figure 20.

The comparison shows that replacing fine-grained signals with a global modulation leads to a decrease in performance, which on `Shuffle CIFAR10` becomes a noticeable drop, particularly on the final tasks. However, performance remains within a relatively acceptable range and does not exhibit a catastrophic collapse. Taken together with the similarity in the modulation trends of neurons within a layer shown in Figures 9 and 10, this ablation suggests a potential path to improve the scalability of our method, especially when combined with the controller introduced in Section 3.7, by moving toward population-level modulatory signals, opening opportunities for future work on designs that are more efficient in both performance and runtime.

## N  EFFECT OF $\alpha_{\text{AL}}$ ON GRADIENT FLOW AND TRAINABILITY

As shown in (59), adaptive linearity (implemented as PReLU when using ReLU as the base activation function) leads to improved gradient flow and, consequently, better trainability, even in continual learning settings. This result is consistent with other studies that aim to embed an approximately linear pathway within deep neural networks (42).

To more directly investigate the effect of this modulatory signal in our method on gradient flow, we consider a variant of our method in which this mechanism is disabled by reverting to standard ReLU activations. We evaluate this variant on two benchmarks, `Random-label CIFAR10` and `Shuffle CIFAR10`, and monitor both the gradient magnitude and the fraction of weights receiving zero gradient in each layer. The results of this ablation study are presented in Figure 21.

The results show that, when $\alpha_{\text{AL}}$ is disabled, the gradient magnitude decreases and the fraction of zero gradients increases across layers, consistently on both benchmarks. In contrast, when this mechanism is enabled, we observe almost no parameters with zero gradient. This finding is also consistent with Section 3.5, where removing this mechanism led to a loss of both plasticity and trainability.

## O  INPUT-CONDITIONED AND INPUT-UNCONDITIONED CONTROLLER

To better assess the impact of conditioning the controller on each input image, we conducted an ablation study in which the controller was conditioned only on neuron features and not on the input. Because the primary motivation of this work is to capture input-driven modulatory effects, this ablation clarifies the practical importance of sample-specific modulation.

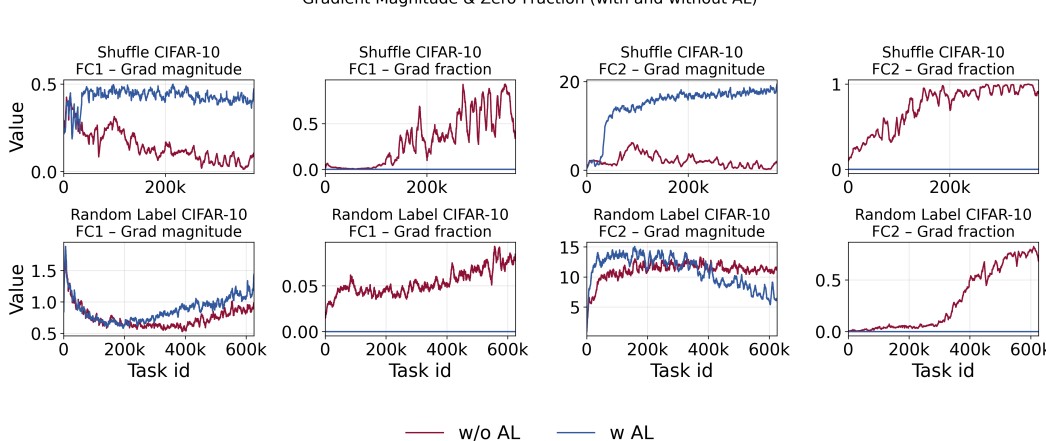

Figure 21: Effect of $\alpha_{\mathrm{AL}}$ modulation on gradients in each layer for two benchmarks, `Random-label CIFAR10` and `Shuffle CIFAR10`. Both the gradient magnitude and the fraction of zero gradients are shown.

To test this, we kept all configurations fixed and evaluated models with and without input-image conditioning on two benchmarks, namely `Random-label CIFAR10` and `Shuffle CIFAR10`. The reported training accuracies were computed using the same random seed for both settings.

As shown in Figure 22, removing the image input to the controller reduces performance on both evaluated benchmarks (`Shuffle CIFAR10` and `Random-label CIFAR10`), with a stronger effect in the memorization setting. On `Random-label CIFAR10`, the input-conditioned controller maintains high training accuracy across tasks, whereas the input-unconditioned variant gradually degrades as new tasks are introduced, indicating a marked loss of plasticity. On `Shuffle CIFAR10`, both variants achieve relatively high accuracy, but a consistent gap remains in favor of the input-conditioned controller. Taken together, these results suggest that conditioning the controller on the input is important for enhancing plasticity by providing a modulatory signal that supports sustained learning performance over long task sequences.

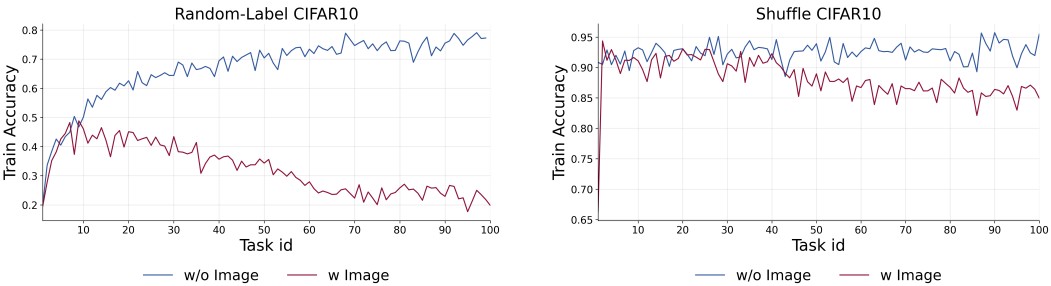

Figure 22: Effect of conditioning the controller on the input image, evaluated on `Random-label CIFAR10` and `Shuffle CIFAR10`. Removing image conditioning (`w/o image`), i.e., using an input-unconditioned controller, reduces performance, with a more pronounced drop on `Random-label CIFAR10`, indicating a severe loss of plasticity.

## P    A DISCUSSION ON THE DYNAMICS OF InferenceNetwork AND ADAPTABILITY

As shown in Section 3, our method strongly preserves plasticity, with particularly large gains on memorization-style tasks. In this section, we analyze the mechanisms underlying this behavior. Specifically, we track the cosine similarity of parameters over time for the three networks, InferenceNetwork,

ConsolidatedNetwork, and MainNetwork, and report the associated modulation coefficients (with emphasis on $\alpha_{\mathrm{SM}}$ and $\alpha_{\mathrm{WC}}$) that determine the final InferenceNetwork. When informative, we also include the $\ell_2$ norms of the relevant weight vectors. Based on these measurements, we propose a hypothesis explaining why our method adapts especially quickly on these tasks. Throughout this analysis, we use `Random-Label CIFAR10` as a representative benchmark.

To adapt to a new task with a shifted data distribution, a classifier must update its weights. Figure 23 visualizes this process by reporting the cosine similarity between the weights of corresponding layers across consecutive tasks (at the end of training in each task) for two methods: `CReLU` (which was the second-best method in this benchmark according to Figure 2) and `Scratch`. The difference is qualitative. For `CReLU`, the first-layer parameters become increasingly similar across tasks as training progresses, whereas training the model from scratch at each task yields consistently low similarity in the first layer. This might indicate that, in addition to preventing loss of plasticity, `CReLU` is also leveraging knowledge transfer across tasks by learning a shared representation in the first layer, while `Scratch` reinitializes parameters and therefore cannot reuse representations. Based on this, we want to investigate wether the final classification network produced by our method, InferenceNetwork, exhibits this transfer behavior as well.

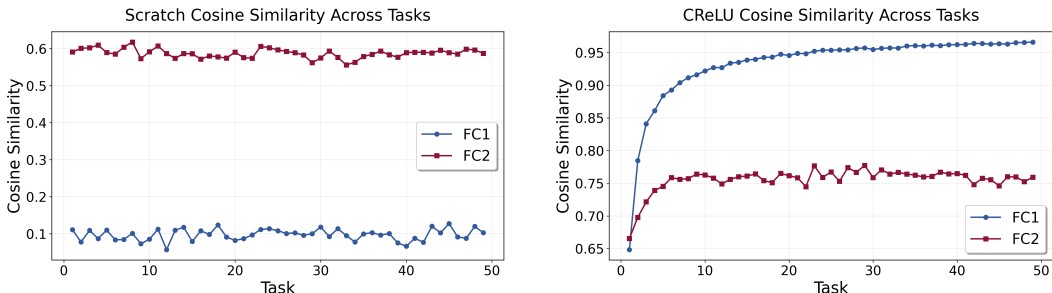

Figure 23: Cosine similarity between weights of the same layers across consecutive tasks, shown separately for `Scratch` and `CReLU`. Both methods exhibit changes in parameter direction, but `Scratch` shows a qualitatively different pattern due to the absence of knowledge transfer.

To test this, we compute the cosine similarity of the layers of the InferenceNetwork across consecutive tasks (Figure 24). We observe a similar pattern as `CReLU`: the first layer becomes more aligned across tasks over time. In contrast, the second layer remains relatively dissimilar, indicating that it continues to reorient to each new task, consistent with the strong task-wise performance. Since the InferenceNetwork is formed by combining multiple components, we next analyze each component to identify which terms drive this behavior.

We first examine the MainNetwork. Surprisingly, the directions of the weights in both of the layers change very little across continual training (Figure 25), which contrasts with `CReLU`, where the second layer of the network shifts direction to accommodate new tasks. Accordingly, since the ConsolidatedNetwork is an EMA of the MainNetwork, its weights should converge to the same directions as the weights of the MainNetwork. This is confirmed in Figure 26: both the across-networks cosine similarity and the $\ell_2$ distance show that MainNetwork and ConsolidatedNetwork converge to highly similar (though not identical) parameters, particularly in the second layer, which is especially interesting given its changing direction in InferenceNetwork. Please note that, even in this case, the ConsolidatedNetwork is still aimed at slowly accumulating general knowledge across tasks (due to the high value of $\beta$); however, because task-specific knowledge remains highly similar from one task to the next in terms of parameter directions, the ConsolidatedNetwork also converges to essentially the same direction.

Since the MainNetwork and consequently the ConsolidatedNetwork have the same direction over consequent tasks (but not in the direction of their corresponding InferenceNetwork), this might suggest that the modulation parameters as likely drivers of adaptation. We therefore group the layer-wise modulation parameters (all $\alpha_{\mathrm{WC}}$ and $\alpha_{\mathrm{SM}}$ values for neurons in a layer averaged over a batch of samples at the end of each task) and compute their cosine similarity across consecutive tasks for each layer, shown in Figure 27. Again, the result is unexpected: in the second layer, the modulation vectors are almost perfectly aligned across tasks (cosine similarity $\approx 1$). At first glance, this seems

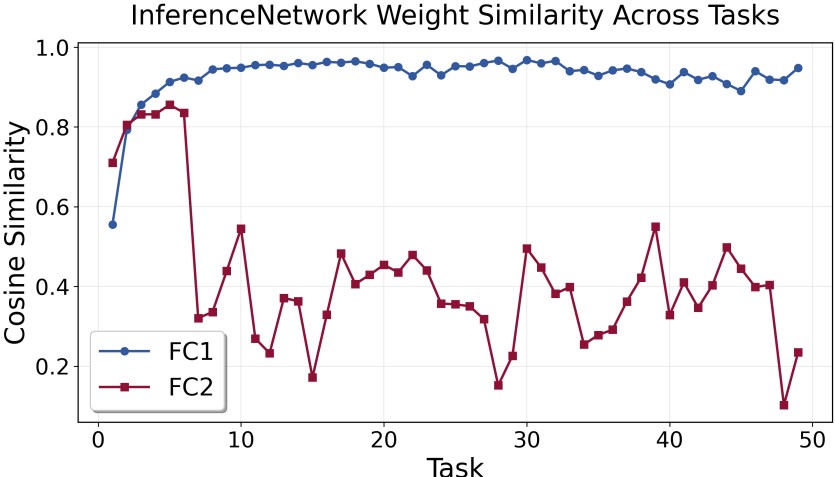

Figure 24: A similar change in parameter direction to that observed in `CReLU` appears in InferenceNetwork: the first layer becomes increasingly similar over training, while the second layer's parameter directions change substantially from one task to the next.

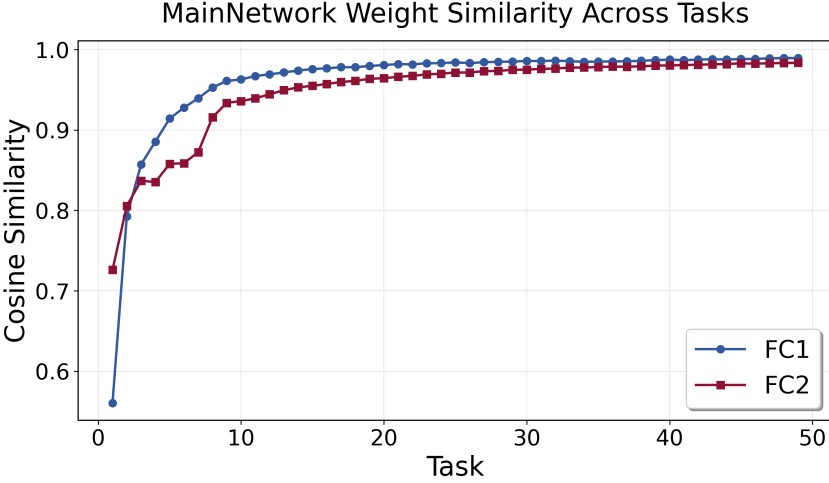

Figure 25: Both layers of MainNetwork become increasingly similar in direction over time, suggesting that the weight directions remain effectively constant across tasks and that the gradients are largely aligned throughout training.

incompatible with the observed task-to-task directional changes in the InferenceNetwork. However, looking only at directions hides an important aspect. The $\ell_2$ norms may capture an important part of the story: Figure 28 shows that the magnitudes of MainNetwork, $\alpha_{\text{WC}}$, and $\alpha_{\text{SM}}$ all change over time, even when their directions remain relatively fixed. This suggests a possible explanation: under certain conditions, changing magnitudes alone can induce large directional changes in the resulting combined weights. We now state a lemma that will help us derive a closed-form expression for the angle between $\theta$ (the weights of the MainNetwork) and $w$ (the weights of the InferenceNetwork), and provide evidence for our hypothesis.

**Lemma 1.** *Let $x \in \mathbb{R}^n$ with $x \neq 0$, and let $y \in \mathbb{R}^n$ be a unit vector such that $x$ is not a scalar multiple of $y$. Let $\gamma \in (0, \pi)$ be the angle between $x$ and $y$, defined by*

$$\cos(\gamma) = \frac{\langle x, y \rangle}{\|x\|}.$$

*Then there exists a unit vector $z \in \mathbb{R}^n$ such that*

$$z \perp y \quad and \quad x = \|x\| \big( \cos(\gamma)\, y + \sin(\gamma)\, z \big).$$

*In particular, $y$ and $z$ form an orthonormal set and $x \in \mathrm{span}\{y, z\}$. The result follows immediately from the Gram-Schmidt process(24).*

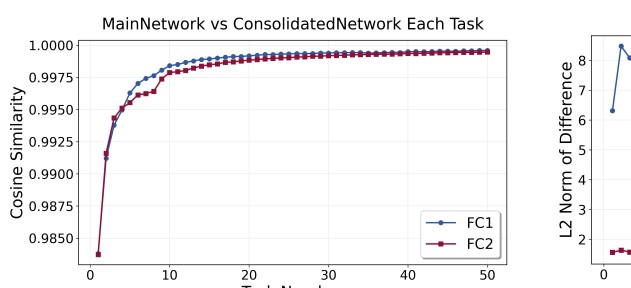 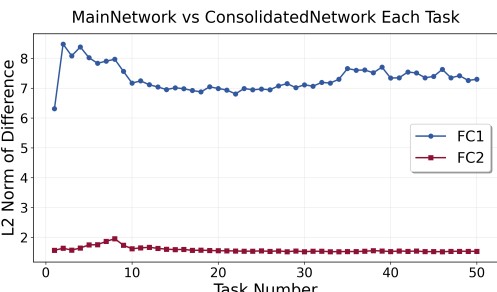

Figure 26: In terms of cosine similarity, the MainNetwork and the ConsolidatedNetwork converge to nearly the same direction. In terms of magnitude, their difference is more pronounced in the first layer, whereas for the second layer (the one we focus on due to its changing direction in InferenceNetwork), the change in magnitude is very close to zero, implying that the two networks are extremely similar in that layer.

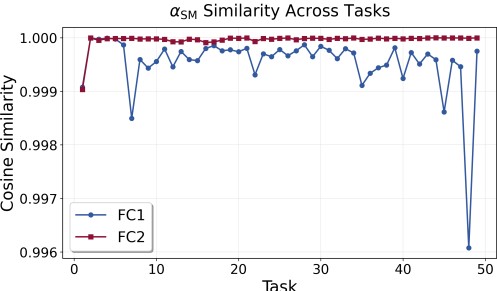 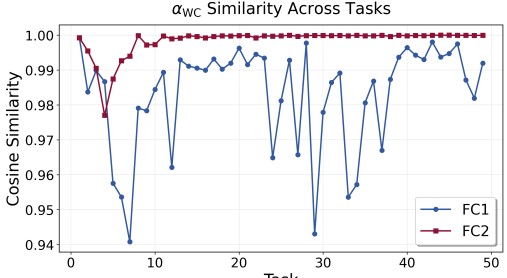

Figure 27: The direction of the vector of $\alpha_{\mathrm{WC}}$ values across neurons in each layer remains almost unchanged (cosine similarity close to 1), indicating that the modulation direction does not vary significantly from one task to the next. This effect is particularly pronounced in the second layer, where the cosine similarity of the modulation vector is very close to 1. Although the directions are very similar across tasks for both the modulation and the constituent networks, their magnitudes change over time, which can translate into changes in the direction of the final InferenceNetwork weight vectors.

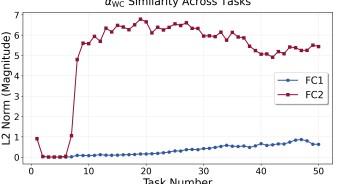 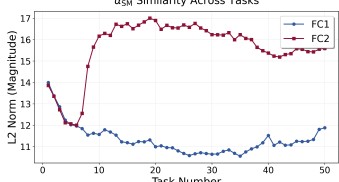 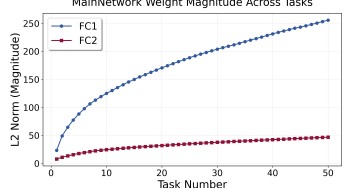

Figure 28: The $\ell_2$ norms of the modulation vectors and MainNetwork change from one task to the next for the second layer. This contrasts with their directions, which remain almost unchanged across tasks.

We now apply Lemma 1 to our setting. Let $\theta^i$ denote the incoming weight vector for neuron $i$ in the MainNetwork, $\phi^i$ denote the corresponding incoming weight vector in the ConsolidatedNetwork, and $w^i$ be the corresponding weight vector in the InferenceNetwork. Let $\hat{v}$ be the unit vector in the direction of $\phi^i$, i.e., $\hat{v} = \phi^i / \|\phi^i\|$.

Empirically, the MainNetwork and the ConsolidatedNetwork are highly similar but not identical (a high cosine similarity which is strictly less than 1 according to Figure 26), so the angle between $\theta^i$ and $\hat{v}$ is nonzero and strictly less than $\pi$, i.e., $\cos(\theta, \hat{v}) \neq \pm 1$. Thus, the condition of Lemma 1 is satisfied with $x = \theta^i$ and $y = \hat{v}$. Consequently, there exist a unit vector $\hat{u}$ orthogonal to $\hat{v}$ and an angle $\delta \in (0, \pi)$ such that

$$\theta^i = \|\theta^i\| \left( \cos(\delta) \, \hat{v} + \sin(\delta) \, \hat{u} \right),$$

where $\hat{u} \perp \hat{v}$ and $\delta$ is the (small) angle between $\theta^i$ and $\phi^i$ (so $\cos(\delta) \approx 1$). Conceptually, $\hat{u}$ might be interpreted as the (small) residual direction influenced by gradient updates, which keeps $\theta^i$ and $\phi^i$ from becoming perfectly aligned (as also reflected in Figure 25, where the MainNetwork changes slightly task to task, i.e., the cosine similarity approaches but never reaches 1).

In our model, the effective incoming weight vector for a neuron in the InferenceNetwork is

$$w^i = \alpha_{\mathrm{WC}}^i \cdot \phi^i + \alpha_{\mathrm{SM}}^i \cdot \theta^i$$

where $\alpha_{\mathrm{WC}}^i$ and $\alpha_{\mathrm{WC}}^i$ are the modulation parameters for the neuron $i$. Then by applying the above Lemma, we can write

$$w^i = \left( \alpha_{\mathrm{WC}}^i \|\phi^i\| + \alpha_{\mathrm{SM}}^i \|\theta^i\| \cos(\delta) \right) \hat{v} + \left( \alpha_{\mathrm{SM}}^i \|\theta^i\| \sin(\delta) \right) \hat{u}.$$

Next, let $\psi$ be the angle between $w$ and $\theta^i$. We have

$$\tan(\psi) = \frac{\alpha_{\mathrm{SM}}^i \|\theta^i\| \sin(\delta)}{\alpha_{\mathrm{WC}}^i \|\phi^i\| + \alpha_{\mathrm{SM}}^i \|\theta^i\| \cos(\delta)}. \tag{14}$$

This expression highlights a regime of high angular sensitivity: when the parallel components nearly cancel (i.e., when $\alpha_{\mathrm{WC}}^i \|\phi^i\| \approx -\alpha_{\mathrm{SM}}^i \|\theta^i\| \cos(\delta)$), the denominator becomes small, and even a small magnitude change in $\alpha_{\mathrm{SM}}^i$ or $\|\theta\|$ can produce a large change in $\psi$, even if directions of $\theta$, $\phi$, and the modulation vectors remain nearly fixed. Importantly, this regime requires $\theta^i$ and $\phi^i$ to be very similar (to enable cancellation), but not perfectly identical (so that $\sin(\delta) \neq 0$ and the orthogonal term can "steer" $w^i$).

Our results are reminiscent of this "cancellation-and-steering" regime. We observe that $\alpha_{\mathrm{SM}}$ and $\alpha_{\mathrm{WC}}$ typically have opposite signs and comparable magnitudes (Section 3.6 and Appendix H.1), and that MainNetwork and ConsolidatedNetwork remain closely aligned during training (Figure 26). Together, these observations make the parallel components of $\alpha_{\mathrm{WC}}^i \phi^i$ and $\alpha_{\mathrm{SM}}^i \theta^i$ prone to cancel.

To further support this picture, Figure 29 shows the $\ell_2$ norms of the two layers of the InferenceNetwork with and without the $\alpha_{\mathrm{WC}}$ term. When $\alpha_{\mathrm{WC}}$ is enabled, the two scaled vectors cancel more strongly, which is reflected in a lower $\ell_2$ norm for the second layer[4] compared to the case without $\alpha_{\mathrm{WC}}$. This implies that the denominator in Eq. 14 has smaller a magnitude and is closer to zero, thereby increasing the sensitivity of the angle $\psi$ to changes in the numerator. Finally, to directly monitor $\psi$ (the angle between InferenceNetwork and ConsolidatedNetwork), we compute the value of Eq. 14 separately for each neuron and then average the values across all neurons in the same layer, using a batch of samples at the end of each task. These results are illustrated in Figure 30. The figure demonstrates that the angle between the MainNetwork and InferenceNetwork parameters ($\psi$) in the second layer shifts from one task to another. This reinforces the finding that proximity to the cancellation-steering regime enables meaningful directional changes in the InferenceNetwork solely by altering the magnitude of the modulation parameters, while the constituting networks (MainNetwork and ConsolidatedNetwork) maintain a very similar directions. This behavior might suggest an interesting hypothesis : the training algorithm and the controller exploit a shortcut available when two nearly similar networks are present, achieving task-specific adaptation primarily by rescaling the controller outputs (since the direction of modulation parameters are the same across tasks according to Figure 27), rather than by relearning an entirely new set of weights for each task.

Overall, this might explain why two very similar networks can still be beneficial. Although the MainNetwork and the ConsolidatedNetwork are very similar, their controlled combination (especially

---

[4]Note that the norm of the denominator in Eq. 14 is approximately equal to the norm of the InferenceNetwork weights, since $\cos(\delta) \approx 1$.

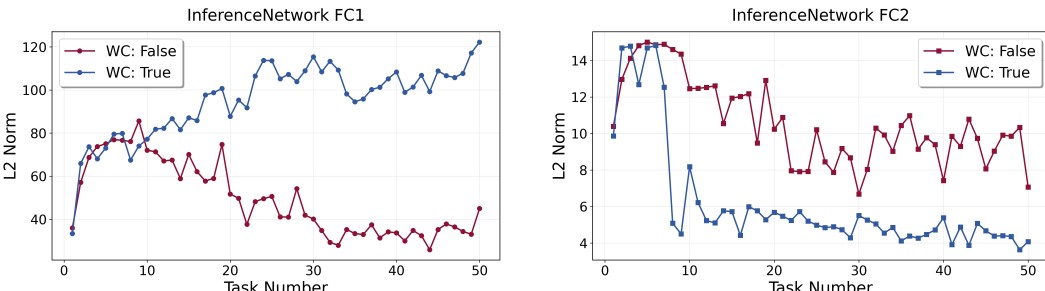

Figure 29: For the second layer, InferenceNetwork exhibits a smaller $\ell_2$ norm when $\alpha_{\mathrm{WC}}$ is enabled than when it is disabled. A lower magnitude implies that the denominator of Equation 14 is closer to the near-cancellation regime, where small changes in the numerator's magnitude can be converted into substantial changes in direction.

with oppositely signed coefficients) creates a sensitive "cancellation-and-steering" regime in which the controller can adapt rapidly through simple scaling, rather than relearning entirely new directions for each task. Thus, even though it may seem counterintuitive, two similar networks can realize solutions that a single network cannot: what appears almost repetitive at the level of weights becomes powerful at the level of how those weights are combined, enabling fast adaptation under specific conditions.

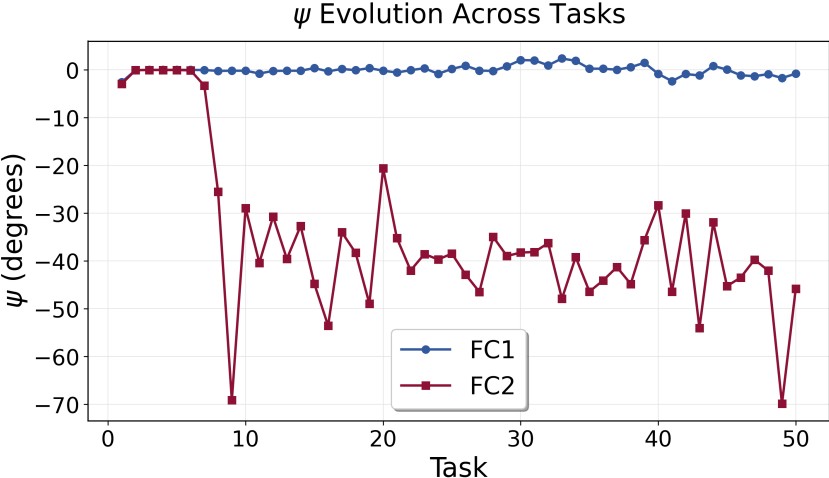

Figure 30: Evolution of the angle between InferenceNetwork and MainNetwork across two layers, measured at the end of each task. While the angle in the first layer remains relatively constant, it shifts significantly between tasks in the second layer, even as MainNetwork and ConsolidatedNetwork maintain a nearly fixed direction.

