# OpenReview forum: "NeuMoSync: End‑to‑End Neuromodulatory Control for Plasticity and Adaptability in Continual Learning"
_ICLR.cc/2026/Conference — Submitted to ICLR 2026_

### Official Review · Reviewer_BWK5 · 2025-10-21

**Soundness:** 3
**Presentation:** 2
**Contribution:** 2
**Rating:** 4
**Confidence:** 3

**Summary:**

The authors propose a method for preserving plasticity when training over successive tasks.

The methods maintains both a normal trained network (MainNetwork), and its slow-changing Exponential Moving Average (EMA - ConsolidatedNetwork).

Furthermore, each neuron maintains a vector of features (position in network, past activation stats, learnable features...)

Then a NeuroSync module takes in the feature vectors of all neurons, together with the current data sample, and outputs 4 modulation coefficients per neuron. 2 of those are used to interpolate between the neuron's weights from fast and slow (EMA) networks, and the other modulate the neuron's output function.

Standard Backpropagation occurs through both Neurosync and MainNetwork, after each training sample/batch.

The model seems much better than multiple alternatives at quickly learning new tasks. This is especially true for tasks that require pure memorization with little shared structure, such as Random-Label MNIST and CIFAR.

Various ablations and experiments attempt to describe the dynamics of the system and explain its performance.

**Strengths:**

The model's performance in quickly learning new tasks, over a succession of tasks, seems much higher than many alternatives, including training from scratch, strong L2 regularization, and various algorithms for maintaining plasticity.

The algorithm itself is clearly explained.

**Weaknesses:**

- The authors make many references to continual learning. However, they implicitly acknowledge that their proposed method itself is not actually capable of continual learning and forgets previous tasks, since they are forced to augment it with Experience Replay in their "Forgetting" experiments. The method seems to improve the capacity of the network to quickly learn new tasks, with little regard to previously learned tasks.

- The authors show many graphs of the system's behavior. Unfortunately some of these graphs seem to contradict each other, making interpretation difficult (see below).

- However, from some of those graphs, a possible explanation for the system's behavior emerges; this explanation is quite different from what the one the authors suggest.

**Questions:**

- Graphs about the alphas produced by the system, as shown, seem contradictory. For the same Random-Labels CIFAR task, Figure 4a shows average alpha_sm as small and slightly positive. Figure 4c shows all alpha_sm as uniformly and strongly negative (<1). Figure 9 shows alpha_sm as moderately negative (~-0.5). Something must be missing from the descriptions.

- Similarly, figure 4b shows that alpha_wc is basically 0 for the first few tasks, before jumping higher. But Figure 4C shows alpha_wc jumping almost immediately to sizeable values. Please clarify.

- Comparison are not helped by the authors constantly changing the x-axis between "epochs", "tasks" and "steps". Some consistent markers for successive tasks would be useful.

- Overall, the graphs suggest that alpha_sm (the dynamic weight on the current network) and alpha_wc (the dynamic weight on the averaged, slow-changing network) are consistently of opposite signs. This remarkable fact is not mentioned by the authors, unless I missed it. If true, it would suggest an immediate explanation: the system simply *subtracts* the accumulated weights from the current, fast-moving network, making the changes "faster" (the specific assignment of which is negative or positive, between alpha_sm and alpha_wc, should make no difference, unless I'm missing something).

- This is particularly relevant for Figure 4b, which suggests that a jump of alpha_wc coincides with and counteracts a dip in performance, presumably caused by loss of plasticity. The authors choose to interpret this as "learned reliance on consolidated knowledge" - which seems counter-intuitive since (as the authors point out) this particular task has no use for past knowledge. Instead, it suggests an increased "active forgetting" of this past knowledge, reducing the burden of accumulated (and now irrelevant) information.

- Please clarify whether the above makes sense. Maybe it doesn't, but there should be at least some mention of the apparent opposite signs between outputted alpha_sm and alpha_wc (assuming it the graphs showing it are the correct ones).

---

> ### Author Response · Authors · 2025-11-22
>
> > The proposed method itself is not actually capable of continual learning and forgets previous tasks ...
>
> This is a valid point and, indeed, our method does not explicitly mitigate forgetting on its own. We clarified our claims in the main paper at the end of Section 3.4.
>
> Nevertheless, in our experiments, we had observed several advantages of our method in regards to stability:
>
> (1) We found a synergistic effect between our proposed method and experience replay, finding that the combination can be better than other continual learning approaches also endowed with ER. We elaborate on this in more details in general response.
>
> (2) We observe forward transfer between tasks, as evidenced by the higher Forward Knowledge Transfer and Forward Learning Curve Area reported in Table 1, indicating that the method is accumulating useful knowledge from previous tasks and not simply maintaining a baseline level of plasticity.
>
> (3) As shown in Table 1, our method also achieves higher Backward Knowledge Transfer and Backward Learning Curve Area compared with other methods, indicating that it can readapt to previously learned tasks more efficiently and that the general knowledge induced by earlier tasks facilitates this re-adaptation.
>
> ## Clarifications on Alpha Values and Plot Axes in Figure 4:
>
> We sincerely thank the reviewer for pointing this out. The original version of this figure contained inconsistencies, which stem from a technical detail in our implementation. In our implementation, instead of multiplying the incoming weights by $\alpha_{\text{SM}}$, we multiply them by $(1 + \alpha_{\text{SM}})$, as this empirically leads to better gradient flow. Since the model already contains multiple modules, we decided not to introduce this extra detail in the main text in order to avoid confusion, as the constant $+1$ offset can, in principle, be absorbed into the learnable parameter. In the revised version, we now explicitly clarify this implementation detail in Appendix I.2.
>
> In the initial submission, Figure 4(c) shows the raw $\alpha_{\text{SM}}$ values, whereas Figure 4(a) shows the magnitude of the actual modulation coefficient, $1 + \alpha_{\text{SM}}$. We originally depicted the magnitude because, in the discussion of the effective learning rate, only the scale of this coefficient matters, not its sign. To make all figures consistent, we have completely updated Figure 4: all subplots now use epochs on the $y$-axis and are aligned with each other, and the same quantity (the true multiplicative coefficient $1 + \alpha_{\text{SM}}$) is used wherever $\alpha_{\text{SM}}$ is reported. Additionally, in the initial submission Figure 4(c) showed a restricted window of $\alpha_{\text{WC}}$ values to highlight the bimodality emphasized in the main text; in the revised version it now contains the full training trajectory for the first 10 tasks. Due to this alignment of the $y$-axes, the apparent discrepancy between Figure 4(b) (showing $\alpha_{\text{WC}}$) and its histogram in Figure 4(c) is also resolved. Finally, the inconsistency between this figure and Figure 9 is resolved, as Figure 9 already showed $1 + \alpha_{\text{SM}}$, which is now used consistently throughout.
>
> ## Response on the Opposite-sign and Dynamics of $\alpha_{SM}$ and $\alpha_{WC}$
>
> We thank the reviewer for highlighting this interesting behavior and for suggesting a possible interpretation. Considering the regime where $\alpha_{\text{WC}}$ is non-zero, our method can be understood as follows. For each neuron $i$, the controller effectively generates two components that determine the weights of the `InferenceNetwork` for each task: (i) how to scale the fixed input weights $\phi^i$ (slow-changing weights from the `ConsolidatedNetwork`) as a function of the input, and (ii) how to scale the learnable weights from the `MainNetwork` through $\alpha_{\text{SM}}^i$. Also the `MainNetwork` weights $\theta^i$ are changing based on gradient descent.
>
> If we absorb $\alpha_{\text{SM}}^i$ into $\theta^i$ for notational simplicity, then $\theta^i$ can be viewed as a task-specific weight vector that is added on top of a (potentially scaled) consolidated weight vector $\phi^i$. Since in practice we almost always have $\alpha_{\text{SM}}^i > 0$, $\theta^i$ plays the role of a residual $\Delta$ that is subtracted from the slow-changing vector $\phi^i$. This residual interpretation is most natural in cases where the nearly constant consolidated weights are beneficial for the task. However, as shown in Figure 9 (FC\_1), this is not always what the controller elects to do: in that example, it effectively suppresses the contribution of the `ConsolidatedNetwork` (with $\alpha_{\text{WC}} \approx 0$) for the first layer, so for that layer the final weights are not constructed from the consolidated weights at all. We also provide a discussion of this observation and the dynamics between $\alpha_{\text{SM}}$ and $\alpha_{\text{WC}}$ in the revised version of Section 3.6.

---

> > ### Author Response · Authors · 2025-11-22
> >
> > > This is particularly relevant for Figure 4b, which suggests that a jump of alpha\_wc coincides with and counteracts a dip in performance, presumably caused by loss of plasticity. The authors choose to interpret this as "learned reliance on consolidated knowledge" - which seems counter-intuitive since (as the authors point out) this particular task has no use for past knowledge. Instead, it suggests an increased "active forgetting" of this past knowledge, reducing the burden of accumulated (and now irrelevant) information.
> >
> > Regarding the terminology, our use of "learning to rely" was intended to describe the empirical observation that, as training progresses, $\alpha_{\text{WC}}$ becomes consistently positive for a noticeable subset of neurons (specifically in the second layer for Random-Label CIFAR-10, as depicted in Figure 9). This signals that the weights of the `ConsolidatedNetwork` are being explicitly incorporated into the `InferenceNetwork`, whereas they were effectively ignored in earlier stages. However, we agree that the reviewer's "active forgetting" interpretation is a plausible explanation for neurons that have $\alpha_{\text{WC}} > 0$. Crucially, because the modulation parameters are generated dynamically per sample, this "active forgetting" is *transient*: the controller selectively suppresses or modifies the consolidated knowledge specifically for the current input, without permanently erasing that knowledge from the slowly evolving `ConsolidatedNetwork`.
> >
> > Moreover, as discussed in Appendix F.2 for the memorization task (`Random-Label CIFAR10`), we argue that using the consolidated weights merely as an initialization is not an effective way to extract common structure, whereas learning a policy that balances the use of previous knowledge and task-specific information is. In our method, the controller does not use the `ConsolidatedNetwork` as a uniform initialization for all neurons; instead, it uses it in a flexible neuron- and input-specific manner, scaling it and potentially combining it with another vector ($\theta^i$). As shown in Figure 4, this combination starts with relatively small task-specific modifications: the effective $\alpha_{\text{SM}}$ has small magnitude at the beginning of each task, so the `ConsolidatedNetwork` dominates. This dynamic implies that the controller is learning to blend two sources of knowledge: a slow, consolidated one and a fast, task-specific one.

---

> > ### Comment · Reviewer_BWK5 · 2025-11-24
> > **Figure 4 still seems contradictory**
> >
> > Unless I'm missing something, it seems that alpha_sm in Figure 4a and Figure 4c differ by a value of roughly -1.0. Please fix this, or explain the difference. (Also specify whether the single-valued alpha_sm in Figure 4a is supposed to be the mean of the values shown for alpha_sm in Figure 4c).

---

> > ### Comment · Reviewer_BWK5 · 2025-11-24
> >
> > - Hopefully you meant "we almost always have alpha_sm < 0", otherwise I'm really confused.
> >
> > - If you actually show 1 + alpha_sm, and the result is negative, does it mean that the actual trained alpha_sm is actually even lower than -1 ? (Please fix the discrepancy between Figure 4a and Figure 4c, which still indicate different values for alpha_sm).
> >
> > (In the following I'm assuming that the actual coefficient 1+alpha_sm is always negative:)
> >
> > - It's not subtracting the fast network from the slow network. It's doing the opposite - subtracting the slow network from the fast one. The slow network stands for accumulated previous knowledge, which is flexibly "removed" by the positive weight alpha_wc (in contrast to the fast network's consistently negative weight 1+alpha_sm).
> >
> > In support of this interpretation (as opposed to seeing the fast network as a corrective residual): In Figure 4b note that the overall accuracy keeps increasing in the period where alpha_wc is effectively 0, and performance seems initially better in the "without wc" curve. This suggests that the fast network is the one doing the actual guess (presumably it's actually guessing the minus of the answer, due to consistently negative 1+alpha_sm from the start).
> >
> > The fast network keeps learning continuously throughout all tasks, but previous task knowledge is irrelevant (and thus potentially harmful). Alpha_wc rises to subtract the slow moving network, which contains more of this accumulated, irrelevant knowledge. This allows the fast network to adapt to the new tasks faster than gradient descent alone would permit.
> >
> > (At least that seems to be the most straightforward interpretation of these results, as shown.)
> >
> > - Please explain what "Neuro sync activation" and "Inference network activation" mean in Figure 4a. What quantities are being plotted? Also, what does the shaded area in Figure 4b indicate?

---

> > > ### Author Response · Authors · 2025-11-27
> > >
> > > ## About the Previous Confusion
> > >
> > > We sincerely appreciate your insightful engagement in the discussion and helpful comments. We also apologize for any additional confusion caused by our earlier response and the partially updated Figure 4. To clarify, we confirm that we indeed meant "we almost always have $\alpha_{\text{SM}} < 0$", and that in the previous version the heatmap of $\alpha_{\text{SM}}$ in Figure 4c was not shifted by +1 to reflect the true modulation gain. The figure has now been corrected, and the current version accurately reflects this modification.
> > >
> > > >  Also specify whether the single-valued alpha_sm in Figure 4a is supposed to be the mean of the values shown for alpha_sm in Figure 4c
> > >
> > > Yes indeed. The $\alpha_{SM}$ reported in Figure 4a has been obtained by averaging the values shown at each step of Figure 4C.
> > >
> > > > Please explain what "Neuro sync activation" and "Inference network activation" mean in Figure 4a. What quantities are being plotted?
> > >
> > > Regarding the "Inference Network Activation" and "Neuro Sync Activation" curves in Figure 4a, we have added a clearer explanation in the main text. In this figure, "Neuro Sync Activation" refers to the average activation of the last layer of the MLP used in the controller, averaged over the samples in a batch. "Inference Network Activation" refers to the mean activation of all neurons in the `InferenceNetwork`, again averaged over a batch of samples.
> > >
> > > >  Also, what does the shaded area in Figure 4b indicate?
> > >
> > > The shaded area marks the specific point when $\alpha_{\text{WC}}$ becomes non-zero, coinciding with a recovery in the accuracy of the model.

---

> > > ### Author Response · Authors · 2025-11-27
> > >
> > > ## A Possible Explanation and Interpretation of NueMoSync
> > >
> > > For explaining why our method outperforms the baselines, particularly on the memorization task, we conducted additional analyses of the underlying dynamics of `MainNetwork`, `ConsolidateNetwork`, and the modulation parameters that together define the `InferenceNetwork`. We present this analysis in detail in Appendix P of the newly revised manuscript, where we use these observations to propose a hypothesis that may account for the large performance gap. Throughout this analysis, we use `Random-Label CIFAR10` as a representative benchmark. Below, we provide a brief summary of our main findings.
> > >
> > > Tracking cosine similarity of the parameter matrices across consecutive tasks (Figure 24), we observe that the `InferenceNetwork`’s first layer becomes increasingly aligned over continual training, while the second layer remains comparatively dissimilar and continues to reorient between tasks. Somewhat surprisingly, the weight directions of the `MainNetwork` change very little across tasks (Figure 25), and since `ConsolidatedNetwork` is an EMA of the `MainNetwork`, the two converge to nearly the same (but not identical) directions (particularly in the second layer - Figure 26). Thus, the substantial task-to-task directional changes observed in the `InferenceNetwork` cannot be attributed to learning new directions in $\theta$ (parameters of `MainNetwork`) or $\phi$ (parameters of `ConsolidatedNetwork`). The direction of modulation parameters ($\alpha_{\texttt{WC}}$ and $\alpha_{\texttt{SM}}$) follows the same story (Figure 27): they are relatively similar from task to task. What we hypothesize in detail in Appendix P is that because $\theta$ and $\phi$ remain highly aligned (but not identical), and because $\alpha_{\texttt{SM}}$ and $\alpha_{\texttt{WC}}$ often have *opposite signs* with comparable magnitudes, the combined terms $\alpha_{\texttt{SM}}\theta$ and $\alpha_{\texttt{WC}}\phi$ can largely cancel in their (shared) dominant direction. This creates a sensitive “cancellation-and-steering” regime (Equation 14)* in which modest task-to-task changes in the controller outputs' magnitude (i.e., rescaling the $\alpha$’s) can induce large changes in the `InferenceNetwork`’s effective weight *direction*, enabling fast adaptation (Figures 29 and 30) without “relearning” new weight directions in the `MainNetwork`.
> > >
> > > In other words, we hypothesize that **learning on the early tasks is done through the `MainNetwork` but, on later tasks, it is the controller that learns to use the $\alpha$ values to quickly adapt while the `MainNetwork` and `ConsolidatedNetwork` preserve mostly the same weight directions.** Furthermore, it is crucial to note that while the `MainNetwork` and `ConsolidatedNetwork` become highly aligned, the existence of both is essential for positioning the model in this “cancellation-and-steering” regime, where rapid adaptation is achievable solely by adjusting the modulation parameters. For further details, please refer to Appendix P.
> > >
> > > \* In the “cancellation-and-steering” regime (or sensitive regime), the combination of the `MainNetwork` and `ConsolidatedNetwork` has a small magnitude (norm), which drives the denominator of Equation 14 close to zero. Because this denominator is small, the angle between the `InferenceNetwork` weights and the `MainNetwork` weights becomes highly sensitive; it can change dramatically with even minute adjustments to the modulation parameters or the `MainNetwork`.
> > >
> > > ## On Active Forgetting
> > >
> > > Concerning the reviewer's proposed forgetting hypothesis, we found this idea intriguing and tried to investigate it. Our current results could be consistent with a form of "active forgetting" where the partial cancellation between the `ConsolidatedNetwork` and the `MainNetwork` weights removes some information about previous tasks.
> > > As a caveat, despite the apparent lack of common structure between tasks (randomizing labels), we conclude there must be useful information being preserved across tasks, as we observe an increase in the final accuracy over the sequence of tasks (faster learning on later tasks).
> > > As such, if there is a forgetting mechanism, it cannot simply be destroying all previous task information; it must strategically preserve certain pieces to enable faster learning and forward transfer.

---

> > > ### Author Response · Authors · 2025-11-27
> > >
> > > ## A Possible Explanation on The Dynamic of $\alpha_{WC}$ (Figure 4b)
> > >
> > > Based on these new insights, provided in the previous comment, we might be able to explain the initial dip in $\alpha_{\text{WC}}$ and the subsequent increase in $\alpha_{\text{WC}}$ followed by a raise in accuracy. At the beginning of training, the `ConsolidatedNetwork` experiences a lower update rate than the `MainNetwork` due to the slowly moving average. Consequently, their weight directions do not completely match. This misalignment prevents the controller from steering the combined networks toward a cancellation regime to achieve fast adaptation solely through modulation parameters.
> > >
> > > However, as training progresses, the `MainNetwork`'s weight direction remains relatively stable across tasks. This allows the `ConsolidatedNetwork` to align its direction with the `MainNetwork`. Once aligned, the controller can effectively adjust the modulation parameters to push the system toward the cancellation regime (resulting in an increase in $\alpha_{\text{WC}}$). This leads to fast adaptation achieved purely by changing modulation parameters, without needing to update the `MainNetwork` itself.

---

### Official Review · Reviewer_eP2Q · 2025-10-28

**Soundness:** 3
**Presentation:** 3
**Contribution:** 2
**Rating:** 6
**Confidence:** 4

**Summary:**

This paper addresses the challenges of plasticity loss and poor knowledge transfer in continual learning by introducing NeuMoSync, a novel architecture inspired by global neuromodulatory mechanisms in the brain. NeuMoSync utilizes a higher-level module that synthesizes current inputs and the network's historical state, allowing it to adaptively regulate activation dynamics and synaptic plasticity. Evaluated on a diverse set of CL benchmarks, the method demonstrated strong performance in retaining plasticity and achieved significant improvements in both forward and backward adaptation compared to existing approaches.

**Strengths:**

The paper's core strengths lie in its clarity, rigorous validation, and the impressive performance of the proposed NeuMoSync architecture.  NeuMoSync demonstrates significant performance gains over a wide array of baselines on diverse continual learning benchmarks. The paper also goes beyond raw accuracy and includes adaptation speed and knowledge transfer for more comprehensive quantification of models' performance. The paper's claims are supported by ablation studies which clarifies the benefit of each component of the model. The analysis of emergent behaviors provides valuable intuition into why the system succeeds.

**Weaknesses:**

1. The paper frames the proposed method as inspired by neuromodulatory mechanisms in the brain, but the link is very weak. The modulation in this work seems more closely linked to conditioning modules, such as FiLM [1], as opposed to being brain-like. The authors also claim the consolidated network is inspired by the memory consolidation process in the brain, but it's not clear to me how averaging weights updated with gradient-descent is tied to any known neural mechanism. While the authors have acknowledged this in the appendix (Appendix I.5), this remains a weakness given that "brain-inspired" seems to be the key motivation for the method.

2. The scalability of the method to larger networks for more complex tasks is unclear. The NeuroSync module needs to take the feature vectors of every neuron as input and produce the modulation coefficient for every neuron. It seems hard to scale this to larger networks, as acknowledged by the authors. One possibility would be to only modulate a subset of neurons in the large network, but that would require additional empirical experiments.

3. The focus of the paper is on quick relearning and adaptation (plasticity) but does not address the stability issue (i.e., catastrophic forgetting) in isolation. However, this is not made clear in the main text. It would be better to mention these limitations in the main text instead of in the appendix.

[1] FiLM: Visual Reasoning with a General Conditioning Layer

**Questions:**

In addition to the points above, I wonder if using global consolidation and plasticity factors $\alpha_{WC}$ and $\alpha_{SM}$ would degrade model performance, this could help clarify if neuron-specific consolidation is necessary.

---

> ### Author Response · Authors · 2025-11-22
>
> > The paper frames the proposed method as inspired by neuromodulatory mechanisms in the brain, but the link is very weak. The modulation in this work seems more closely linked to conditioning modules, such as FiLM [1], as opposed to being brain-like.
>
> We thank the reviewer for pointing out the connection to conditioning modules such as FiLM. We fully agree that, at the level of elementary operations, our controller uses the same primitives as many conditioning methods, namely, input-dependent multiplicative and additive factors. Our goal is not to claim that these mathematical operations themselves are biologically novel, but rather that the *way* they are orchestrated in NeuMoSync is functionally inspired by global neuromodulatory systems.
>
> Concretely, FiLM computes a set of scale-and-shift parameters $\gamma(h), \beta(h)$ from a conditioning input $h$, and applies them as a channel-wise affine transformation to intermediate feature maps of a single backbone network. In contrast, NeuMoSync's NeuroSync module operates at a different level and with a different purpose. First, it constructs effective synaptic weights for each neuron by dynamically combining a fast, plastic `MainNetwork` and a slow `ConsolidatedNetwork`, as described in Equation (1) in the paper. As a result, the balance between plasticity and long-term memory is explicitly controlled at the weight level.
>
> Another key difference is *what* the modulation is conditioned on. In FiLM, the modulation depends only on the current conditioning signal (e.g., a text embedding), whereas NeuroSync takes as input both the current sample and a feature vector for each neuron encoding its identity (layer/index), learnable features, and an EMA of its past activations. This makes the modulatory signal explicitly *state-dependent* and history-dependent, inspired by neuromodulatory systems that regulate plasticity and excitability based on ongoing network state rather than purely on the current stimulus [4].
>
> Additionally, several computational neuroscience and deep learning works described as neuromodulation-inspired also instantiate neuromodulation via such multiplicative/additive modulation. Examples include neuromodulated plastic networks where a learned modulatory signal gates Hebbian plasticity through a scalar factor [1], modulation layers that up- and down-regulate input channels based on reliability scores, motivated by cortical neuromodulation [3], and neuro-modulated Hebbian learning where feedback signals multiplicatively adjust neuron responses at test time [2].
>
> Our goal is similar: we do not claim a biophysically accurate model of any particular neuromodulatory system. Rather, we adopt the computational motif of neuromodulation, input-dependent gain/bias control to obtain an input-dependent “effective network.” In our case, this is achieved through the two scaling signals $\alpha_{\text{SM}}$ and $\alpha_{\text{WC}}$, which modulate two subnetworks with distinct functional roles.
>
> We will make this relationship to FiLM more explicit in the revised version by (i) adding FiLM and related conditioning modules to the hypernetwork/conditioning part of the related work, and (ii) clarifying in the main text that NeuMoSync is neuro-inspired at the *architectural and functional* level (global, state-dependent modulation of plasticity and activity), while still using standard multiplicative/additive primitives that are also present in prior deep learning models.
>
>
> [1] Salinas E, Sejnowski TJ. Gain modulation in the central nervous system: where behavior, neurophysiology, and computation meet. Neuroscientist. 2001 Oct;7(5):430-40. doi: 10.1177/107385840100700512. PMID: 11597102; PMCID: PMC2887717.
>
> [2] Miconi, Thomas, et al. "Backpropamine: training self-modifying neural networks with differentiable neuromodulated plasticity." arXiv preprint arXiv:2002.10585 (2020).
>
> [3] Abdelhack, Mohamed, et al. "A modulation layer to increase neural network robustness against data quality issues." arXiv preprint arXiv:2107.08574 (2021).
>
> [4] Nadim, Farzan, and Dirk Bucher. "Neuromodulation of neurons and synapses." Current opinion in neurobiology 29 (2014): 48-56.

---

> > ### Author Response · Authors · 2025-11-22
> >
> > > The authors also claim the consolidated network is inspired by the memory consolidation process in the brain, but it's not clear to me how averaging weights updated with gradient-descent is tied to any known neural mechanism.
> >
> > To clarify our motivations, we are not aiming to exactly replicate the brain's neuronal structures. We only draw inspiration at a higher level. Specifically, utilizing an EMA as slow-moving weights has been described as *neuro-inspired* in other works [1, 2].
> >
> > Commonly, biological memory consolidation is modeled as a two-timescale learning process: a fast system (e.g., hippocampus) that rapidly incorporates new experiences, and a slower system (e.g., neocortex) that integrates these experiences over longer periods through replay and gradual synaptic change. This is precisely the level of abstraction that motivates our *consolidated network* construction.
> >
> > In our case, the fast learning corresponds to the usual gradient-descent updates on the task-specific network, while the slow consolidation is implemented by averaging these intermediate weight configurations into a more stable set of parameters. From a computational perspective, this averaging is a simple discrete approximation to the idea that neocortical synapses integrate many transient hippocampal updates over time, smoothing out noise in individual experiences and extracting stable regularities [3]. In other words, we are not claiming that the brain directly implements weight averaging, but that our method captures the computational role of systems consolidation: aggregating many short-term updates into a robust long-term representation.
> >
> > We clarified this point in the revised manuscript by explicitly stating that the connection is at the algorithmic/computational level rather than a detailed biophysical model in Section 1.
> >
> > [1] Lee, Hojoon, et al. *Slow and steady wins the race: Maintaining plasticity with hare and tortoise networks.* arXiv preprint arXiv:2406.02596 (2024).
> >
> > [2] Arani, Elahe, Fahad Sarfraz, and Bahram Zonooz. *Learning fast, learning slow: A general continual learning method based on complementary learning system.* arXiv preprint arXiv:2201.12604 (2022)
> >
> > [3] McClelland, James L., and Nigel H. Goddard. *Considerations arising from a complementary learning systems perspective on hippocampus and neocortex.* Hippocampus 6.6 (1996): 654–665.
> >
> > > The scalability of the method to larger networks for more complex tasks is unclear.
> >
> > We address this concern in the general response. Please refer to “Scaling to ResNet.”
> >
> > > The focus of the paper is on quick relearning and adaptation (plasticity) but does not address the stability issue (i.e., catastrophic forgetting) in isolation.
> >
> > Thank you for pointing this out. In response, we have (1) provided an explanation of our observation on the synergistic effect between our method and ER in the general response, and (2) provide additional clarification in the main paper, at the end of Section 3.4, to more clearly state our claim and clarify our contribution on plasticity based on the point you raised in this comment.
> >
> > > I wonder if using global consolidation and plasticity factors and would degrade model performance, this could help clarify if neuron-specific consolidation is necessary.
> >
> > Thanks for highlighting this insightful ablation. Based on your comment, we conducted an additional ablation study targeting the scenario you highlighted. Specifically, we applied shared global $\alpha_{\text{SM}}$ and $\alpha_{\text{WC}}$ parameters across the entire network, while preserving the two neuron-specific activity-modulation parameters ($\alpha_{\text{ARM}}$ and $\alpha_{\text{AL}}$) as in the original setup. This design isolates the effect of introducing global plasticity-modulation parameters. We evaluated this variant on two tasks, `Random-label CIFAR10` and `Shuffle CIFAR10`, as reported in Appendix M. As a brief summary, we observed that replacing fine-grained signals with a global modulation leads to a decrease in performance, which on `Shuffle CIFAR10` becomes a noticeable drop, particularly on the final tasks. However, performance remains within a relatively acceptable range and does not exhibit a catastrophic collapse. Taken together with the similarity in the modulation trends of neurons within a layer shown in Figures 9 and 10, this ablation suggests a potential path to improve the scalability of our method, especially when combined with the controller introduced in Section 3.7, by moving toward population-level modulatory signals, opening opportunities for future work on designs that are more efficient in both performance and runtime.

---

> > > ### Comment · Reviewer_eP2Q · 2025-11-24
> > >
> > > Thanks for the clarification and the extensive additional results, but the connection to biology still appears weak to me. In my view, relying too heavily on a “brain-inspired” framing creates more confusion than value, and may even mislead future work by stressing a biological grounding that only holds at a very high level. I do think the experiments are solid, and I will maintain my original score.

---

> > > > ### Author Response · Authors · 2025-11-27
> > > >
> > > > > Thanks for the clarification and the extensive additional results, but the connection to biology still appears weak to me. In my view, relying too heavily on a “brain-inspired” framing creates more confusion than value, and may even mislead future work by stressing a biological grounding that only holds at a very high level. I do think the experiments are solid, and I will maintain my original score.
> > > >
> > > > We sincerely thank the reviewer for their insightful feedback and for recognizing that our experiments are solid. We appreciate the direction you have pointed us toward regarding the framing of the paper. We totally agree that the strength of this work lies in its computational results and that the biological motivation should serve only as a high-level abstraction, rather than implying empirical grounding.
> > > >
> > > > In response, we have substantially modified the neuroscience parts of the manuscript to address this concern, ensuring the focus remains on the engineering contribution. Given that your reservation stems primarily from the framing rather than the method's performance, we earnestly request that you review these specific revisions and consider if they alleviate your concerns sufficiently to reconsider your score.
> > > >
> > > > To prevent any confusion or risk of misleading future work, we have realigned the revised version as follows:
> > > >
> > > > *   **Clarifying Scope and Motivation (Section 1):** We have refined the introduction (specifically the third paragraph) to clearly scope our contribution. This ensures the reader immediately approaches NeuMoSync as a machine learning solution inspired by high-level principles.
> > > >
> > > > *   **Prioritizing Continual Learning Rationale (Section 2):** We have restructured the Method section to ground our design choices in established Continual Learning literature first. We now introduce the modulation parameters by linking them to known mechanisms: for instance, connecting $\alpha_{\text{WC}}$ to dual-memory CL methods like [1,2], and $\alpha_{\text{ARM}}$ to feature-wise shifting operations similar to those in FiLM [3]. We have removed stronger biological claims and now explicitly label the remaining high-level parallels as "loose conceptual resemblances" or "speculative analogies" to ensure they are not interpreted as mechanistic explanations.
> > > >
> > > > *   **Reframing Analysis (Section 3.6):** In our analysis of emergent behaviors, we have moderated the language to avoid implying direct biological equivalence. The discussion now focuses on the model's computational dynamics (e.g., its reaction to loss spikes), ensuring the results are interpreted through a computational lens.
> > > >
> > > > *   **Appendix and Related Work:** We have reviewed the Appendix to ensure a consistent tone. We emphasize that "neuro-inspired" refers strictly to the high-level architectural principle of global modulation, removing speculative claims to avoid giving an unintended impression of empirical grounding in neuroscience.
> > > >
> > > > We believe these revisions have significantly improved the clarity of the paper and removed the risk of confusion, allowing the experimental results to stand on their own. We hope you find this revised version to be a more precise representation of the work.
> > > >
> > > > [1] Arani, Elahe, Fahad Sarfraz, and Bahram Zonooz. "Learning fast, learning slow: A general continual learning method based on complementary learning system." arXiv preprint arXiv:2201.12604 (2022).
> > > >
> > > > [2] Sarfraz, Fahad, Elahe Arani, and Bahram Zonooz. "Synergy between synaptic consolidation and experience replay for general continual learning." Conference on lifelong learning agents. PMLR, 2022.
> > > >
> > > > [3] Perez, Ethan, et al. "Film: Visual reasoning with a general conditioning layer." Proceedings of the AAAI conference on artificial intelligence. Vol. 32. No. 1. 2018.

---

### Official Review · Reviewer_H3S2 · 2025-11-02

**Soundness:** 4
**Presentation:** 4
**Contribution:** 3
**Rating:** 6
**Confidence:** 4

**Summary:**

This paper proposes a biologically inspired continual learning architecture, NeuMoSync, which introduces three core components: a Main Network (for rapid adaptation to new tasks), a Consolidated Network (for long-term memory), and a NeuroSync module (as a global regulator). NeuMoSync dynamically modulates neuron-level plasticity, activation functions, and synaptic weights to address the loss of plasticity and inefficient knowledge transfer commonly observed in deep neural networks during continual learning. Across six categories of continual learning benchmarks, NeuMoSync demonstrates superior performance in maintaining plasticity and achieving fast forward/backward adaptation compared to existing methods. Systematic experiments further reveal that the modulation parameters exhibit neuroscience-like behaviors, such as dopamine-like responses during task switching and neuron functional specialization, which validates the biological plausibility of the proposed approach.

**Strengths:**

This paper presents an innovative integration of neuromodulatory mechanisms with continual learning, achieving a dynamic balance between plasticity and stability through neuron-level modulation. The experimental design is rigorous and extensive, covering six benchmark types and multi-dimensional metrics, such as plasticity, adaptation speed, generalization. And the comparisons with meta-learning approaches and stability-enhanced methods  further demonstrate the method's robustness. Besides, the emergence of neuroscience-like modulation behaviors (e.g., dopamine-like responses and neuron specialization) observed through systematic experiments strengthens the credibility and interpretability of the biologically inspired design. Moreover, the paper features a clear structure with detailed methodological descriptions. Illustrations such as architecture diagrams, learning curves, and modulation parameter analyses provide intuitive support for key arguments.

**Weaknesses:**

Although the paper mentions that the parameter overhead of NeuMoSync is only 5–8%, the scalability of the NeuroSync module, which relies on Transformer-based network, is not sufficiently discussed for networks such as ResNet. Besides, the forgetting experiments (Appendix F.3) depend on experience replay, which does not directly demonstrate NeuMoSync’s intrinsic ability to mitigate Catastrophic Forgetting. Moreover, some biological analogies (e.g., αARM as “tonic neural modulation”) lack direct empirical validation, which weakens the persuasiveness of these claims.

**Questions:**

How can NeuMoSync be extended to very large-scale architectures?

Would adjustments to the neuron grouping strategy or sparsification in NeuroSync be necessary?

Are the observed phenomena such as the “dopamine-like responses” of modulation parameters supported by neuroscientific experiments results, or are they qualitative analogies?

The paper states that “in this manner enables input-dependent amplification or attenuation of each network’s contribution within the Inference Network.” in line 171-172. Could the authors elaborate on the implementation details of this mechanism?

In Lines 185-188, is there corresponding ablation experiments results supporting the described behavior

---

> ### Author Response · Authors · 2025-11-22
>
> > Although the paper mentions that the parameter overhead of NeuMoSync is only 5–8%, the scalability of the NeuroSync module, which relies on Transformer-based network, is not sufficiently discussed for networks such as ResNet.
>
> We thank the reviewer for raising this important point regarding scalability, which is critical for the practical applicability of our method. We address this concern in the General Response, under the section “Scaling to ResNet.”
>
> > Besides, the forgetting experiments (Appendix F.3) depend on experience replay, which does not directly demonstrate NeuMoSync’s intrinsic ability to mitigate Catastrophic Forgetting.
>
> We would like to draw your attention to the General Response, under “Forgetting,” where we provide an explanation and clarification of the forgetting experiments and our contribution in this setting.
>
> > The paper states that “in this manner enables input-dependent amplification or attenuation of each network’s contribution within the Inference Network.” in line 171-172. Could the authors elaborate on the implementation details of this mechanism?
>
> In the architecture, we assign two synaptic modulation coefficients to each neuron: one for its instance in the `MainNetwork` and one for the corresponding neuron in the `ConsolidatedNetwork`. Each coefficient is multiplied with its corresponding neuron's weights.
>
> All modulation coefficients are produced by a controller that is conditioned on both the current input and the network state (captured by each neuron’s learnable features and statistics of its activity). Consequently, when $\alpha_{\text{WC}}$ is large, or when $\alpha_{\text{SM}}$ has a large magnitude, the contribution of the corresponding network's neurons is amplified; when either coefficient is close to zero, the contribution of that network is effectively suppressed. For example, in Figure 9, $\alpha_{\text{WC}}$ for the first layer is close to zero for almost all neurons, while in the second layer it takes non-zero values, mirroring attenuation and amplification of the `ConsolidatedNetwork`, respectively. We further elaborate on this in revised Section 3.6, where we provide additional insight into the dynamics between $\alpha_{\text{SM}}$ and $\alpha_{\text{WC}}$.
>
> At the implementation level, the `InferenceNetwork` can be constructed in two equivalent ways. One option is to use vectorized operations to compute the final effective weights of the network in a single step. Alternatively, the network can be built layer by layer by combining the weights of the `MainNetwork` and `ConsolidatedNetwork` with the modulation outputs of the controller. The first approach places greater pressure on GPU VRAM when we exploit the GPUs’ parallel matrix multiplication capabilities, whereas the second approach increases runtime because all layers must be constructed serially. Once the effective weights are obtained, standard functional implementations of convolutional and fully connected layers can be used to compute the activations at each layer.
>
> > In Lines 185-188, is there corresponding ablation experiments results supporting the described behavior
>
> Thank you for pointing this out. Following your comment, we added an ablation study in which we disable the $\alpha_{\text{AL}}$ mechanism and examine its effect on the gradient in each layer (see Appendix N). As anticipated, removing $\alpha_{\text{AL}}$ results in degraded gradient flow, evidenced by lower gradient magnitudes across both layers and tasks (`Random-Label CIFAR10` and `Shuffle CIFAR10`), as well as an increased fraction of zero gradients compared to the original method. This reduction in effective gradient flow is consistent with our previous ablation studies and manifests as reduced plasticity and lower accuracy, as analyzed in Figure 3 in Section 3.5.

---

> > ### Author Response · Authors · 2025-11-22
> >
> > > Moreover, some biological analogies (e.g., αARM as “tonic neural modulation”) lack direct empirical validation, which weakens the persuasiveness of these claims.
> > > Are the observed phenomena such as the “dopamine-like responses” of modulation parameters supported by neuroscientific experiments results, or are they qualitative analogies?
> >
> > We first want to clarify that the design of our method is inspired by neuronal mechanisms but we do not, a priori, expect quantitative resemblance between the two. Nonetheless, we find that there may be evidence of an interesting parallel between the evolution of modulation signals and neural modulation which we expand on below. We also do not present any new neuroscience experiments; instead, our interpretations are purely qualitative and draw on previously reported findings on neuromodulation, in particular its input-dependent functional effects.
> >
> > In our continual learning setup, we are especially interested in neuromodulatory behavior in two cases: (i) when the model encounters a novel situation, and (ii) when the loss increases due to a distribution shift, which can be viewed as an unexpected negative outcome. In our experiments, both situations arise at the beginning of each new task, when the underlying data distribution changes.
> >
> > For the first case (novelty), we added a new section in Appendix L that discusses a potential qualitative alignment between the behavior of our method and selected findings from the neuroscience literature. In the `Random-label CIFAR10` benchmark (used here as a representative setting), we observe that $\alpha_{\text{ARM}}$ tends to increase at the beginning of each task and then gradually return to a more stable value as training progresses. This temporal profile can be loosely compared to descriptions of norepinephrine (NE) dynamics and tonic neuronal activity [1, 2, 4]: in response to novel or unexpected stimuli, NE levels have been reported to transiently increase in specific brain regions and then decline due to diffusion and regulatory mechanisms. The initial surge has been suggested to lower firing thresholds in neurons with the appropriate receptors [4], thereby increasing their tonic firing rate even in the absence of strong sensory input; as the environment becomes familiar and NE levels recede, neuronal activity tends to return toward a baseline tonic state.
> >
> > In our method, we see a pattern that we interpret as qualitatively similar at a high level: increases in $\alpha_{\text{ARM}}$ at task boundaries are followed by increases in neuronal activity when the model encounters novel tasks. At the same time, NE’s effects in the brain are known to be cell-type and receptor-specific, and thus do not uniformly elevate tonic firing [3]. In a loose analogy, although $\alpha_{\text{ARM}}$ varies across and within layers in our model, it generally induces a positive offset in many neuron states rather than a uniform effect. As illustrated in Figures 9 and 10, with additional details in Figures 17 and 18 in the newly added Appendix L, the generated $\alpha_{\text{ARM}}$ values are heterogeneous across neurons and layers, which we tentatively view as qualitatively consistent with this notion of neuron-specific variability.
> >
> > For the second case (unexpected negative outcomes), it is well established that a negative prediction error is associated with a dip in the phasic activity of dopaminergic neurons [5]. This reduction in dopaminergic activity can lead to decreased activity in neurons on which dopamine is normally excitatory, which, under Hebbian learning rules, has been linked to reduced synaptic plasticity. In our method, we also observe a dip-like phasic change in $\alpha_{\text{SM}}$ at the onset of new tasks, where the distribution shift induces a transient drop in accuracy, shown in Figure 4(a). We interpret this, again only at a qualitative level, as analogous to a down-regulation of plasticity in response to unexpected negative outcomes, as analyzed in Section 3.6.
> >
> > We hope this additional analysis clarifies how we intend these analogies: as suggestive qualitative correspondences that help interpret the learned modulatory signals, not as direct claims of biological fidelity.
> >
> > [1] Craig W Berridge, et al., *The locus coeruleus-noradrenergic system: modulation of behavioral state and state-dependent cognitive processes* (2003)
> >
> > [2] Susan J. Sara, *The locus coeruleus and noradrenergic modulation of cognition* (2009)
> >
> > [3] Radnikow, Gabriele, and Dirk Feldmeyer. "Layer-and cell type-specific modulation of excitatory neuronal activity in the neocortex." *Frontiers in neuroanatomy* 12 (2018): 1.
> >
> > [4] Zhang, Zizhen, et al. "Norepinephrine drives persistent activity in prefrontal cortex via synergistic $\alpha_1$ and $\alpha_2$ adrenoceptors." *PloS One* 8.6 (2013): e66122.
> >
> > [5] Schultz, Wolfram. "Dopamine reward prediction error coding." *Dialogues in Clinical Neuroscience* 18.1 (2016): 23–32.

---

> > > ### Comment · Reviewer_H3S2 · 2025-11-23
> > >
> > > In summary, while I appreciate the additional text and explanation, the rebuttal does not fully address the conceptual concern. The biological analogies remain weakly justified, risk being misinterpreted as mechanistic claims, and do not have a clearly articulated role in the model’s design or interpretation.
> > >
> > > I strongly encourage the authors to:
> > > 	•	reduce the biological claims substantially,
> > > 	•	explicitly label all analogies as speculative,
> > > 	•	avoid giving the impression of empirical grounding,
> > > 	•	and clearly delineate what insight (if any) the analogy contributes.
> > >
> > > Unless these issues are addressed in a revised submission, I cannot increase my score based on the current rebuttal.

---

> > > > ### Author Response · Authors · 2025-11-27
> > > >
> > > > > The biological analogies remain weakly justified, risk being misinterpreted as mechanistic claims, and do not have a clearly articulated role in the model’s design or interpretation.
> > > >
> > > > We sincerely thank the reviewer for their thoughtful guidance. We appreciate your perspective that the biological analogies, while serving as the initial inspiration for our design, must be carefully framed to avoid being interpreted as mechanistic claims. We agree that the primary contribution of NeuMoSync lies in its computational efficacy for Continual Learning, and that sharpening the distinction between its deep learning goals and biological inspiration significantly improves the clarity of the paper.
> > > >
> > > > In line with your recommendations, we have substantially revised the neuroscience parts in the manuscript to explicitly frame them as high-level conceptual inspiration rather than empirical grounding. To be more specific, we have modified the following sections in the revised paper to address your concerns.
> > > >
> > > > *   **Clarifying Scope and Motivation (Section 1, Introduction):** We have refined the introduction (specifically the third paragraph) to clearly articulate the scope of our contribution. This ensures the reader understands immediately that the architecture is designed for computational advantage, using biology as a high-level conceptual inspiration.
> > > >
> > > > *   **Grounding Methodology in Deep Learning Literature (Section 2):** We have modified the Method section to prioritize the continual learning rationale for each modulation parameter. We now introduce the alpha parameters by first linking them to established mechanisms in the literature. For instance, we explain $\alpha_{\text{WC}}$’s grounding in the context of dual-memory CL methods like [1,2], or $\alpha_{\text{ARM}}$ to feature-wise shifting operations similar in FiLM [3]. By establishing this computational grounding first, we removed a number of biological connections and now also present the remaining as "loose conceptual resemblances", ensuring they are viewed as speculative analogies rather than functional explanations.
> > > >
> > > > *   **Speculative Framing of Emergent Behaviors (Section 3.6):** In our analysis of the learned dynamics, we have moderated the language to avoid implying direct biological equivalence. This keeps the focus on the model's actual computational behavior (e.g., reacting to loss increase).
> > > >
> > > > *   **Appendix and Related Work:** We have similarly reviewed the Appendix to ensure a consistent tone throughout the manuscript. We emphasize that our use of "neuro-inspired" refers strictly to the high-level architectural principle, specifically, the use of global, state-dependent signals to modulate local activity, and describe any biological resemblance in the additional experiments as speculative.
> > > >
> > > > We believe these revisions effectively address your concerns by ensuring the paper is judged on its empirical merits in Continual Learning, while keeping the biological context properly scoped as high-level inspiration. We hope these changes allow you to reconsider your score.
> > > >
> > > > [1] Arani, Elahe, Fahad Sarfraz, and Bahram Zonooz. "Learning fast, learning slow: A general continual learning method based on complementary learning system." arXiv preprint arXiv:2201.12604 (2022).
> > > >
> > > > [2] Sarfraz, Fahad, Elahe Arani, and Bahram Zonooz. "Synergy between synaptic consolidation and experience replay for general continual learning." Conference on lifelong learning agents. PMLR, 2022.
> > > >
> > > > [3] Perez, Ethan, et al. "Film: Visual reasoning with a general conditioning layer." Proceedings of the AAAI conference on artificial intelligence. Vol. 32. No. 1. 2018.

---

> > ### Comment · Reviewer_H3S2 · 2025-11-23
> >
> > I would still like to understand the scalability of NeuroSync when extended to Transformer-based architectures. The “General Response” section mentions the extension to ResNet, but does not address this point.

---

> > > ### Author Response · Authors · 2025-11-27
> > >
> > > > I would still like to understand the scalability of NeuroSync when extended to Transformer-based architectures
> > >
> > > We are grateful for engaging in the discussion and for pointing this out. In the revised version, we directly evaluate NeuMoSync with a Transformer-based classifier by adding Appendix K.3 (Extending to Transformer). Specifically, we employ a Vision Transformer (ViT) and integrate NeuMoSync in three complementary ways:
> > > (i) `ViT_NeuMoSync-MLP`: we modulate the neurons in the MLP blocks at the end of each encoder layer (same mechanism as in Sections 2 and 3.7);
> > > (ii) `ViT_NeuMoSync-Head`: we apply $\alpha_{\mathrm{SM}}$ and $\alpha_{\mathrm{WC}}$ to control each attention head's reliance on the current parameters versus an EMA "consolidated" version of that head's $Q/K/V$ matrices;
> > > (iii) `ViT_NeuMoSync-Both`: we combine both mechanisms.
> > > For these variants, we treat each attention head as a unit controlled by NeuMoSync and construct its feature vector using the same components as in the main method (learnable embedding, positional information, and EMA activity statistics).
> > >
> > > Interestingly, on Random-Label CIFAR-10 (where a vanilla ViT exhibits loss of plasticity, as previously reported in Table 2 in Section 3.5.2), all three NeuMoSync-enhanced ViT variants substantially mitigate the plasticity loss; the best-performing variant is `ViT_NeuMoSync-Both`, achieving faster accuracy gains and higher final-task accuracy. Extending this to the Shuffle Mini-ImageNet benchmark with a larger ViT backbone (the same which is used in Section 3.7), `NeuMoSync` continues to demonstrate improvements over the `Vanilla ViT` in performance, displaying that the method scales effectively with negligible parameter overhead (approximately 1% of the whole network in Random-Label CIFAR-10 and 0.002% in Shuffle Mini-ImageNet). Overall, these experiments provide evidence that the learned global modulatory signals can be integrated into attention-based Transformer architectures and improve the plasticity and adaptability while keeping the parameter overhead low.

---

### Official Review · Reviewer_uQUk · 2025-11-03

**Soundness:** 3
**Presentation:** 2
**Contribution:** 3
**Rating:** 6
**Confidence:** 3

**Summary:**

NeuMoSync is quite a complex architecture. The authors successfully demonstrate that the model does not lose plasticity on online continual learning tasks. The model can not only adapt quickly to new tasks but can also re-adapt quickly to previously trained tasks. A central component is the NeuroSync controller, which takes the current input and a feature vector for every neuron in the main network and outputs coefficients that govern how the inference network behaves.

It’s not clear to me how exactly the NeuroSync controller learns to produce effective coefficients for the inference network, given that there is no meta-learning loop. But it’s great that it does! The authors show that parameter sharing in the controller’s architecture is crucial, though I’m still not quite sure about the reasons for this.

Recommendation: Weak accept. The core idea is interesting, the empirical results are encouraging, and the ablation study suggest that all the components of the architecture matter. I did have a hard time grasping what was done at first. I do think the presentation could be improved. Figure readability is an issue, and quite a lot of important details are deferred to the appendix, but I don’t see fundamental issues that would block publication.

**Strengths:**

The authors demonstrate that NeuMoSync works for online continual learning and appears to enable fast adaptation and re-adaptation.
Results in Figure 2 and Table 1 are positive, even if some metrics are not immediately intuitive.
The ablation studies indicate that removing components degrades performance, supporting the claim that each part of the architecture matters.

**Weaknesses:**

For the results, it’s hard for me to know how impressive the performance is from the information about the comparisons provided. It would be helpful to know parameter counts for each of the comparisons and a little bit more information about the choice of hyperparameters. A lot of this information is in the appendix, but I think some of it should be moved to the main text if possible.
Some figures (especially Figure 4) are hard to read due to small text, and several practical details are mostly in the appendices, making it harder to judge the main claims from the body alone.

**Questions:**

One thing I’m curious about is that the controller requires the current input: how important is this? It would be interesting to see an ablation without this. (It may already be in the appendix, I may have missed it.)

---

> ### Author Response · Authors · 2025-11-22
>
> > It would be helpful to know parameter counts for each of the comparisons and a little bit more information about the choice of hyperparameters. Some figures (especially Figure 4) are hard to read due to small text, and several practical details are mostly in the appendices, making it harder to judge the main claims from the body alone.
>
> We thank the reviewer for their precise feedback. We agree that bringing more key practical and experimental details into the main body and improving figure legibility are crucial for a clear presentation of our work. In our revised manuscript, we have addressed this directly by adding two new paragraphs at the beginning of Section 3 (Experiments), just before Section 3.1. These additions provide a clear overview of the architectures and parameter counts for all compared methods.
>
> We adopt different network architectures depending on the difficulty and scale of the datasets. Specifically, for benchmarks based on `MNIST` and `CIFAR10`, we use a two-layer MLP with hidden widths of [100, 100]. For experiments on `Tiny-ImageNet` and `CIFAR100`, we employ a four-layer CNN with 8, 16, 32, and 64 filters, respectively. We additionally use a ResNet-18 model for the experiments presented in Section 3.7.
>
> Except for the ResNet-18 classifier, all experiments utilize the controller described in Section 2. For the ResNet-based classifier, we use the controller described in Section 3.7. In experiments using the full-input controller, the controller accounts for approximately 5–8% of the total network parameters, whereas in experiments using the sparse controller, it constitutes only about 0.002% of the overall parameters.
>
> More specifically, the MLP with input size $1 \times 28 \times 28$ and two hidden layers $100 \rightarrow 100 \rightarrow 10$ has 89,610 base parameters, to which the controller adds 6,222 parameters. The MLP with input size $3 \times 32 \times 32$ and the same topology has 318,410 base parameters and 17,398 controller parameters. For the CNN architectures, the $(8, 16, 32, 64)+\text{FC }200$ classifier has 75,928 base parameters with 6,702 additional controller parameters, while the $(8, 16, 32, 64)+\text{FC }100$ variant has 50,228 base parameters and 4,406 controller parameters.
>
> We recognize that the original Figure 4 was difficult to read due to small text and inconsistent axes, hindering the interpretation of the model's emergent behaviors. We have completely updated the figure in the revised version. Also, in the revised image, the x-axes are aligned and the font sizes are increased.
>
> Finally, we have performed a thorough review of the manuscript to ensure all claims are well-supported and that every reference to an Appendix is clear and accurate, guiding the reader to the correct supplementary details when needed. We are, of course, open to incorporating further details if the reviewer feels it would be beneficial.
>
> >One thing I’m curious about is that the controller requires the current input: how important is this? It would be interesting to see an ablation without this.
>
> Thank you for highlighting this important ablation. Because our primary motivation is to study modulatory effects, typically driven by stimulus-dependent signals, we primarily focused on input-dependent controllers in our method. To investigate this question more thoroughly, we additionally evaluated a variant of our approach in which the controller is conditioned solely on neuron feature vectors, using the `Random-label CIFAR10` dataset. The corresponding results and discussion are provided in Appendix O of the revised paper. As a brief summary, removing the image input from the controller harms performance on both benchmarks (`Shuffle CIFAR10` and `Random-label CIFAR10`), with a particularly large impact in the memorization setting. On `Random-label CIFAR10`, the input-conditioned controller sustains high training accuracy over tasks, whereas the input-unconditioned version shows a gradual decline as new tasks arrive, reflecting a clear reduction in plasticity. On `Shuffle CIFAR10`, both variants perform well overall, but the input-conditioned controller consistently outperforms its unconditioned counterpart. Overall, these findings indicate that input conditioning is crucial for maintaining plasticity, providing a modulatory signal that enables stable learning across long task sequences.

---

### Author Response · Authors · 2025-11-22
**General Response**

We sincerely thank the reviewers for their detailed and constructive feedback. We conducted additional experiments and updated the manuscript to directly address each of their concerns.  First, we address their common concerns below. To preserve the current section numbering, most of the new experiments and ablations prompted by the reviews have been added in a section at the end of the Appendix; in the final version, these results will be integrated into the most relevant parts of the paper. Newly added text and sections in the revised version of the paper are highlighted in **red**.

# Scaling to ResNet

We fully agree that controller complexity is a key consideration when extending the approach to larger backbone models. To address this, we extended our framework to support a ResNet-18 backbone and introduced a more scalable controller architecture. In this variant, described in detail in Appendix K and evaluated in Section 3.7, we replace the encoder-only controller with a lightweight encoder–decoder structure inspired by [1]. The encoder processes a fixed number of randomly sampled neurons per input via self-attention, and all neuron feature embeddings (including positional and learnable components) attend to the encoder output through a single, lightweight cross-attention module. This design allows the controller to remain efficient even when paired with a larger network: when applied to ResNet-18, it adds only about 0.002% additional parameters, and its time complexity remains practical as the classifier size increases.

We evaluate this scalable variant on the `Shuffle Mini-ImageNet` benchmark, which is more challenging than our CIFAR-based benchmarks (more classes, more samples, and higher-resolution images) and therefore more appropriate for a ResNet-18–based `MainNetwork`. In this setting, our method with the new controller achieves substantially better performance than strong baselines such as continual backprop and a ViT with a comparable number of parameters. We also report parameter counts, FLOPs, and hyperparameters in Appendix K to make the computational overhead explicit.

Taken together, these new experiments provide direct evidence that our method can be extended to larger models while keeping the additional overhead minimal. While our current implementation relies on random neuron sampling, we agree that developing more structured or learnable neuron selection/grouping strategies is a promising direction for future work, and we see our controller as a foundation for such extensions.

[1] Andrew Jaegle, et al., *Perceiver IO: A General Architecture for Structured Inputs & Outputs* (2021)

# Forgetting

We thank the reviewers for highlighting the importance of forgetting. To clarify, our method has been designed to improve adaptability and plasticity, and we acknowledge that it does not possess explicit mechanisms to prevent forgetting. Accordingly, preliminary experiments showed that, without experience replay (ER), the method could still suffer from forgetting. However, with a small replay buffer, as shown in Appendix F.3, it consistently achieves strong stability compared to other plasticity-focused approaches that use the same replay buffer. In most cases, this combination even surpasses stability-oriented methods such as EWC and A-GEM, and in some cases performs favorably against HAT, despite HAT having access to privileged task information.

Importantly, while our inductive biases alone are not sufficient to fully prevent forgetting, the results in Tables 19 (forgetting) and 20–21 (plasticity) suggest that the method becomes very effective when paired with an explicit stabilization component such as ER. Under this setup, our method maintains high plasticity, evidenced by fast adaptation and minimal loss of learning capacity in this setting, while also offering competitive or better stability across nearly all evaluated benchmarks. These empirical results suggest that the introduced inductive biases produce a synergistic effect that cannot be achieved by ER alone. In particular, the inductive biases steer the model toward improved stability without sacrificing plasticity. Consequently, the method, when enhanced with ER, achieves the most favorable stability–plasticity trade-off across nearly all benchmarks when compared to the evaluated baselines.

---

### Author Response · Authors · 2025-12-03
**Authors' Rebuttal Summary (Part 1/2)**

As a brief summary, our method uses a global controller (`NeuroSync`) that, given the input and each neuron's state (learnable features, position, and activation statistics), outputs four neuron-wise modulation signals: two for plasticity and two for activity. Using a learnable `MainNetwork` with parameters $\theta$, a `ConsolidatedNetwork` with parameters $\phi$ whose weights are a slowly updated exponential moving average of $\theta$, and plasticity signals $\alpha_{\text{SM}}$ and $\alpha_{\text{WC}}$, it constructs an input-dependent `InferenceNetwork` with effective weights $w$ such that, for neuron $i$,
$
w^i = \alpha_{\text{SM}}^i \theta^i + \alpha_{\text{WC}}^i \phi^i.
$
The remaining two signals modulate activity by setting each neuron's `PReLU` parameters and an additive output bias in the constructed network.

To summarize the discussions, we would first like to highlight positive aspects identified by the reviewers. They referred to the submission as *“an innovative integration of neuromodulatory mechanisms with continual learning”* (Reviewer H3S2) and *“NeuMoSync demonstrates significant performance gains over a wide array of baselines on diverse continual learning benchmarks”* (Reviewer eP2Q). During the discussion period we addressed the reviewers' concerns through additional experiments and paper revisions. We outline five major points below:

**Scalability of the proposed method**
*Concern: The proposed method uses an attention mechanism over the neurons of a neural network. Does this pose a challenge for scaling to larger networks?*
*Response:*
Regarding the scalability concerns raised by Reviewers H3S2 and eP2Q, we proposed a new variant of our controller that scales to larger classifiers (Section 3.7 and Appendix K of the revised paper). Appendix K.1 describes this variant of the controller, which enables our global controller to scale to larger networks such as ResNet, as requested by Reviewers eP2Q and H3S2 (Appendix K.2) and Transformer-based architectures like ViT, as requested by Reviewer H3S2 (Appendix K.3). For these larger networks, we extended our experiments to a more challenging benchmark, `Shuffle Mini-ImageNet`, to better reflect the increased capacity of the classifier. Our results show substantial improvements over other methods.

**Clarifying connections between the algorithm design and biological neurons**
*Concern: The current text draws parallels between the function of biological neurons and the proposed algorithm. This can be misleading by suggesting a direct match between the two when there are only looser connections.*
*Response:*
We substantially reframed the neuroscience-related parts of the manuscript to foreground the engineering contribution and clarify that biological inspirations are used only as high-level abstractions. Specifically, we revised specific parts of the Introduction, Method, Experiments, and Appendix sections (highlighted in blue in the revised paper) to anchor our design choices in established continual learning literature, while explicitly labeling potential biological parallels as speculative, qualitative analogies to avoid any misconception or misinterpretation that they are empirically grounded claims.

**Forgetting in continual learning**
*Concern: How does NeuMoSync deal with the forgetting problem in continual learning?*
*Response:*
As raised by Reviewers H3S2, eP2Q, and BWK5, our method relies on Experience Replay (ER) to mitigate forgetting; however, we emphasized that `NeuMoSync` interacts with ER in a synergistic way, leading to better performance than plasticity-preservation methods equipped with the same ER and better or comparable performance to stability-oriented methods. This indicates that the proposed global controller introduces effective inductive biases for stability that become evident when ER is present. We clarified our contribution regarding forgetting in the main text.

---

> ### Author Response · Authors · 2025-12-03
> **Authors' Rebuttal Summary (Part 2/2)**
>
> **Additional analysis of $\alpha$ values**
> *Concern: Can you explain how the $\alpha$ values generated by the controller help learning? The analysis in Fig.4 is unclear.*
> *Response:*
> As confirmed by the reviewers, our method achieves strong empirical results and consistently outperforms competing approaches, with the performance gap being especially pronounced on memorization tasks. In the original submission (Section 3.6 and Figure 4), we provided a partial explanation for this superior performance. Inspired by a insightful discussion with Reviewer BWK5, we carried out a deeper and more detailed analysis. This careful study led to Appendix P, where we closely examine the evolution of the `MainNetwork` and `ConsolidatedNetwork` and their associated $\alpha_{\text{SM}}$ and $\alpha_{\text{WC}}$ values. Very briefly, we observed that after the initial tasks, the `MainNetwork` and `ConsolidatedNetwork` remain almost unchanged in terms of parameter direction, and it is the controller that learns how to modulate these networks to adapt to new tasks.
>
> Next, as requested by Reviewer H3S2, to more closely examine the effect of $\alpha_{\text{ARM}}$ on neurons in the constructed network, we empirically evaluated its impact on neuron activations across various layers on two benchmarks, `Random-Label CIFAR10` and `Shuffle CIFAR10`. The consistent behavior observed across these settings and layers reinforces our claim that this modulation has a predictable effect on neuronal activity. These results were added to Appendix L.
>
> Finally, we updated Figure 4 in the paper because inconsistencies between the subplots caused confusion, as pointed out by the Reviewer BWK5. Additionally, the figure was hard to read due to the small font sizes. The inconsistencies were originally due to each subplot emphasizing a particular emergent behavior of our method. We have now completely revised this figure to remove these inconsistencies, increase readability, and ensure that it is clear and easy to interpret.
>
> **Additional ablations**
> *Concern: Miscellaneous other ablations were requested.*
> *Response:*
> We conducted additional ablation studies that cover all of the requested analyses. Briefly, in Appendix M we evaluate our method when plasticity modulations are coarse-grained and shared across the entire network, as requested by Reviewer eP2Q. In Appendix N, we study the effect of $\alpha_{\text{AL}}$ on gradient flow, both in terms of magnitude and the fraction of zero gradients, in detail, clarifying the impact of this modulation on the gradients, as requested by Reviewer H3S2. Additionally, in Appendix O, we comprehensively investigate the effect of conditioning the controller on the input by comparing our original method with an input-unconditioned variant, providing experimental results on the two most challenging benchmarks in our suite, as requested by Reviewer uQUk.

---

### Meta-Review · Area_Chair_6vQA · 2026-01-02

**Summary:**

In their initial reviews, all four reviewers judged this paper to be either marginally above the acceptance threshold or marginally below it. Generally, the reviewers found the motivation and core idea of the paper interesting, and the empirical results encouraging. However, concerns that were raised that prevented the reviewers from clearly supporting acceptance of the paper were (1) the rather weak link between the neuroscience motivation and the proposed method, (2) that the paper largely ignores the forgetting issue, and (3) concerns regarding scalability of the method.

Of these three concerns, the first two mostly informed my suggested decision for this paper.

**Reviewer Concerns:**

Regarding the weak link between the neuroscience motivation and the proposed method, in the rebuttal the authors addressed this concern mostly by weakening their claims in this regard. In particular, the authors clarified that the neuroscience motivation and biological analogies are only relevant at a rather high, abstract level. To some extent this addressed reviewer concerns regarding over-claims, but at the same time it took away some of the paper's initial appeal. Both reviewer H3S2 and reviewer eP2Q indicate in their responses to the author rebuttal that they are not fully satisfied with the authors' rebuttal on this point.

Regarding the concern that the paper largely ignores the forgetting issue, the author's main response was to discuss (mostly in section 3.4) that they're method is designed to improve adaptability and plasticity, not to protect stability or prevent forgetting. Further, in the Appendix, there are some experiments showing that the proposed method can be combined with ER, and the authors claim that this can be done in a "synergistic way", but this last claim is not clearly discussed or extensively tested.
The reviewers do not indicate that they are convinced by this response by the authors. My own interpretation is that the author's response is not satisfactory.

**Reviewer Scores:**

My expectation is that if there had been a normal discussion phase, there would have not been much changes in the reviewer's scores. As discussed above, my impression is that two main concerns raised by the reviewers in their initial reviews were not satisfactorily addressed in the rebuttal.

---

### Decision · Program_Chairs · 2026-01-26

Reject